**EMBO** *reports*

# Uridine phosphorylase-1 supports metastasis by altering immune and extracellular matrix landscapes

Declan Whyte [1], Sophie L Fisher [1,2], Christopher G J McKenzie [1], David Sumpton [1,2], Sandeep Dhayade [1], Emmanuel Dornier [1], Madeleine Moore [1], David Novo [1], Jasmine Peters [1,2], Robert Wiesheu [1,2], Michalis D Gounis [1], Dale M Watt [1], John B G Mackey [1], Amanda J McFarlane[1], Frédéric Fercoq [1], Carolina Dehesa Caballero [2], Keara L Redmond[3], Louise E Mitchell [1], Eve Anderson [1], Gemma Thomson [1], Ann Hedley [1], William Clark [1], Shannen Leroi [2], Lindsey N Dzierozynski [4], Juan J Apiz Saab [4], Caroline A Lewis [5], Alexander Muir [4], Christopher J Halbrook [6,7], Douglas Strathdee [1], Rene Jackstadt [1], Colin Nixon [1], Philip Dunne [1,3], Leo M Carlin [1,2], Iain R Macpherson [2], Edward W Roberts [1,2], Seth B Coffelt [1,2], Karen Blyth [1,2], Owen J Sansom [1,2], Jim C Norman [1,2 ✉], Johan Vande Voorde [1,2 ✉] & Cassie J Clarke [1,2 ✉]

## Abstract

Understanding mechanisms that facilitate early events in metastatic seeding is key to developing therapeutic approaches to reduce metastasis. Here we identify uracil as a metastasis-associated metabolite in genetically engineered mouse models of cancer and in patients with metastatic breast cancer. Uracil is generated by the enzyme uridine phosphorylase-1 (UPP1), and we find that neutrophils are a significant source of UPP1 in metastatic cancer. Mammary tumours increase expression of adhesion molecules on the neutrophil surface, in a UPP1-dependent manner, leading to decreased neutrophil motility in the pre-metastatic lung. UPP1-expressing neutrophils suppress T-cell proliferation, and the UPP1 product uracil increases fibronectin deposition in the extracellular microenvironment. Knockout or inhibition of UPP1 in mice with mammary tumours increases T-cell numbers and reduces fibronectin content in the lung, and decreases the proportion of mice that develop lung metastasis. These data indicate that UPP1 influences neutrophil behaviour and extracellular matrix deposition in the lung, and suggest that circulating uracil could be a marker of metastasis, and that pharmacological inhibition of UPP1 could be a strategy to reduce recurrence.

**Keywords** Metastasis; Neutrophils; T Cells; Uridine Phosphorylase; Fibronectin
**Subject Categories** Cancer; Immunology; Metabolism

## Introduction

Metastasis is the process whereby cancer cells move from their original location in the primary tumour to distant sites in secondary organs. It is responsible for >90% of cancer-related deaths (Chaffer and Weinberg, 2011; Gupta and Massague, 2006) and thus represents a pressing area of unmet clinical need. Whilst metastatic disease may be detected at original diagnosis, more often it occurs as recurrent disease downstream of primary presentation. This is a significant problem for patients diagnosed with breast cancer, a heterogenous disease of different subtypes where relapse can occur across wide-ranging timescales from months to decades after primary tumour surgery (Riggio et al, 2021). Understanding mechanisms that facilitate metastasis is therefore important, not only to identify new targets for therapeutic treatment strategies, but to also identify the patients at highest risk of developing metastatic disease, ensuring high-risk patients are stratified for appropriate treatment regimens to reduce their risk of recurrence, whilst those with indolent disease may be spared aggressive over-treatment.

Metastasis occurs through a series of well-defined steps (Welch and Hurst, 2019), culminating in the outgrowth of cancer cells in distant organs. Factors released into the circulation by primary tumours can evoke alterations at metastatic target sites prior to the

[1]Cancer Research UK Scotland Institute, Garscube Estate, Switchback Road, Glasgow G61 1BD, UK. [2]School of Cancer Sciences, University of Glasgow, Garscube Estate, Switchback Road, Glasgow G61 1QH, UK. [3]Queen's University Belfast, University Rd, Belfast BT7 1NN, UK. [4]Ben May Department for Cancer Research, University of Chicago, Chicago, IL, USA. [5]Whitehead Institute for Biomedical Research, Cambridge, MA, USA. [6]Department of Molecular Biology and Biochemistry, University of California Irvine, Irvine, CA, USA. [7]Chao Family Comprehensive Cancer Center, University of California Irvine, Orange, CA, USA. ✉E-mail: j.norman@crukscotlandinstitute.ac.uk; johan.vandevoorde@glasgow.ac.uk; c.clarke@crukscotlandinstitute.ac.uk

arrival of disseminated cancer cells, leading to the formation of pre-metastatic niches which facilitate metastatic seeding. Whilst the immune system can contribute to tumour surveillance and elimination of incipient cancers, primary tumours can also evoke inflammatory responses which promote tumour growth and invasiveness. Furthermore, tumour-induced alterations in immune landscapes also contribute to the formation of pre-metastatic niches. For example, primary tumour-driven mobilisation of neutrophils from the bone marrow to distant target sites promotes vascular leakiness which supports extravasation (Shojaei et al, 2008; Shojaei et al, 2007), whilst increased numbers of neutrophils in metastatic target organs creates immunosuppressed microenvironments that enable disseminated cancer cells to evade immune challenge, thus facilitating metastatic outgrowth (Casbon et al, 2015; Coffelt et al, 2015; Jackstadt et al, 2019; Li et al, 2020b; Steele et al, 2016; Wculek and Malanchi, 2015). Neutrophils also play key roles in mediating cell-cell communication within pre-metastatic niches, attracting additional cell types and supplying endopeptidases to facilitate remodelling of the extracellular matrix (ECM) (Hirai et al, 2014). Indeed, structural changes in the composition, organisation and mechanical properties of the ECM can further contribute to the establishment of microenvironments that enable cancer cells to thrive (Hoye and Erler, 2016).

Altered metabolism can also influence metastasis, and whilst a predominant focus has been on the role of cancer cell-intrinsic metabolism in this process (Davis et al, 2020), the nature of the metastatic cascade means that the metabolism of other cell types in local and distant microenvironments may also be important. Resident mesenchymal cells in the pre-metastatic lung have been shown to promote metastasis of breast cancer through metabolic reprogramming of tumour cells and natural killer (NK) cells (Gong et al, 2022), whilst neutrophils, which are traditionally thought of as being highly glycolytic (Jeon et al, 2020), have been shown to engage mitochondrial metabolism to circumvent nutrient limitations and maintain immune suppression (Rice et al, 2018). As a consequence, altered immune landscapes (Barnes and Amir, 2017), ECM deposition (Riaz et al, 2012) and dysregulated metabolism (DeBerardinis et al, 2008) have all been associated with poor prognosis for cancer patients, and whilst these mechanisms may have the ability to influence disease independently, it is likely that they will also conspire to collectively facilitate metastatic progression.

This study began with the aim of identifying metabolites specifically involved in the metastatic process. In characterising serum metabolites from genetically engineered mouse models (GEMMs) of metastatic cancer, we found that hallmarks of the metastasis-associated circulating metabolome can be generated by mobilisation of pro-metastatic neutrophils. Specifically, we show that a key aspect of pyrimidine metabolism (uridine phosphorylase-1 (UPP1)-catalysed phosphorolysis of uridine into uracil and ribose-1 phosphate) influences priming of the lung metastatic niche, and consequent metastasis of mammary cancer to this organ. Mechanistically, our data highlight that neutrophils are a significant source of UPP1 and circulating levels of uracil in metastatic cancer, and demonstrate roles for UPP1 in influencing immune landscapes and ECM deposition in the metastatic lung. Importantly, we find that the ability of mammary cancer to metastasise to the lungs is reduced in the absence of UPP1. Our data, therefore, highlight a previously unknown and unexpected role for UPP1 in facilitating metastasis.

# Results

## Circulating uracil correlates with metastasis in mouse models of cancer

To characterise the circulating metabolome of metastatic cancer we performed liquid chromatography-mass spectrometry (LC–MS) on polar metabolites extracted from the serum of cancer GEMMs, focusing on previously well-characterised models of metastatic cancer (Appendix Fig. S1A) (Guy et al, 1992; Jackstadt et al, 2019; Morton et al, 2010). To model metastasis of mammary cancer we used the MMTV-*PyMT* mouse model, in which polyomavirus middle T antigen is expressed specifically in the mammary epithelium resulting in formation of mammary adenocarcinomas that recapitulate the progression of human breast cancer subtypes with poor prognoses (Attalla et al, 2021; Guy et al, 1992; Lin et al, 2003). Importantly, when expressed in an FVB/N mouse strain, this results in mammary tumours with high propensity to metastasise to the lungs (Mei et al, 2021). Comparing serum from MMTV-*PyMT* female mice bearing mammary tumours to age-matched non-tumour-bearing FVB/N controls, 214 compounds were identified as being significantly altered in mammary tumour-bearing animals (Appendix Fig. S1B). The extent of metastatic burden in the lungs of tumour-bearing mice was quantified, and a signature of circulating metabolites which correlated with metastasis—but not primary tumour burden—was identified (Fig. 1A; Appendix Fig. S1C; Appendix Table S1). This indicated that 17 compounds had positive, and 19 compounds had negative, correlations with metastatic burden independent from any correlation with primary tumour burden (Appendix Table S1). Parallel metabolic characterisation was also performed in GEMMs of other cancer types. For colorectal cancer, *villin*Cre^ER; *Kras*^G12D/+; *Trp53*^fl/fl; *R26*^N1icd/+ (KPN) mice were used, a model in which NOTCH1 activation is induced specifically in the intestinal epithelium to generate highly metastatic *Kras*^G12D-driven cancer; representative of the serrated colorectal cancer subtype (Jackstadt et al, 2019). KPN mice at clinical endpoint (100% metastatic penetrance) were compared with KPN mice at 30-, 60- and 90 days post induction. For pancreatic adenocarcinoma (PDAC), *Pdx1*-Cre; LSL-*Kras*^G12D/+; LSL-*Trp53*^R172H/+ (KPC) mice were compared to *Pdx1*-Cre; LSL-*Kras*^G12D/+; LSL-*Trp53*^fl/+ (KP^flC) mice. In these models, pancreas-specific expression of *Kras*^G12D leads to the development of pancreatic tumours with similar penetrance across both mouse cohorts. However, those with mutant p53 (KPC) develop highly metastatic disease, whereas those with p53 loss (KP^flC) develop PDAC that has a tendency not to metastasise (Morton et al, 2010). For KPC and KP^flC mice, comparisons were made at palpable primary tumour, a timepoint in which pro-metastatic structural changes are apparent in distant target organs but that precede the formation of overt metastases (Novo et al, 2018), providing the opportunity to identify metabolites that may contribute to metastatic niche priming. Assessment of serum LC–MS data generated from these GEMMs identified uracil as a metabolite that was consistently upregulated in metastatic mammary cancer (Fig. 1B,C) and colon cancer (Fig. 1D), and metastatic niche priming of PDAC (Fig. 1E). Furthermore, analysis of plasma from patients with metastatic breast cancer indicated that increased levels of circulating uracil were present in the circulation of breast cancer patients with metastatic disease by comparison with healthy

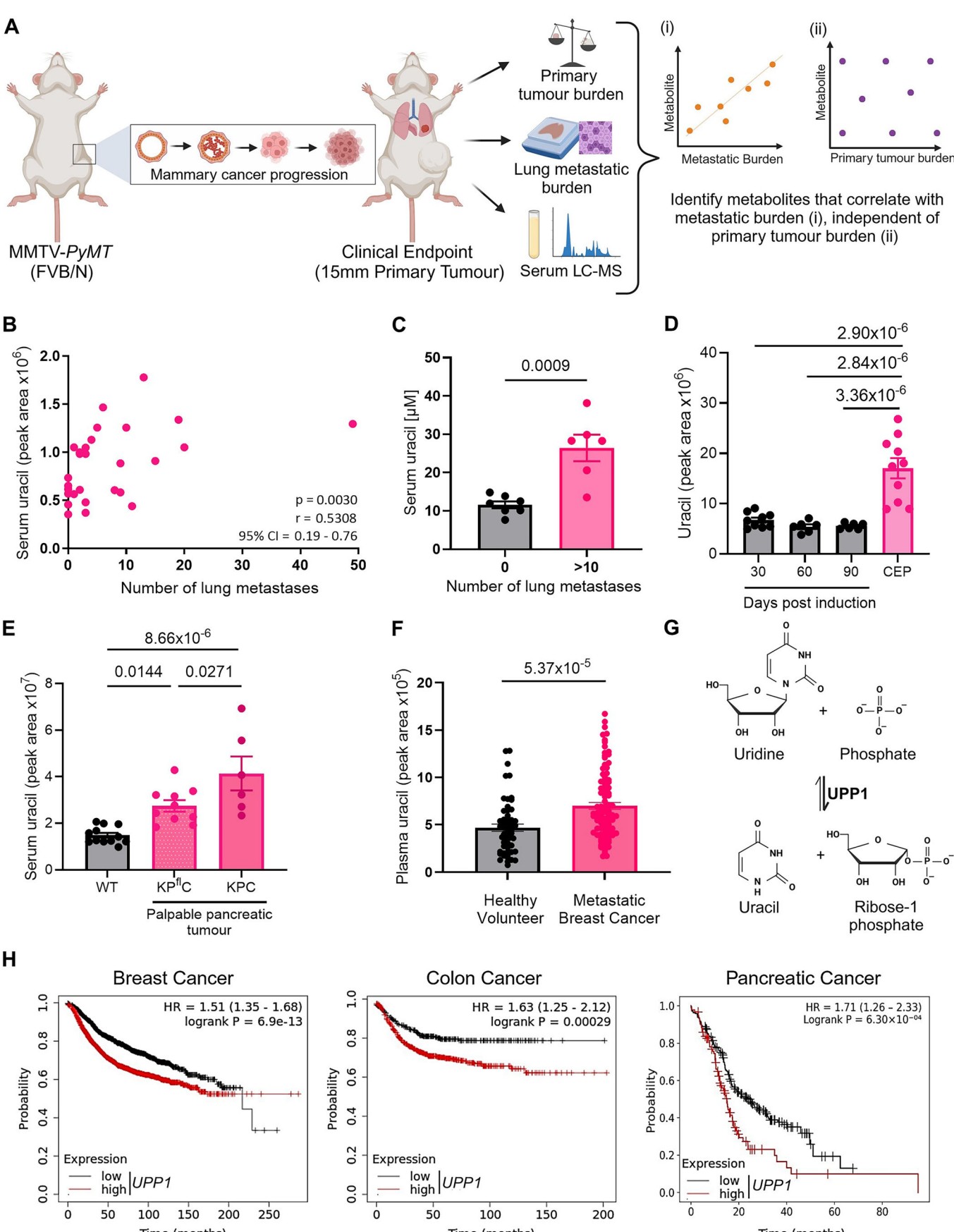

**Figure 1. The UPP1 product uracil positively correlates with metastasis in mouse models of metastatic cancer.**

(A) Schematic representation of the experimental approach used to identify serum metabolites in MMTV-*PyMT* mice with significant correlations with lung metastatic burden, independent of any correlation with primary mammary tumour burden. Examples shown are pictorial representations of the trend of interest. (B) Serum uracil levels measured by LC–MS from MMTV-*PyMT* tumour-bearing mice, plotted against the number of lung metastases ($n = 29$ MMTV-*PyMT* mice). (C) Absolute quantification of serum uracil levels in MMTV-*PyMT* mice with no identified lung metastases ($n = 7$ mice) and MMTV-*PyMT* mice with >10 lung metastases ($n = 6$ mice). (D) Serum uracil levels were measured by LC–MS in the KPN mouse model of colon cancer, comparing serum from tail bleeds at 30 ($n = 9$ mice), 60 ($n = 6$ mice) and 90 ($n = 6$ mice) days post induction to serum from mice at clinical endpoint (metastatic colon cancer, $n = 10$ mice). (E) Serum uracil levels were measured by LC–MS from KP^flC mice modelling poorly metastatic pancreatic cancer ($n = 10$ mice) and KPC mice modelling highly metastatic pancreatic cancer ($n = 6$ mice) at palpable pancreatic tumour, compared to age-matched wild-type (WT) controls ($n = 12$ mice). (F) Uracil levels were determined by LC–MS from plasma of human metastatic breast cancer patients ($n = 99$ patients) and healthy volunteers ($n = 56$ volunteers). (G) Schematic representation of the enzymatic reaction catalysed by uridine phosphorylase-1 (UPP1). (H) Increased expression of *UPP1* correlates with decreased relapse-free survival in human breast cancer and colon cancer and decreased overall survival in pancreatic cancer (Data source: Gyorffy, 2024a, 2024b). Data Information: In (B–F), each dot represents an individual mouse, or human, as appropriate. In (B), Spearman Correlation statistics are presented. In (C–F), data are presented as mean ± SEM, with unpaired *t* test for comparison of two groups (C, F), and one-way ANOVA for more than two groups (D, E). In (D, E), when one-way ANOVA produced $P < 0.0001$, *t* and DF values provided by Šídák's multiple comparisons test were used to calculate the exact *P* value. In (H), hazard ratio (HR) from the Cox model, and log-rank *P* value, are presented. Source data are available online for this figure.

volunteers (Fig. 1F). Of note, no correlation was observed between plasma uracil levels and clinical subtype of breast cancer, or organ of metastasis, in metastatic breast cancer patients (Appendix Fig. S2).

Uracil is one of the four primary nucleobases in RNA and, whilst pyrimidine nucleotides can be produced through de novo biosynthesis, uracil is a product of the pyrimidine salvage pathway, namely the UPP1-catalysed phosphorolysis of uridine into uracil and ribose-1 phosphate (Fig. 1G). Indeed, mice which lack the gene for *Upp1* (*Upp1^{-/-}* mice), display increased levels of uridine and decreased levels of uracil in tissues and serum (Appendix Fig. S3A–B) (Cao et al, 2005), whilst mammary tumours from MMTV-*PyMT* mice displayed reduced uracil, but unaltered uridine levels (Appendix Fig. S3C). Importantly, in human cancer patients, increased expression of *UPP1*, in tumour core biopsies, consistently correlated with decreased survival in breast, colon and pancreatic cancer (Fig. 1H) (Gyorffy, 2024a, 2024b). Collectively, these data suggest that UPP1-dependent generation of uracil is associated with metastasis, and poor prognosis, in several cancer types.

## Neutrophils are a source of UPP1 in metastatic cancer

To uncouple alterations in uracil that resulted from primary tumour growth from those attributable to metastasis, we orthotopically transplanted mammary tumour fragments obtained from *K14*-Cre; *Trp53^{fl/fl}* (KP) mice into the 4th mammary fat pad of *Upp1^{+/+}* and *Upp1^{-/-}* recipient mice and assessed circulating uracil levels when primary tumours were measurable but metastatic dissemination was not. This showed that primary mammary tumours alone were not sufficient to increase circulating uracil levels (Fig. EV1A). To determine whether tumour-intrinsic alterations in UPP1 were occurring during metastatic progression, and whether this was responsible for increased systemic uracil, we then compared the expression of *Upp1* in clinical endpoint MMTV-*PyMT* tumours to normal mammary gland tissue from FVB/N mice. No significant difference in expression of the mRNA encoding *Upp1* was detected between normal mammary gland and MMTV-*PyMT* tumours (Fig. EV1B). *Upp1* expression also did not differ between mammary tumours that came from mice that did or did not present with lung metastases (Fig. EV1C), nor was there any correlation between *Upp1* expression in the primary tumour and levels of serum uracil (Fig. EV1D). Expression of *Upp1* was also unaltered between cancer cell lines derived from primary

tumours grown in the mammary fat pad and cancer cell lines derived from lung micrometastases (Fig. EV1E)(Gounis et al, 2025). Collectively, these data suggest that the increased levels of circulating uracil previously characterised are not due to tumour-specific alterations in *Upp1* expression. We also considered whether dietary uracil could influence circulating uracil levels. However, dosing of mice by oral gavage daily with 10 mM uracil for a total of 4 days was unable to influence the levels of serum or tissue uracil (Fig. EV1F,G). Circulating uracil levels were also unchanged with age in FVB/N mice (Fig. EV1H), and age did not significantly contribute to metastatic burden in this cohort of MMTV-*PyMT* mice (Fig. EV1I), suggesting that UPP1 activity in the primary tumour is not responsible for increasing uracil levels in the circulation of mice with lung metastasis, and that age is not a factor in dictating serum uracil concentration at tumour endpoint in the MMTV-*PyMT* model of mammary cancer.

The innate immune system plays a well-characterised role in facilitating progression of metastatic disease (Tommasi et al, 2022). We, therefore, investigated the serum metabolome of mice injected with lipopolysaccharide (LPS), an established inflammatory trigger, to understand which circulating metabolites associated with metastatic cancer could be generated by activated immune cells. LPS administration led to a decrease in serum uridine and an increase in serum uracil, indicating that UPP1-activation occurs during mobilisation and/or activation of immune cells (Figs. 2A and EV2A). To further define the immune cell source of UPP1, transgenic mice expressing the diphtheria toxin receptor (DTR) under control of the CD11b promoter (CD11b-DTR) were treated with diphtheria toxin (DT) to deplete CD11b-expressing cells. As CD11b is a pan-myeloid marker, this enabled non-specific ablation of cells across the myeloid lineage. Depletion of myeloid cells, for only 24 h, resulted in a striking decrease in serum uracil (Fig. EV2B,C), suggesting that the myeloid compartment is responsible for a significant proportion of uracil detected in the circulation. To further delineate the cell type responsible for inflammatory, and metastasis, associated uracil production, we assessed *Upp1* in published scRNA-seq datasets. Given the positive correlation of uracil with lung metastases, we assessed *Upp1* expression in RNA-seq datasets of lung immune cells from tumour-bearing MMTV-*PyMT* mice (Data Ref: (McGinnis et al, 2024)). This indicated that *Upp1* was upregulated in neutrophils from the lungs of MMTV-*PyMT* tumour-bearing mice (Fig. 2B), and not in other immune cell types. To understand whether this neutrophil-

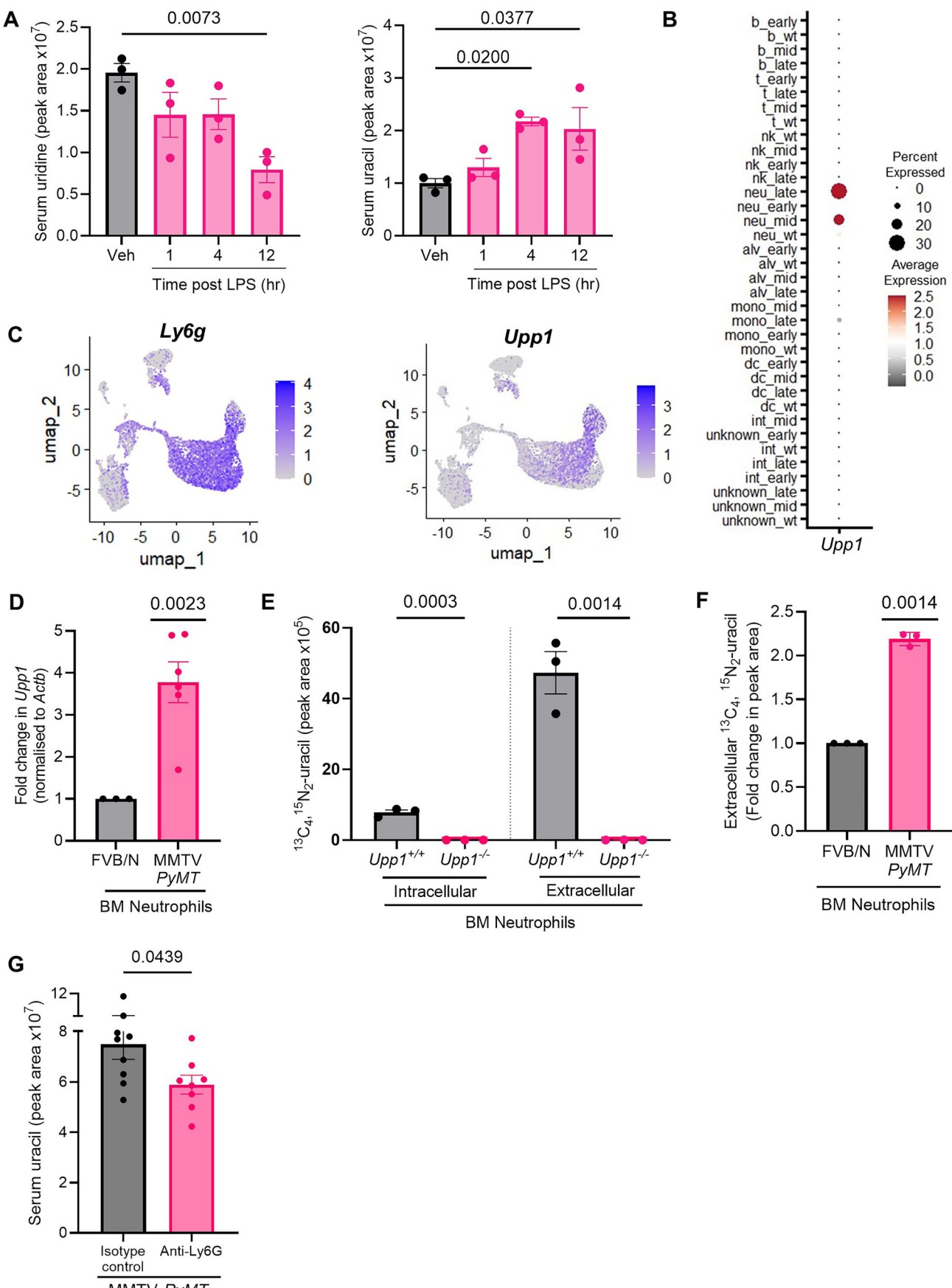

◄ **Figure 2. Neutrophils are a source of *Upp1*.**

(A) Wild-type (WT) mice on a C57BL/6 background were dosed intraperitoneally with 1 mg/kg lipopolysaccharide (LPS), and serum uridine and uracil measured by LC-MS at defined timepoints post dosing ($n = 3$ mice per experimental arm). (B) *Upp1* data were extracted from https://github.com/chris-mcginnis-ucsf/pymt_atlas, allowing *Upp1* to be assessed in immune cells from the lungs of WT or MMTV-*PyMT* tumour-bearing mice at varying timepoints in tumour progression (as assigned by early, mid or late). Cells were classified as b (B cells), t (T cells), nk (NK cells), neu (neutrophils), alv (alveolar macrophages), mono (monocytes), dc (dendritic cells), int (interstitial cells) and cells of unknown origin. Dot size represents the percentage of cells expressing *Upp1*, and dot colour represents average *Upp1* expression levels (Data source: McGinnis et al, 2024, $n = 29$ samples total). (C) UMAP plots of *Ly6g* (neutrophil marker) and *Upp1* in splenic cells isolated from MMTV-*PyMT* tumour-bearing mice, colour represents average *Upp1* expression levels (Data source: Alshetaiwi et al, 2020, $n = 3$ mice pooled). (D) *Upp1* was detected by qRT-PCR in neutrophils isolated from the bone marrow (BM) of FVB/N ($n = 3$ mice) or tumour-bearing MMTV-*PyMT* mice ($n = 6$ mice). (E) Extracellular $^{13}C_4,^{15}N_2$-uracil was detected by LC-MS in either cells, or media, from BM neutrophils incubated in media containing $^{13}C_9,^{15}N_2$-uridine for 24 h at 37 °C/5%CO$_2$ ($n = 3$ mice per experimental arm). (F) Extracellular $^{13}C_4,^{15}N_2$-uracil was detected as described for (E), using BM neutrophils isolated from FVB/N or tumour-bearing MMTV-*PyMT* mice ($n = 3$ mice per experimental arm). (G) Serum uracil assessed by LC-MS in tumour-bearing MMTV-*PyMT* mice treated with IgG control ($n = 9$ mice) or anti-Ly6G ($n = 8$ mice). Data Information: In (A, D–G), each dot represents an individual mouse, and data are presented as mean ± SEM. Statistical testing used to calculate $P$ values as follows: (A) one-way ANOVA; (D, F) one-sample $t$ test assesses the MMTV-*PyMT* difference to 1; (E, G) unpaired $t$ test. For (B, C), statistics were not performed. Source data are available online for this figure.

specific *Upp1* upregulation was unique to lung neutrophils, we also assessed *Upp1* in splenic cells isolated from MMTV-*PyMT* tumour-bearing mice (Data Ref: (Alshetaiwi et al, 2020)). Increased neutrophil *Upp1* expression was also a feature of splenic neutrophils from tumour-bearing mice (Fig. 2C; Appendix Fig. S4A), whilst *Upp1* was largely undetectable in splenic cells of other lineages (Appendix Fig. S4B–E). Notably, this pattern of expression was specific to *Upp1* and was not observed with the uridine phosphorylase homologue *Upp2* (Appendix Fig. S5A–C).

Neutrophils are produced in the bone marrow (BM) before being mobilised to other organs via the circulation. To understand the compartments in which LPS-driven *Upp1* upregulation was occurring, we quantified *Upp1* in flow cytometry-sorted neutrophils from the BM, blood and spleen. Consistent with our previous observations, LPS challenge potently increased *Upp1* expression in neutrophils isolated from all tissues assessed (Appendix Fig. S6A). Notably, LPS-driven *Upp1* expression had a tendency to be greater in Ly6G-high (mature) than in Ly6G-intermediate (immature) neutrophils, suggesting that *Upp1* expression might be upregulated during neutrophil maturation in addition to following neutrophil activation (Appendix Fig. S6A, left panel) (Data Ref: (Mackey et al, 2021)). Cytokines are key regulators of neutrophil production, and whilst granulocyte colony stimulating factor (G-CSF) has been identified as a key regulator of granulopoiesis, granulocyte–macrophage colony stimulating factor (GM-CSF) has been found to promote rapid release of mature neutrophils from the bone marrow in response to exogenous stimuli (Bayne et al, 2012; Fossati et al, 1998; Pylayeva-Gupta et al, 2012). Interestingly, we found increased levels of GM-CSF in the circulation of MMTV-*PyMT* tumour-bearing mice (Fig. EV2D). Furthermore, incubation of BM neutrophils with GM-CSF significantly increased their *Upp1* expression (Fig. EV2E), suggesting that tumour-induced alterations in cytokine expression can stimulate increased *Upp1* expression.

Given the pronounced neutrophilia associated with tumour progression in the MMTV-*PyMT* model of mammary cancer (Fig. EV2F), and the positive correlation between circulating neutrophil counts and metastatic burden in these mice (Fig. EV2G), we assessed *Upp1* expression in neutrophils from MMTV-*PyMT* mice bearing mammary tumours. Increased *Upp1* was detected in BM neutrophils from tumour-bearing MMTV-*PyMT* mice compared to non-tumour-bearing FVB/N controls (Fig. 2D). Furthermore, using Microenvironment Cell Populations-counter (MCP-counter) analysis to assess immune infiltrates in human tissue (Becht et al, 2016), the co-occurrence of increased *UPP1* expression

and a neutrophil signature was also reflected in transcriptomic data from human colorectal tumours, whilst no significant correlations were associated between *UPP1* expression and T-cell signatures (Appendix Fig. S6B).

To interrogate the functional/enzymatic activity of neutrophil-derived UPP1, we isolated neutrophils from *Upp1*$^{+/+}$ and *Upp1*$^{-/-}$ mice, incubated them with $^{13}C_9,^{15}N_2$-uridine, and used LC-MS to measure the $^{13}C_4,^{15}N_2$-uracil produced from phosphorolysis of the isotopomer-labelled nucleoside. This indicated that UPP1 in neutrophils catalyses the phosphorolysis of uridine (Fig. EV2H) and, moreover, that the uracil generated from this was exported into the medium (Fig. 2E). To assess the influence of a primary tumour on neutrophil UPP1 activity, BM neutrophils isolated from MMTV-*PyMT* tumour-bearing and non-tumour-bearing FVB/N mice were incubated with labelled uridine. This indicated that neutrophils from tumour-bearing mice had increased UPP1 activity, and consistent with the previous data, the increased uracil produced from UPP1-mediated cleavage of uridine was exported from the cell (Fig. 2F).

To determine the extent to which neutrophils are responsible for increased serum uracil in metastatic cancer, we depleted neutrophils in tumour-bearing MMTV-*PyMT* mice using an antibody depletion approach. Intraperitoneal administration of anti-Ly6G three times a week, for 2–3 weeks (Coffelt et al, 2015), was sufficient to deplete the number of neutrophils detected in the blood and lungs of tumour-bearing MMTV-*PyMT* mice (Fig. EV2I,J) and, importantly, significantly decreased levels of serum uracil (Fig. 2G).

Collectively, these data indicate that inflammatory triggers, and the presence of primary tumours, can not only drive the expansion of neutrophils but also increase *Upp1* expression within neutrophils, and that neutrophils represent a significant source of UPP1, and circulating uracil, in these pathologies.

## UPP1 influences neutrophil motility in the pre-metastatic lung through altered surface expression of α$_M$ integrin

Several studies have described the importance of neutrophils in the establishment of pre-metastatic niches, and how this contributes to metastasis of mammary cancer to the lung (Coffelt et al, 2015; Wculek and Malanchi, 2015). More recent work has also described a slow-moving neutrophil population in the pre-metastatic lung of mice bearing mammary tumours (Fercoq et al, 2024). Given the role of neutrophils in metastasis, and the upregulation of *Upp1* in

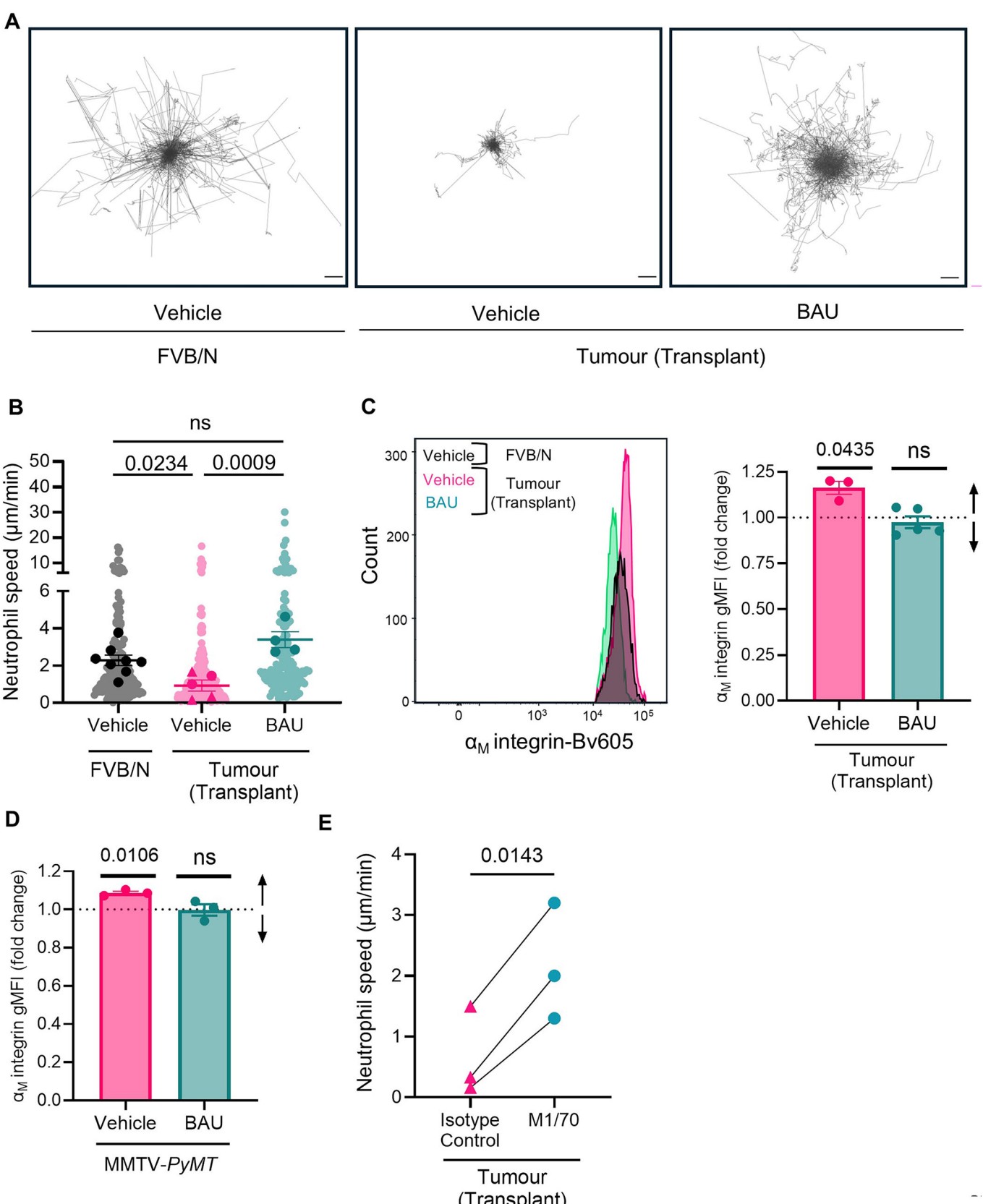

◄

**Figure 3. UPP1 influences neutrophil motility in the pre-metastatic lung through affecting surface expression of α_M integrin.**

(A) Centred neutrophil tracks obtained from confocal time-lapse microscopy of precision-cut lung slices (PCLS) from vehicle-treated FVB/N mice ($n = 8$ mice), vehicle-treated KP tumour-transplanted mice ($n = 5$ mice), and BAU treated KP tumour-transplanted mice ($n = 4$ mice), scale bar 7 μm. (B) Quantified motility of neutrophils from (A) ($n = 8$ vehicle-treated FVB/N mice; $n = 5$ vehicle-treated KP tumour-transplanted mice (3 of which, represented with triangles, were treated with isotype control antibody, and are presented in (E)), $n = 4$ BAU treated KP tumour-transplanted mice). (C) Representative histogram (left) of surface α_M integrin on lung neutrophils (right) measured by quantifying the geometric median fluorescence intensity (gMFI) by flow cytometry of neutrophils from the lungs of FVB/N mice (represented by the dotted line at 1, $n = 6$ mice), and FVB/N mice with tumours from transplant of KP tumour fragments ($n = 3$ vehicle-treated and $n = 5$ BAU treated tumour-bearing mice). (D) Surface α_M integrin measured as described in (C), for MMTV-*PyMT* mice treated with vehicle or BAU from palpable tumour ($n = 3$ mice per experimental group). (E) Fold change in average neutrophil speed, measured by live-cell imaging of PCLS, from tumour-bearing vehicle-treated mice described in (A) followed by ex vivo incubation with α_M integrin blocking antibody, M1/70, or isotype control ($n = 3$ mice per experimental group), matched PCLS are indicated by the adjoining line. Data Information: In (B), lighter dots represent individual neutrophils, darker dots represent mean neutrophil speed per mouse, horizontal line is mean of averages ± SEM. In (C, D), dots represent individual mice, bar graph is mean ± SEM. In (B) one-way ANOVA was performed on the mean speed per mouse, and in (C, D) data were analysed with a one-sample *t* test to assess whether each experimental group was different to 1, ns = not statistically significant. In (E), paired *t* test assesses differences between treatment with isotype control and M1/70. Source data are available online for this figure.

neutrophils from mouse models of metastatic cancer, we sought to understand whether blocking UPP1 activity could influence neutrophil behaviour in the pre-metastatic lung. Firstly, we assessed the ability of 5-benzylacyclouridine (BAU), a competitive inhibitor of uridine phosphorylase, to block activity of UPP1 in vivo. Oral administration of 30 mg/kg BAU, twice daily, suppressed uracil and elevated uridine levels, indicating effective inhibition of UPP1 in vivo (Appendix Fig. S7A). Next, we orthotopically transplanted mammary tumour fragments obtained from KP mice into the 4th mammary fat pad of FVB/N recipient mice. Upon detection of a palpable mammary tumour, BAU was administered to mice orally, twice daily, and this treatment regimen was maintained until primary tumours grew to 10–12 mm in diameter—a timepoint at which neutrophils are mobilised by the primary tumour, but overt lung metastases are not yet detectable (Fercoq et al, 2024). Because BAU did not influence primary tumour growth (Appendix Fig. S7B), this provided the opportunity to assess the ability of UPP1 to influence the pre-metastatic microenvironment of the lung. Live-cell imaging of precision-cut lung slices indicated that the migration speed of neutrophils was significantly decreased in tumour-bearing mice (Fig. 3A,B), consistent with a previous report (Fercoq et al, 2024). Strikingly, however, inhibition of UPP1 with BAU completely opposed the ability of the primary tumour to decrease neutrophil motility in the pre-metastatic lung (Fig. 3A,B). Of note, neither BAU treatment nor deletion of *Upp1* influenced neutrophil motility in the lungs of non-tumour-bearing mice (Appendix Fig. S7C,D). To understand how UPP1 might be influencing neutrophil motility we measured the expression of several cell surface molecules associated with neutrophil activation and adhesion (Appendix Table S2). This indicated that expression of the adhesion molecule α_M integrin (also known as CD11b) was consistently increased on the surface of neutrophils isolated from lungs of mice bearing mammary tumours, which were either orthotopically transplanted (KP tumour) or autochthonous (MMTV-*PyMT*), by comparison with the α_M integrin surface expression detected in appropriate non-tumour-bearing controls (Fig. 3C,D). Importantly, no increase in cell surface α_M integrin was detected in lung neutrophils from mammary tumour-bearing mice treated with BAU to inhibit UPP1 activity (Fig. 3C,D; Appendix Fig. S7E). To understand whether the UPP1-dependent increase in cell surface α_M integrin expression might be responsible for decreased neutrophil motility in the lungs of mammary tumour-bearing animals, we treated precision-cut lung slices with M1/70, an antibody that blocks the interaction of α_M integrin with its ECM ligands. Treatment of precision-cut lung slices, from mammary

tumour-bearing mice, with M1/70 restored motility to lung neutrophils (Fig. 3E). Collectively, these data suggest that mammary tumours can influence the surface expression of molecules such as α_M integrin in a UPP1-dependent manner, and the consequent cell surface availability of α_M integrin affects the migratory capacity of neutrophils in the pre-metastatic lung.

## UPP1-expressing neutrophils regulate the number of T cells in the lung

Neutrophils are capable of suppressing T-cell proliferation, and this can generate immunosuppressed microenvironments that favour metastasis (Aarts et al, 2019). Whilst no significant *Upp1*-dependent changes to immune landscapes were identified in organs such as the lymph nodes and spleen (Appendix Fig. S8A,B), given the accumulation of slow-moving neutrophils in the pre-metastatic lung (Fig. 3A,B) (Fercoq et al, 2024) we hypothesised that UPP1-driven alterations to the neutrophil phenotype might affect the landscape of other immune cell types within this metastatic target organ. Of all the cell populations assessed (Appendix Table S3), the most significant and consistent alterations following pharmacological inhibition or genetic deletion of *Upp1* were apparent in the numbers of T cells in the lung (Fig. 4A–C). Specifically, treatment of MMTV-*PyMT* tumour-bearing mice (but not non-tumour-bearing FVB/N mice) with BAU led to an increase in the number of CD4+ and CD8+ T cells in the lungs (Figs. 4A and EV3A,B). Consistently, increased numbers of CD8+ T cells were also observed in the lungs of tumour-bearing MMTV-*PyMT* mice in which *Upp1* had been deleted (Figs. 4B and EV3C), corroborating the ability of UPP1 to influence T-cell number in metastatic target organs. We also assessed T-cell effector functions and, whilst no statistically significant alterations in the number of granzyme B, IFN-γ or TNF-α-positive CD8+ T cells were detected (Fig. EV4A–C), significantly increased numbers of IL-2+ CD8+ T cells were observed in the lungs of tumour-bearing MMTV-*PyMT* mice lacking *Upp1* (Fig. 4C). IL-2 is a cytokine known to enhance T-cell proliferation (Gillis et al, 1978; Gillis and Smith, 1977; Morgan et al, 1976; Smith, 1988). To understand whether alterations in local levels of UPP1 substrate and product, uridine and uracil, respectively, could influence T-cell proliferation, T cells were isolated from FVB/N mice, induced to proliferate by incubation with CD3/CD28 positive beads, and the ability of uridine or uracil to enhance or suppress proliferation was determined. This indicated that neither 30, 60 or 120 μM uridine or uracil could alter the proliferation of T cells (Fig. EV4D). However, to study whether the physical presence of

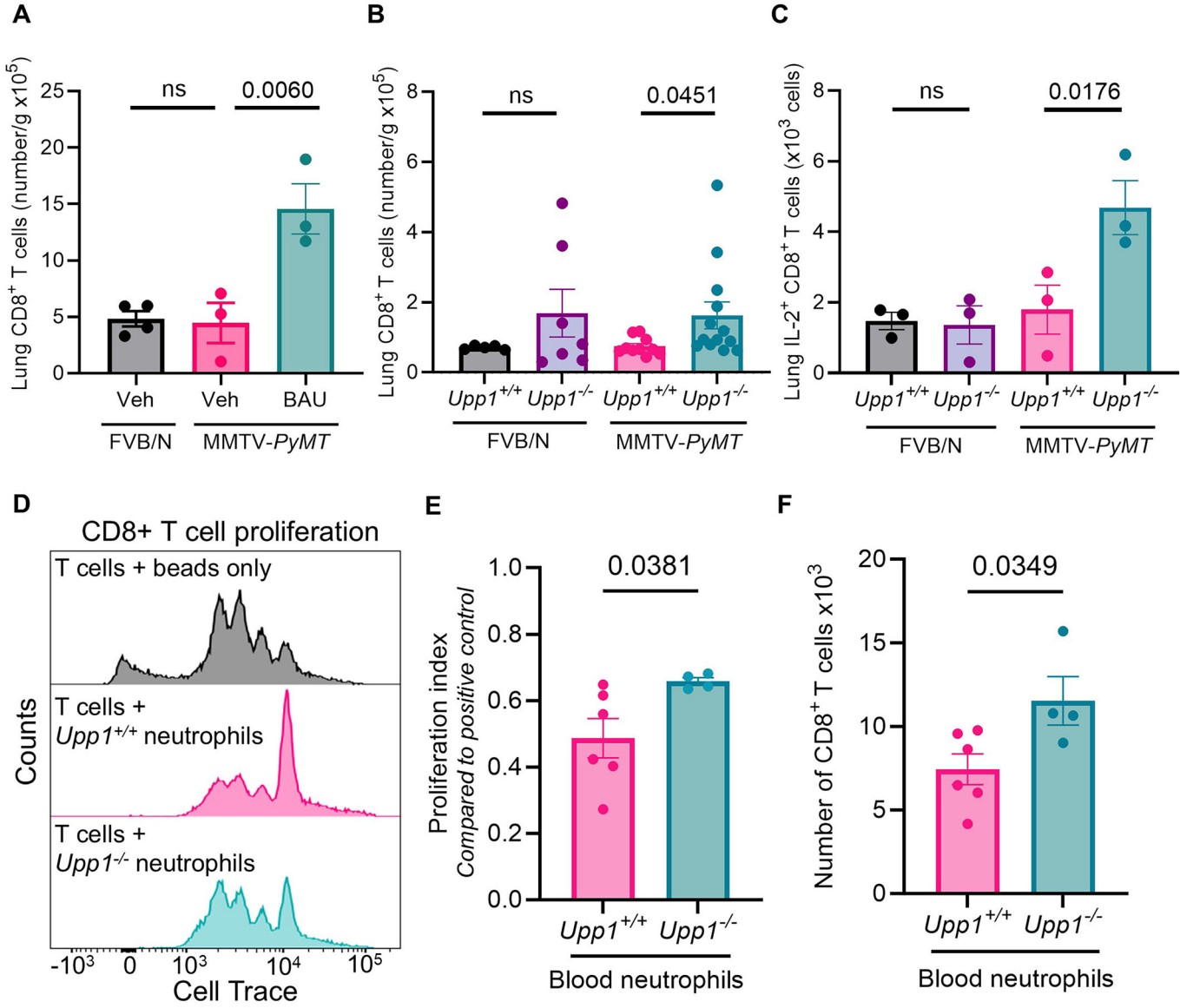

**Figure 4. UPP1 influences the immune landscape of the pre-metastatic lung.**

(A) MMTV-*PyMT* mice were treated with vehicle (*n* = 3 mice) or BAU (*n* = 3 mice) following detection of a palpable tumour, with FVB/N age-matched controls treated with vehicle for matched timepoints (*n* = 4 mice). Lungs of mice were harvested when one mammary tumour reached 10–12 mm in diameter, and the number of CD8+ T cells assessed by flow cytometry. (B) Lungs of MMTV-*PyMT*;*Upp1*+/+ (*n* = 10 mice) and MMTV-*PyMT*;*Upp1*−/− (*n* = 13 mice) mammary tumour-bearing mice were harvested when one tumour measured 10–15 mm diameter and the number of CD8+ T cells assessed by flow cytometry (*n* = 5 and *n* = 7 FVB/N *Upp1*+/+ and *Upp1*−/− mice as age-matched controls, respectively). (C) Cells were prepared as (B) but with a 3-h incubation with PMA and ionomycin together with Brefeldin A, to enable assessment of intracellular interleukin 2 (IL-2) levels (*n* = 3 mice per experimental group). (D) CD8+ T cells from the spleen and lymph nodes of FVB/N mice were incubated with equal numbers of neutrophils from the blood of tumour-bearing MMTV-*PyMT*;*Upp1*+/+ (*n* = 6 mice) and MMTV-*PyMT*;*Upp1*−/− mice (*n* = 4 mice) and stimulated to proliferate with CD3/CD28 dynabeads for 48 h. Representative histograms are shown. (E, F) Quantification of (D), as cell proliferation index (E) and total number (F) of CD8+ T cells as assessed by CellTrace staining and flow cytometry (neutrophil preparations from MMTV-*PyMT* mice;*Upp1*+/+ (*n* = 6 mice) and MMTV-*PyMT* mice;*Upp1*−/− (*n* = 4 mice)). Data Information: In (A–C, E, F), each dot represents an individual mouse, and data are mean ± SEM. In (A–C) statistical test used to determine *P* values is one-way ANOVA (using Kruskal–Wallis test in (C) due to non-normally distributed data, determined by Kolmogorov–Smirnov test), and in (E, F) unpaired *t* test, ns = not statistically significant. Source data are available online for this figure.

neutrophils might influence T-cell proliferation in a way that was dependent on their UPP1 status, we measured T-cell proliferation in the presence and absence of neutrophils from tumour-bearing MMTV-*PyMT* mice that were either *Upp1*+/+ or *Upp1*−/−. This indicated that *Upp1*-expressing neutrophils from tumour-bearing mice strongly suppressed T-cell proliferation, whilst neutrophils

isolated from *Upp1*−/− tumour-bearing mice had a reduced ability to suppress T-cell proliferation (Fig. 4D–F). Taken together, these data suggest that the increased numbers of IL-2+ CD8+ T cells in the lungs of mammary tumour-bearing *Upp1*−/− mice may be accounted for by the reduced ability of neutrophils from MMTV-*PyMT* *Upp1*−/− mice to suppress T-cell proliferation.

## UPP1 activity can influence ECM deposition in the lung

In addition to altered immune landscapes, the deposition of ECM proteins is also considered to be a key feature of pre-metastatic niche priming. Increased levels of ECM components are reported to occur in several organs—including the lung and liver—prior to metastasis (Zhang et al, 2016). Indeed, increased fibronectin deposition in the lung was recently highlighted as a contributor to metastasis in mouse models of mammary cancer (He et al, 2024). To understand whether UPP1 might influence the ECM-driven pre-metastatic niche priming, we used immunofluorescence to evaluate fibronectin deposition in the lungs of mice that were either wild-type or knockout for *Upp1*. This indicated that lung fibronectin levels were significantly decreased in MMTV-*PyMT* *Upp1*$^{-/-}$ tumour-bearing mice, but not by UPP1 knockout in non-tumour-bearing FVB/N animals (Figs. 5A,B and EV5A).

Although fibronectin may be deposited by several cell types, fibroblasts are generally accepted to be the principal depositor of ECM proteins, and altered fibroblast phenotypes are a key niche priming event in metastatic target organs (Dong et al, 2021). Assessment of UPP1 expression in fibroblasts found that although fibroblasts can express UPP1, levels are not upregulated in cancer-associated fibroblasts by comparison to their normal counterparts (Appendix Table S4) (Data Ref: (Jaeschke et al, 2020; Manouso-poulou et al, 2018; Torres et al, 2013)). Given our previous data showing that UPP1-expressing neutrophils from tumour-bearing mice export uracil into the extracellular microenvironment (Fig. 2F), and the consistent finding that increased uracil is detected in the circulation of mice and humans with metastatic cancer (Fig. 1B–F), we hypothesised that increased extracellular uracil generated by UPP1-expressing cells may influence ECM deposition by fibroblasts. To test this, we treated fibroblasts with uracil and measured the fibronectin content of the ECM deposited by these cells. Addition of 30 µM uracil (the concentration of uracil associated with high metastatic burden in MMTV-*PyMT* tumour-bearing mice (Fig. 1C)) enabled mouse fibroblasts to deposit ECM with increased fibronectin content in both 2D and 3D microenvironments, whereas 10 µM uracil (associated with little or no metastatic burden) was less effective in these regards (Figs. 5C and EV5B). Notably, these changes in fibronectin deposition were not accompanied by alterations in underlying fibrillar collagen deposition or organisation (Fig. EV5C,D). As fibronectin is a key contributor to the motility and invasiveness of many cell types including cancer cells, we allowed fibroblasts to deposit ECM in the presence and absence of extracellular uracil, de-cellularised the deposited ECM, and assessed the ability of this matrix to influence the migration of cancer cells (cells derived from *PyMT*$^{+}$ primary mammary tumours, named *PyMT*$^{+}$ cells). This indicated that the length of invasive protrusions extended by the cancer cells, a surrogate marker of their invasive capacity (Caswell et al, 2007), was increased when they were plated onto ECM that had been deposited in the presence of increased extracellular uracil (Fig. 5D). Furthermore, the ability of fibroblasts to deposit a UPP1-dependent pro-migratory ECM was driven only by exogenous addition of uracil, as endogenous expression of *Upp1* in fibroblasts did not influence ECM deposition by this cell type (Appendix Fig. S9A,B).

The $\alpha_5\beta_1$ integrin heterodimer is the cell's main fibronectin receptor, and the behaviour of this integrin is key to the deposition of fibronectin-containing ECM. The expression levels and ligand-binding affinity of $\alpha_5\beta_1$ are important for its ability to drive fibronectin polymerisation. However, neither of these indices of $\alpha_5\beta_1$ activation were influenced by addition of extracellular uracil to fibroblasts (Fig. EV5E–H). Integrin functions, and their ability to influence ECM deposition, are also controlled by the rates at which these ECM receptors are endocytosed and subsequently returned, or recycled, to the plasma membrane (Sundararaman and Mellor, 2021). We, therefore, measured the rate at which internalised $\alpha_5\beta_1$ was recycled from endosomes to the plasma membrane in the presence of increased extracellular uracil. This indicated that a concentration of uracil associated with high levels of metastasis (30 µM) significantly increased the recycling of $\alpha_5\beta_1$ and use of an antibody recognising the ligand-engaged conformation of $\beta_1$ integrin indicated that exogenous addition of uracil drove recycling of the $\alpha_5\beta_1$ heterodimer in its active conformation (Figs. 5E and EV5I). Moreover, uracil also increased recycling of other 'cycling' receptors, such as the transferrin receptor (Fig. EV5J) indicating that the ability of uracil to influence ECM deposition could be via general modulation of endosomal recycling and not specifically due to control of $\alpha_5\beta_1$ trafficking. And so, in addition to the influence of UPP1 status on neutrophil motility and CD8$^{+}$ T-cell infiltration, we find that the increased extracellular uracil produced by these neutrophils can also have non-cell autonomous effects in the local microenvironment, collectively contributing to the establishment of microenvironments that would be likely to promote metastasis.

## UPP1 promotes lung metastasis in a mouse model of mammary cancer

Our data indicate that UPP1 can influence neutrophil behaviour, T-cell number and ECM deposition in the lungs of tumour-bearing mice, suggesting that this enzyme can make key contributions to the progression of metastatic disease. The ability of UPP1 to influence the progression of mammary cancer was therefore assessed by comparing primary tumour development and incidence of lung metastasis in MMTV-*PyMT*:*Upp1*$^{+/+}$ mice to that in MMTV-*PyMT*:*Upp1*$^{-/-}$ mice. Whole-body genetic deletion of *Upp1* did not influence the time to clinical endpoint (one primary tumour reaching 15 mm in diameter) (Fig. 6A), nor did it influence the final primary tumour burden of MMTV-*PyMT* mice (Fig. EV6A; Appendix Fig. S10A,B). However, the proportion of MMTV-*PyMT* mice that had lung metastases at clinical endpoint was significantly reduced, from 85% in *Upp1*$^{+/+}$ mice to 52% in *Upp1*$^{-/-}$ mice (Figs. 6B and EV6B–D), indicating that the role of UPP1 in mammary cancer is specific to the metastatic process (Fig. 6C).

To determine if UPP1 influenced metastasis in another cancer type that we have found to display increased levels of *Upp1* and circulating uracil, we assessed the influence of *Upp1* knockout on disease progression in the KPC GEMM of pancreatic cancer. Overall survival of KPC mice was not different between *Upp1*$^{+/+}$ and *Upp1*$^{-/-}$ mice (Fig. EV6E), and whilst there was a tendency for deletion of *Upp1* to reduce metastasis to the liver, lung and diaphragm, these trends were not statistically significant (Fig. EV6F). However, local invasion of pancreatic tumours was significantly reduced in *Upp1*$^{-/-}$ mice (Fig. EV6G,H), suggesting that in pancreatic cancer, although *Upp1* does not contribute to distant metastasis in the same way as we report for mammary cancer, it may play a role in invasive behaviour at the primary tumour.

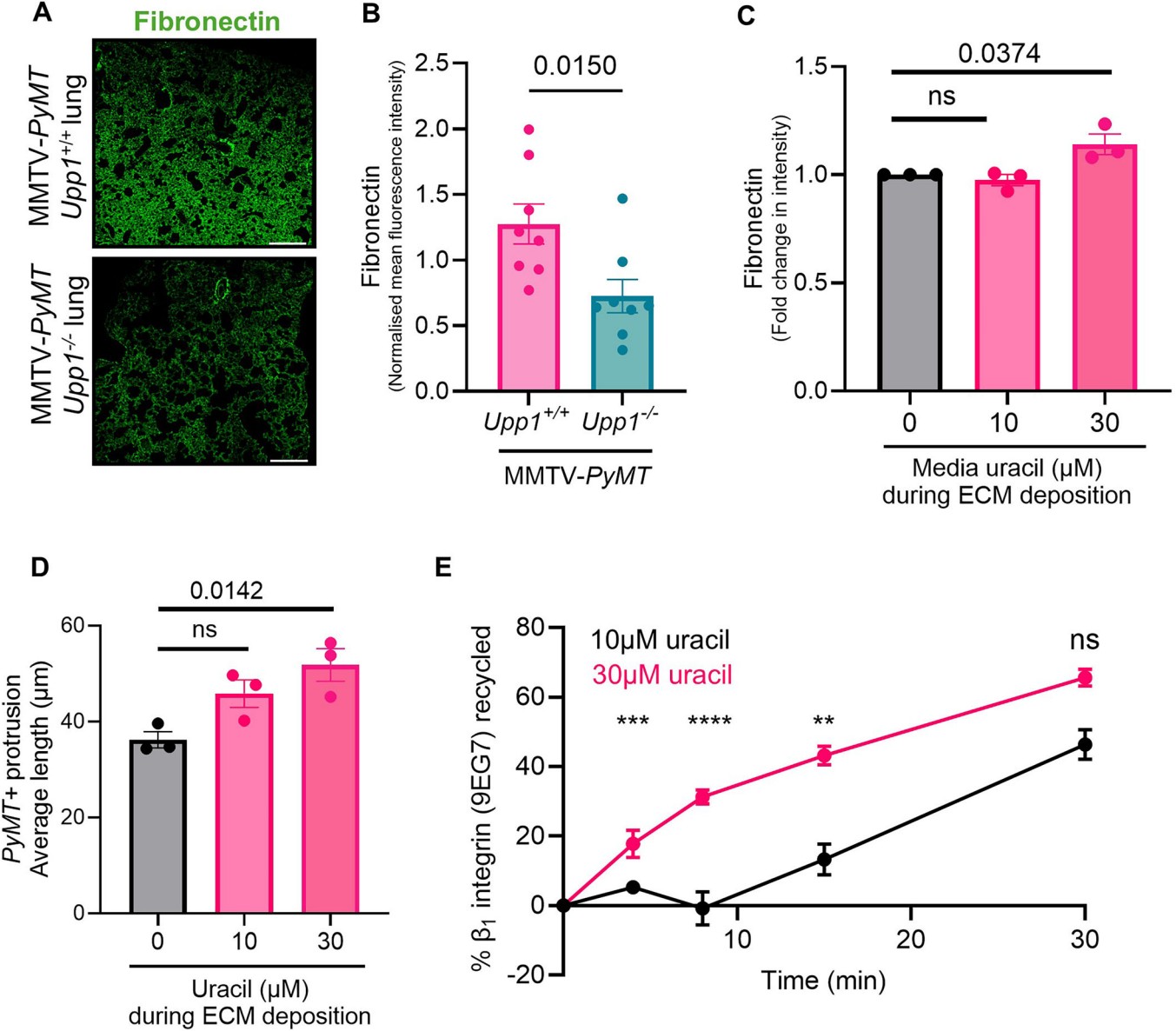

**Figure 5. Extracellular uracil increases fibronectin deposition by stromal cells to generate a pro-migratory extracellular matrix.**

(A) Fibronectin in the lungs of clinical endpoint MMTV-PyMT;Upp1+/+ and MMTV-PyMT;Upp1−/− mice assessed by immunofluorescence. Representative images shown (from a total of n = 8 mice per experimental group), scale bar 200 µm. (B) Quantification of lung fibronectin from (A), each dot represents an individual mouse (n = 8 mice per experimental group). (C) Fibronectin content of cellular-derived matrices made from fibroblasts treated with vehicle, 10 µM or 30 µM uracil quantified using immunofluorescence, data normalised to 0 µM uracil condition (n = 3 biological repeat matrix preparations using one fibroblast cell line). (D) Protrusion length of cells from MMTV-PyMT mammary tumours migrating on cellular-derived matrix determined using time-lapse microscopy. 10 cells were measured per field of view, for 6 positions per well. Each dot represents the mean protrusion length per experiment (n = 3 biological repeat matrix preparations). (E) The recycling of the active confirmation of the β1 integrin was assessed in fibroblasts treated with 10 µM or 30 µM uracil for 24 h (n = 3 biological repeats of the assay). Data Information: In (B), dots represent individual mice, in (C–E), dots represent biological replicates. In (B–E), data are presented as mean ± SEM. Statistical tests used to calculate P values are: (B) unpaired t test; (C, D) one-way ANOVA; (E) unpaired t tests for each timepoint with the following key: ***P = 0.0007, ****P = 0.0002, **P = 0.0045, ns = not statistically significant. Source data are available online for this figure.

## Discussion

Altered metabolism is a hallmark of cancer recognised to play an important role in the progression of metastatic disease. UPP1, the enzyme responsible for the phosphorolysis of uridine into uracil and ribose-1 phosphate, is a key nucleotide metabolism enzyme that has

been the subject of several papers in recent years, in part due to its consistent and strong relationship with decreased survival in several cancer types. The ability of cells to proliferate and migrate is dependent on the de novo generation of macromolecules, and targeting nucleic acid synthesis through nucleotide antimetabolite therapy represents the backbone of therapy for many cancers

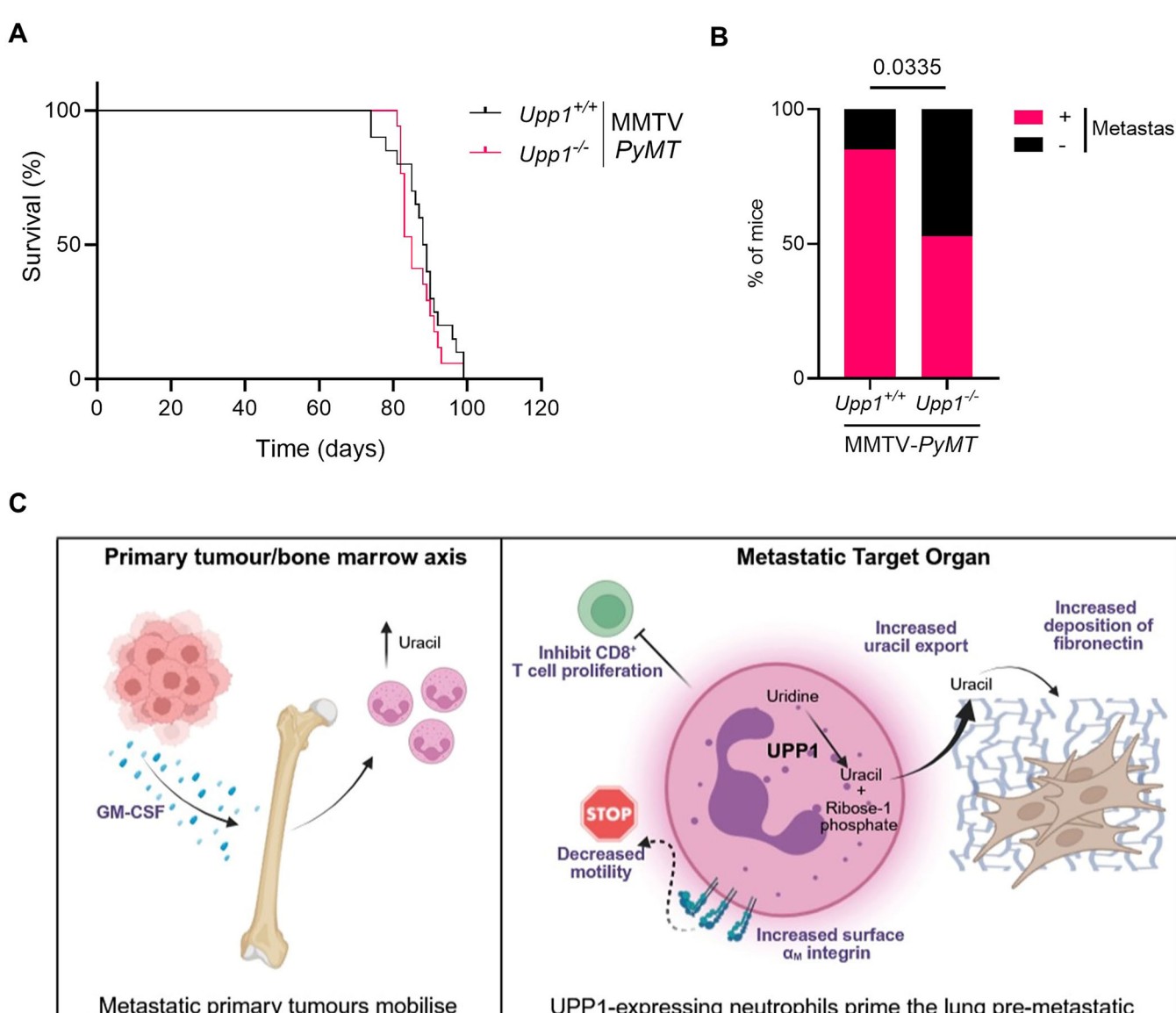

**Figure 6. UPP1 influences metastasis in a mouse model of mammary cancer.**

(A) Survival of MMTV-*PyMT;Upp1*[+/+] (n = 20 mice) and MMTV-*PyMT;Upp1*[−/−] (n = 17 mice), with mice culled at clinical endpoint corresponding to one mammary tumour reached 15 mm in diameter. (B) Lungs of mice from (A) were formalin-fixed and paraffin-embedded (FFPE) and metastatic burden assessed. The proportion of MMTV-*PyMT;Upp1*[+/+] (n = 20 mice) and MMTV-*PyMT;Upp1*[−/−] (n = 17 mice) mice with detectable lung metastases is presented. (C) Neutrophils increase expression of *Upp1* in response to the presence of a primary tumour. UPP1-high neutrophils secrete uracil into the extracellular microenvironment, and increased extracellular uracil promotes deposition of a fibronectin-rich ECM. UPP1-expressing neutrophils have high cell surface $\alpha_M$ integrin, resulting in decreased neutrophil motility in the pre-metastatic lung. Neutrophils accumulate and inhibit T-cell proliferation. Collectively these mechanisms create niches that support metastasis. Data Information: In (A) log-rank (Mantel–Cox) test shows that survival was not significantly different between experimental groups. In (B), statistics are Chi-squared test. Source data are available online for this figure.

(Tennant et al, 2010). While UPP1 catalyses a reversible reaction, the catabolic route which breaks down nucleosides is generally favoured. Paradoxically, expression levels of UPP1 are consistently upregulated in cancer—a disease canonically associated with increased formation of macromolecules rather than their disassembly. The upregulation of UPP1 in cancer cells has inspired studies focused on the tumour-intrinsic effects of UPP1. This includes studies in pancreatic cancer,

where UPP1-dependent production of ribose-1 phosphate was found to fuel central carbon metabolism in nutrient-deprived conditions (Nwosu et al, 2023; Skinner et al, 2023). Interestingly, we find that levels of pentose phosphates are not altered in *Upp1*[−/−] tumours, nor are pentose phosphates altered in the serum of neutrophil-depleted tumour-bearing mice (Appendix Fig. S11A,B). Tumour-specific *UPP1* increases have also been associated with disease progression in lung

adenocarcinoma (Li et al, 2024). In these instances, the authors used in vivo transplantation models to show that increased UPP1 expression could drive primary tumour growth. In the present study we have utilised GEMMs to study how UPP1 contributes to cancer progression in both mammary and pancreatic cancer. In the case of mammary cancer, UPP1 does not contribute to growth of primary mammary tumours, but is key to the establishment of lung metastases. In pancreatic cancer, deletion of *Upp1* in KPC mice also did not significantly influence primary tumour growth, but local invasion of PDAC was significantly decreased in the absence of *Upp1*.

Whilst LPS-driven systemic inflammatory responses are distinct from those evoked by the presence of a primary tumour, we demonstrate that a key hallmark of the metastasis-associated metabolome –elevated circulating uracil—is a common metabolic feature of both inflammation and metastasis. Importantly, the GEMMs employed in the present study all recapitulate metastasis through mechanisms associated with altered neutrophil biology (Casbon et al, 2015; Coffelt et al, 2015; Jackstadt et al, 2019; Steele et al, 2016; Wculek and Malanchi, 2015). We show here that neutrophils mobilised in response to the presence of a primary tumour generate and release uracil which makes a key contribution to the metastasis-associated circulating metabolome, and that this may influence ECM deposition by fibroblasts to help prime the lung metastatic niche. Importantly, UPP1 status also influences neutrophil behaviour in the lung and the ability of these cells to affect T-cell proliferation—mechanisms that can generate immunosuppressed environments that are permissive to metastatic seeding and outgrowth.

We have found that there are differences in the roles played by UPP1 in the progression of mammary and pancreatic cancer. Indeed, although circulating uracil is increased in a metastatic (KPC) model of pancreatic cancer prior to metastasis, *Upp1* knockout does not significantly influence distant metastasis (colonisation of the lung and liver) at clinical endpoint in this cancer type, but instead reduces local invasion of PDAC cells in the pancreas. Although neutrophils can contribute to metastasis of PDAC, neutrophil signalling at the tumour border is also associated with poor outcome in human pancreatic cancer (Steele et al, 2016), and so it will, therefore, be interesting to determine whether therapeutic inhibition of UPP1 early in tumour development may reduce progression of local invasive disease in pancreatic cancer.

In the context of mammary cancer, the ability of neutrophils to facilitate metastasis has been well described, and increased numbers of neutrophils in metastatic target organs result in immunosuppressed microenvironments that allow disseminated cancer cells to thrive (Casbon et al, 2015; Coffelt et al, 2015; Wculek and Malanchi, 2015). Recent work has also highlighted differences in the movement of neutrophils in the pre-metastatic lung, with lung neutrophils having decreased motility in mice bearing primary mammary tumours (Fercoq et al, 2024). The migration of leukocytes can be influenced by the cell surface expression of transmembrane receptors, including integrins. Indeed, integrins are known to influence cell adhesion and motility in a complex way. Very low levels of specific integrins at the cell surface are associated with decreased leukocyte migration due to a reduced ability of cells to engage the substratum (Yipp et al, 2017). However, high levels of cell surface integrin can also result in decreased motility due to excessive adhesion to the ECM (DiMilla et al, 1991; Lishko et al, 2003). Thus, cells struggle to migrate when surface levels of integrin are either very high or very low, whilst intermediate levels are associated with efficient cell migration. We have found that the presence of a primary tumour in the mammary gland increases neutrophil $\alpha_M$ integrin surface expression, which stalls neutrophil migration in the lung. Moreover, this primary tumour-driven stalling of neutrophil motility is relieved by blocking engagement of $\alpha_M$ integrin with its ECM ligands. Expression of cell surface proteins can commonly be regulated post-transcriptionally to enable efficient cellular responses to the local microenvironment, and post-transcriptional regulation of integrins is commonly exerted via modulation of endo- and exocytosis (Paul et al, 2015). As $\alpha_5\beta_1$ integrin recycling in fibroblasts is promoted by the addition of extracellular uracil (Figs. 5E and EV5I), this sets a precedent for UPP1's ability to regulate $\alpha_M$ integrin surface expression via modulation of endosomal trafficking. However, the addition of extracellular uracil to neutrophils does not increase $\alpha_M$ integrin surface expression (Appendix Fig. S12). Thus, although the data presented here suggest that UPP1 can, via release of uracil from neutrophils, influence integrin function in fibroblasts, this is unlikely to be how it controls integrin trafficking within the neutrophils themselves. It is interesting, therefore, to speculate how UPP1 might exert post-transcriptional control over neutrophil integrin trafficking and function. Indeed, integrin function is known to be impacted by metabolic perturbations that compromise glycosylation (Rossi et al, 2022), and $\alpha_M$ integrin is a glycoprotein, with N-linked mannose high glycan epitopes being a described feature of $\alpha_M$ integrin from human neutrophils (Kelm et al, 2020). Moreover, glycosylation of cell surface receptors decreases their rate of internalisation (Gollapudi et al, 2023), and pathway analysis finds that endocytosis is enriched in UPP1-high cells (Nwosu et al, 2023)—perhaps as a means to compensate for the decreased internalisation of surface proteins. Furthermore, UPP1 expression positively correlates with mannose metabolism in situations defined by high immune infiltration (Fan et al, 2022). UDP-N-acetylglucosamine (UDP-GlcNAc) is a key precursor for the synthesis of glycan oligosaccharides used for N-linked glycoprotein assembly and glycosylation of integrins, and UDP-GlcNAc synthesis requires UTP. Excessive consumption of uridine, due to increased UPP1 activity, may thus be a way in which UPP1 can impact the global glycosylation state of the cell. Thus, the ability of UPP1 to influence glycosylation states and how this, in turn, affects integrin trafficking and function will be an interesting hypothesis to investigate in the future.

In the absence of UPP1 expression/activity, we show that the number of T cells in the lungs of mice bearing mammary tumours increases. Furthermore, the expanded T-cell population in the lungs of mammary tumour-bearing *Upp1* knockout mice express IL-2, a cytokine that actively promotes T-cell proliferation (Ross and Cantrell, 2018). Indeed, whilst lung neutrophils from *Upp1*[+/+] tumour-bearing mice oppose T-cell proliferation, a factor that contributes to immunosuppressed microenvironments, neutrophils from tumour-bearing *Upp1*[−/−] mice have a decreased ability to suppress the proliferation of cytotoxic CD8[+] T cells. Suppression of anti-tumour leukocytes at metastatic target sites by myeloid cells may be mediated through several mechanisms, including via increased production of inducible nitric oxide synthase (iNOS) or reactive oxygen species (ROS) (Li et al, 2020a; Li et al, 2020b). Neutrophil extracellular traps (NETs) have also been shown to promote immunosuppressive microenvironments (Kaltenmeier et al, 2021). However, we find that expression of markers of

NETosis are not altered in the lungs of MMTV-*PyMT* *Upp1*[+/+] and *Upp1*[−/−] mice (Appendix Fig. S13). The ability of myeloid cells to reduce T-cell responses can also depend on other interactions with the local microenvironment. Indeed, fibronectin can polarise myeloid cells towards immunosuppressive states via direct engagement of the ITL3 receptor (Paavola et al, 2021), and chronic stress was recently shown to increase lung metastasis through a mechanism involving fibronectin accumulation, neutrophilia, and reduced T-cell numbers (He et al, 2024). Furthermore, UPP1 inhibition has previously been shown to protect against the generation of fibrotic microenvironments (Goncalves da Silva et al, 2021). Given our observations that fibronectin deposition in the lungs of tumour-bearing mice is reduced by *Upp1* knockout, and that the UPP1 product, uracil, promotes fibroblast-mediated fibronectin deposition, it will be interesting to determine the extent to which UPP1-dependent immune landscapes and ECM deposition combine to generate the immunosuppressed environment necessary to foster metastatic seeding in the lung. Indeed, it was our intention to address this using antibody-mediated approaches to deplete neutrophils, and whilst this reinforced that neutrophils are a significant source of circulating uracil (Fig. 2G), we have been unable to deplete neutrophils for long enough to significantly reduce metastasis without affecting the welfare of experimental animals (Appendix Fig. S14). This technical limitation has thus hindered our current ability to directly test the link between neutrophil UPP1, CD8[+] T cells, fibronectin and lung metastasis, and whilst this is a limitation of this current study it will interesting to pursue this line of enquiry using neutrophil-specific *Upp1* knockout mouse models in the future.

Collectively, our data show that neutrophils are a significant source of UPP1 in cancer, and that UPP1 can influence the immunosuppressive and motile behaviour of neutrophils and ECM deposition in the lung. Importantly, in the absence of UPP1 the ability of mammary tumours to metastasise is reduced, and we thus highlight a specific role for UPP1 in the metastatic process (Fig. 6C). The results of this study thus identify a clear role for UPP1 in neutrophil driven metastatic niche priming, providing exciting opportunities to investigate the clinical applications of these observations, including understanding whether circulating uracil could be used as a biomarker to detect metastatic disease, and testing the ability of UPP1 inhibitors to decrease the risk of distant metastasis, and/or improve responses to cancer immunotherapy.

# Methods

### Reagents and tools table

| Reagent/resource | Reference or source | Identifier or catalogue number |
| --- | --- | --- |
| **Experimental models** | | |
| *Upp1*[−/−] mouse (*M. musculus*) | This study | N/A |
| MMTV-*PyMT* (*M. musculus*) | CRUK Scotland Institute | N/A |
| KPN (*M. musculus*) | CRUK Scotland Institute | N/A |
| KPC & KP[fl]C (*M. musculus*) | CRUK Scotland Institute | N/A |

| Reagent/resource | Reference or source | Identifier or catalogue number |
| --- | --- | --- |
| FVB/N (*M. musculus*) | Charles River UK | Strain Code: 207 |
| C57BL/6J (*M. musculus*) | Charles River UK | Strain Code: 662 |
| CD11b-DTR (*M. musculus*) | University of California, Irvine | N/A |
| **Antibodies** | | |
| Zombie Green | Biolegend | 423111 |
| Ter119 FITC | Biolegend | 116206 |
| Nkp46 FITC | Biolegend | 137606 |
| CD115 (CSF1R) FITC | TONBO Bioscience | 35-1152-u100 |
| CD3 FITC | Biolegend | 100306 |
| CD19 FITC | Biolegend | 101506 |
| CD45 BV650 | Biolegend | 103151 |
| CD11b BV605 | Biolegend | 101257 |
| Ly6G Bv510 | Biolegend | 127633 |
| CD54 APC | Biolegend | 116120 |
| CD62L BUV395 | Biolegend | 740218 |
| CD101 PE-Cy7 | Life Tech (Invitrogen) | 25-1011-82 |
| CD117 Bv711 | Biolegend | 105835 |
| CXCR4 BV421 | Biolegend | 146511 |
| CXCR2 PE | Biolegend | 149304 |
| CD63 Percp-Cy5.5 | Biolegend | 143912 |
| Siglec-F APCCy7 | BD bioscience | 565527 |
| Zombie NIR | Biolegend | 423105 |
| CD45 AF700 | Biolegend | 103128 |
| CD19 Bv605 | Biolegend | 115540 |
| CD3 Percp-Cy5.5 | Biolegend | 100328 |
| CD8 FITC | Biolegend | 100706 |
| CD4 PeCy7 | Biolegend | 116016 |
| gdTCR PE | Biolegend | 118108 |
| Siglec-F AF647 | BD Pharmigen | 562680 |
| CD115 Bv421 | Biolegend | 135513 |
| CD11b Bv650 | Biolegend | 101259 |
| Ly6G Bv711 | Biolegend | 127463 |
| F4/80 Bv510 | Biolegend | 123135 |
| CD11c BV605 | Biolegend | 117334 |
| PD-1 APC | Biolegend | 135210 |
| CD62L PE-Dazzle | Biolegend | 104448 |
| CD44 BUV395 | BD Horizon | 568507 |
| NKp46 BV421 | Biolegend | 137611 |
| CD27 BV510 | Biolegend | 124229 |
| CD69 BV650 | Biolegend | 104541 |
| ICOS BV711 | Biolegend | 313548 |
| CD3 BV785 | Biolegend | 100355 |
| granzyme B AF647 | Biolegend | 515406 |

| Reagent/resource | Reference or source | Identifier or catalogue number |
| --- | --- | --- |
| IL-2 PE | Biolegend | 503808 |
| IFN gamma PE-Cy7 | Life Tech (Invitrogen) | 25-7311-82 |
| CD8 PerCP-Cy5.5 | Biolegend | 100734 |
| TNF alpha BV711 | Biolegend | 506349 |
| CD3 BV785 | Biolegend | 100355 |
| InVivoMAb rat IgG2a isotype control | 2B Scientific | BE0075-1-50MG |
| InVivoMAb rat anti-Mouse Ly6G, Clone—1A8 | 2B Scientific | BE0089-50MG |
| Ly6G-AF647 | Biolegend | 127610 |
| CD31-AF488 | Biolegend | 102414 |
| Fibronectin | Abcam | ab2413 |
| CD29 Clone 9EG7 | BD Pharmingen | 550531 |
| CD49e Clone 5H10-27 | BD Pharmingen | 553319 |
| Transferrin receptor | BD Pharmingen | 553264 |
| biotinylated goat anti-rabbit Ab | Vector Labs | BA-9400 |
| fibronectin | BD Pharmingen | 610078 |
| **Oligonucleotides** | | |
| qRT-PCR primer *Actb* F | Life Tech (Invitrogen) | Appendix Table S6 |
| qRT-PCR primer *Actb* R | Life Tech (Invitrogen) | Appendix Table S6 |
| qRT-PCR primer *Upp1* F | Life Tech (Invitrogen) | Appendix Table S6 |
| qRT-PCR primer *Upp1* R | Life Tech (Invitrogen) | Appendix Table S6 |
| **Recombinant DNA** | | |
| lentiCRISPR vector | Addgene | 52961 |
| **Chemicals, enzymes and other reagents** | | |
| Uracil | Merck | U0750-100G |
| Uridine | Merck | U3003;5G |
| $^{13}C_9, ^{15}N_2$-uridine | Cambridge Isotopes Laboratories | F004715 |
| 5-Benzylacyclouridine | CK Isotopes Limited | Custom |
| $^{13}C$-labelled yeast extract | Cambridge Isotope Laboratory | ISO1 |
| $^2H_9$ choline | Cambridge Isotope Laboratory | DLM-549 |
| $^{13}C_4$ 3-hydroxybutyrate | Cambridge Isotope Laboratory | CLM-3853 |
| $^{13}C_6 ^{15}N_2$ cystine | Cambridge Isotope Laboratory | CNLM4244 |
| $^{13}C_3$ lactate | Sigma-Aldrich | 485926 |
| $^{13}C_6$ glucose | Cambridge Isotope Laboratory | CLM-1396 |
| $^{13}C_3$ serine | Cambridge Isotope Laboratory | CLM-1574 |
| $^{13}C_2$ glycine | Cambridge Isotope Laboratory | CLM-1017 |

| Reagent/resource | Reference or source | Identifier or catalogue number |
| --- | --- | --- |
| $^{13}C_5$ hypoxanthine | Cambridge Isotope Laboratory | CLM8042 |
| $^{13}C_2 ^{15}N$ taurine | Cambridge Isotope Laboratory | CNLM-10253 |
| $^{13}C_3$ glycerol | Cambridge Isotope Laboratory | CLM-1510 |
| $^2H_3$ creatinine | Cambridge Isotope Laboratory | DLM-3653 |
| MojoSort Mouse Neutrophil Isolation Kit | Biolegend | 480058 |
| μ-slide 8 well high chamber slides | Thistle Scientific | 80806 |
| Agarose, low gelling temperature | Thermo Scientific | J66319.22 |
| DNase 1 | Roche | 03724778103 |
| Collagenase I | Worthington Biochemical | NC9482366 |
| Collagenase IV | Worthington Biochemical | NC9919937 |
| Foxp3 / Transcription Factor Staining Buffer Set | eBioscience | 00-5523-00 |
| MojoSort Mouse CD8 T-cell isolation kit | Biolegend | 480008 |
| CellTrace Yellow Proliferation kit | ThermoFisher Scientific | C34567 |
| Proteome Profiler Mouse XL Cytokine Array | Bio-techne | ARY028 |
| $H_2O_2$ | Sigma-Aldrich | 95329 |
| avidin block | Invitrogen | 004303 |
| biotin block | Invitrogen | 004303 |
| Protein blocking solution | Abcam/SpringBio | DFB125 |
| Avidin-HRP | Vectastain Elite | PK-7100 |
| Tyramide Signal Amplification (TSA-FITC) system | PerkinElmer | NEL749A001KT |
| BD Horizon™ brilliant stain buffer | BD Biosciences | 563794 |
| Polybrene | Sigma-Aldrich | H9268 |
| **Software** | | |
| Tracefinder | Thermo Scientific | Version 4.1 |
| Skyline | Adams et al 2020 | Version 21.2.0.369 |
| XCalibur | Thermo Scientific | Version 2.2 |
| Halo | Indica Labs | V3.6.4134.137 |
| Zen Black | Carl Zeiss | 2.3 SP1 FP3 |
| FlowJo | BD Biosciences | Version 10.9.0 |
| CFX Manager Software | BioRad | Version 4.0.2325.0418 |
| ImageJ | ImageJ | Version 1.49 |
| Harmony | PerkinElmer | Version 4.9 |
| Columbus | PerkinElmer | Version 2.8.0 |
| Imaris | Bitplane | V10.1.1 |

## Mice

The *Upp1* knockout mouse line was generated at the CRUK Scotland Institute by CRISPR gene editing technology (*Upp1* gene ENSMUSG00000020407/GRCm39:CM001004.3). Two CRISPR guides were identified which cut in the intronic sequence surrounding exon 4 of the *Upp1* gene and were demonstrated to have high cutting efficiency in vitro; GTAGGGCCCTTGA-CAATGTA, ATAAACGCAGGCCACTTGTA. Guide RNAs were generated in-house using Highscribe™ T7 Quick High Yield RNA synthesis kit (New England Biolabs Inc.), Alt-R™ S.p. Cas9 Nuclease was purchased from (Integrated DNA Technologies UK., Inc.). To generate the electroporation solution; the guide RNAs and spCas9 protein were combined in Optimem™ (ThermoFisher Life Technologies, Ltd). Approximately 7 h after in vitro fertilisation, the 1-cell stage embryos of 5–6-week-old FVB/N mice (Charles River UK) were introduced into electroporation solution, and electroporated using a NEPA21 electroporator (Nepa Gene Co., Ltd). The following day, 2-cell embryos were transferred into the oviducts of pseudopregnant CD1 females (Charles River UK). Genotyping of subsequent pups was performed from ear samples by PCR using the primers F: GCAGAGTGGGCTAGCAACTA and R: CAGTTG-CAAGCAGGGATCAT followed by Sanger sequencing.

Female mice were used in all experimental studies using the mammary cancer model. To model mammary cancer MMTV-*PyMT* mice (Guy et al, 1992) were backcrossed >20 generations and maintained on an FVB/N background. MMTV-*PyMT* mice were always compared with their FVB/N littermate controls. *K14*-Cre; *Trp53*$^{fl/fl}$ (KP) mice were gifted from Jos Jonkers (Liu et al, 2007). These mice were backcrossed to an FVB/N background for 5 generations. For orthotopic transplantation models of mammary cancer, 1-mm³ mammary tumour fragments obtained from KP mice were transplanted into the 4th mammary fat pad of 10–12-week-old female recipient FVB/N mice (Charles River UK) using appropriate anaesthesia and analgesia as previously described (Millar et al, 2020). Mice were monitored for tumour development 2–3 times per week. Once detected, tumour growth was monitored by calliper measurement three times per week. For all mammary models, endpoints (time or size) are described in the appropriate figure legends. In the MMTV-*PyMT* model, mammary tumours arise spontaneously in any of the five pairs of mammary glands, and thus multiple mammary tumours may be present. The clinical endpoint is reached when one tumour reaches 15 mm in diameter.

For colon cancer, KPN mice (*villin*Cre$^{ER}$; *Kras*$^{G12D/+}$; *Trp53*$^{fl/fl}$; *R26*$^{N1icd/+}$) on a C57BL/6 background described previously (Jackstadt et al, 2019) were induced with a single injection of 2 mg tamoxifen by intraperitoneal injection at an age of 6–12 weeks. Both sexes were used. Serum was isolated from tail bleeds at 30, 60 and 90 days post induction to provide baseline levels of metabolites. Clinical endpoint was defined as weight loss and/or hunching and/or poor body conditioning and confirmed metastatic cancer upon dissection.

For pancreatic cancer, KPC (*Pdx1*-Cre; LSL-*Kras*$^{G12D/+}$; LSL-*Trp53*$^{R172H/+}$) mice were used to model metastatic pancreatic cancer, and KP$^{fl}$C (*Pdx1*-Cre; LSL-*Kras*$^{G12D/+}$; LSL-*Trp53*$^{fl/+}$) were used to model pancreatic cancer that has a decreased propensity to metastasise (Morton et al, 2010). These mice are maintained on a mixed strain background, and both sexes were used. For Fig. 1E, mice were sampled when pancreatic tumours were detectable by palpation, a timepoint in which pro-metastatic structural changes are apparent in distant target organs but that precede the formation of overt metastases (Novo et al, 2018). Average KP$^{fl}$C age at death was 19.30 weeks, and KPC 17.30 weeks: Difference in means $-1.705 \pm 2.778$, 95% confidence interval $-7.663$ to $4.253$, $P$ value 0.5492 (unpaired $t$ test). Only mice with confirmed pancreatic tumours upon dissection were included for downstream analysis. KPC mice used to evaluate survival or metastatic burden (Fig. EV6E–H) were culled when exhibiting defined endpoint symptoms of PDAC, and censored when showing model-specific signs not associated with PDAC (e.g. lymphoma, lung adenoma or papillomas according to limitations of the license). Local invasion was scored at necropsy and involved attachment and invasion of the primary PDAC to organs within the abdominal cavity, for instance, to the liver, kidney or intestine. Inconclusive annotation refers to samples that were unclear at necropsy as to whether full invasion had occurred. Evaluation of metastasis was only performed in animals that reached humane clinical endpoint due to pancreatic cancer and are described further in 'histological assessment of fixed murine tissue'. Any further exclusions were due to either sampling by personnel not trained in macroscopically scoring of metastases in this model, or the unavailability of tissue for microscopic evaluation.

For experiments utilising the CD11b-DTR mice, mice were produced as an F1 cross between B6.FVB-Tg(*ITGAM*-DTR/EGFP) 34Lan/J mice and C57BL/6J mice (Jackson Laboratory). Male mice between 8–12 weeks of age were used experimentally. Orthotopic PDAC tumours were established from $5 \times 10^4$ KPC-7940B PDAC cells injected into the pancreas in 50 μL of 1:1 Matrigel-DMEM + 10%FBS (Corning) of F1 C57BL/6J CD11b-DTR mice and allowed to establish tumours for 14 days. Mice were then treated with diphtheria toxin (Enzo Life Science) at a concentration of 25 ng/kg or vehicle by intraperitoneal injection. Mice were humanely euthanised 24 h after treatment.

In experiments where the immune system was mobilised in the absence of cancer, mice were dosed intraperitoneally with 1 mg/kg (C57BL/6) or 0.5 mg/kg (FVB/N) lipopolysaccharide (LPS) for 4–24 h, as defined in individual figure legends. Both sexes were used at an age of 8–12 weeks.

As per ARRIVE (Animal Research: Reporting of In Vivo Experiments) guidelines 2.0: In all instances, the experimental groups of comparison are described in the figure legend. Sexes of mice used per model are described above, and experimental endpoints (time or size) are described in the appropriate figure legends. Experimental units are a single animal, and the number of mice is represented as dots within graphs and reported in the corresponding figure legend. Exclusion criteria in cancer models applied to mice that did not have cancer upon dissection (for example, mass due to cyst instead of tumour, or those that were culled for welfare reasons). Any other exclusion criteria for specific methods are described in the appropriate methods section. Sample sizes are described in the appropriate figure legends and influenced by the power needed to understand the biological effect being measured. Mice were randomly allocated to treatment arms. Confounding factors such as order of treatment and measurements were minimised by delegation of these tasks to staff blinded to the experimental design. Samples were labelled during processing and analysis in a way that did not indicate the treatment group to the researcher. Unblinding was performed once the processing and

analysis of samples were complete. Outcome measures, statistical methods and experimental animals are all described within the appropriate legends, procedures are described in the methods section, and results in the main body of text.

Except for CD11b-DTR mice, all mice were housed at the CRUK Scotland Institute, with environmental proactive enrichment, and in a ventilated barrier facility (12-h light/dark cycle). Procedures were performed in accordance with the UK Animal (Scientific Procedures) Act 1986, approved by the local animal welfare (AWERB) committee, and conducted under UK Home Office licences 70/8645, 60/4181, 70/8646, P72BA642F, and PP6345023. Genotypes of genetically engineered mouse models were confirmed by Transnetyx genotyping services. Experiments using CD11b-DTR mice were conducted at the University of California, Irvine, approved by the Institutional Animal Care and Use Committee (IACUC) under protocol AUP-23-084.

## Neutrophil depletion in MMTV-*PyMT* mice

MMTV-*PyMT*[+] tumour-bearing mice were treated with either InVivoMAb rat IgG2a isotype control or InVivoMAb rat anti-Mouse Ly6G, Clone—1A8 (2B Scientific) once one tumour had reached 5 mm in diameter, using a dosing regimen of one 400 µg loading dose followed by 100 µg maintenance doses, with dosing via the intraperitoneal route on a Monday–Wednesday–Friday dosing schedule. Mice were culled once one primary tumour reached 15 mm in diameter. Circulating neutrophil counts were assessed in blood samples by IDEXX (IDEXX BioAnalytics). One mouse was censored in the isotype control experimental arm during data analysis due to abnormally low neutrophil counts, whilst two mice needed to be censored from the anti-Ly6G experimental arm as neutrophil depletion was found to be ineffective in these animals.

## Plasma from metastatic breast cancer patients and healthy volunteers

Plasma samples from patients with metastatic breast cancer and female healthy volunteers were originally collected under a protocol approved by West of Scotland REC4 (reference 10/S0704/32). Analysis of these archival samples for the current research was approved by the Office for Research Ethics Committees Northern Ireland HSC REC B (reference 17/NI/0228).

## Untargeted metabolomics of MMTV-*PyMT* serum

Polar metabolites were extracted as previously described in (Villar et al, 2023), and this methodology is thus shared under the Creative Commons Attribution 4.0 International (CC BY 4.0) license (https://creativecommons.org/licenses/by/4.0/). Briefly, polar metabolites were extracted from the serum by the addition of ice-cold extraction solution (1:50 dilution, 20% water, 50% methanol and 30% acetonitrile) followed by centrifugation at $16,000 \times g$ for 10 min at 4 °C. The metabolite extract (supernatant) was stored at $-74$ °C until LC–MS analysis. Metabolite extracts (5 µL) were injected and separated using a ZIC-pHILIC guard-analytical column (SeQuant; 20 mm × 2.1 mm; SeQuant; 150 mm × 2.1 mm, 5 µm; Merck) on an Ultimate 3000 HPLC system (ThermoFisher Scientific). Chromatographic separation was performed using a 30-min linear gradient starting with 20% ammonium carbonate (20 mM, pH 9.2) and 80%

acetonitrile, terminating at 20% acetonitrile at a constant flow rate of 100 µL min⁻¹. The column temperature was held at 45 °C. An online Q Exactive Orbitrap mass spectrometer (ThermoFisher Scientific) equipped with electrospray ionisation was used for both metabolite profiling and metabolite identification (LC–MS). For profiling, the polarity switching mode was used with a resolution (RES) of 70,000 at 200 *m/z* to enable both positive and negative ions to be detected across the mass range (75 to 1000 *m/z*, with automatic gain control (AGC) target of $1 \times 10^6$ and maximal injection time (IT) of 250 ms). Data-dependent fragmentation was performed to aid metabolite identification using a pooled sample comprised of a mixture of all extracts. The Q Exactive was operated in positive and negative polarity mode separately (35,000 RES, AGC target of $1 \times 10^6$ and max IT of 100 ms), and the 10 most abundant ions were chosen for fragmentation (minimum AGC target of $1 \times 10^3$, AGC target of $1 \times 10^5$, max IT of 100 ms, 17,500 RES, stepped normalised collision energy of 25, 60 and 95, isolation width of 1 *m/z*, dynamic exclusion of 15 s and charge exclusion of >2) per survey scan. The pooled sample was also used as a quality control (QC) and was repeatedly measured throughout the acquisition of the randomised biological extracts. Untargeted metabolomics analysis was performed using Compound Discoverer software (Thermo Scientific v2.1). Retention times were aligned across all data files (maximum shift of 2 min and mass tolerance of 5ppm). Unknown compound detection (minimum peak intensity of $1 \times 10^6$) and grouping of compounds were performed (mass tolerance of 5ppm and retention time tolerance of 0.2 min). Missing values were filled using the software's 'Fill Gap' node (mass tolerance of 5ppm and signal/noise tolerance of 1.5). Data were corrected for batch effects using the QC replicate injection data and a QC-based area correction regression model (Linear, 50% QC coverage, 30% RSD). Compound identification was assigned by matching the mass and retention time of observed peaks to an in-house library generated using metabolite standards (mass tolerance of 5ppm and retention time tolerance of 0.5 min) or by matching fragmentation spectra to mzCloud (www.mzcloud.org; precursor and fragment mass tolerance of 10ppm and match factor threshold of 50).

## Uracil quantification in MMTV-*PyMT* serum

Uracil was quantified in serum by the standard addition method. Briefly, a uracil stock solution was used to spike serum (pool of experimental samples) during extraction, generating a six-point calibration curve ranging from 5 µM to 160 µM (0.1 µM to 3.2 µM at 1:50 extraction ratio). The calibration curves and samples were analysed together with the LC–MS profiling method described above. Targeted quantification was performed using Tracefinder (4.1, Thermo Scientific), a linear calibration curve was constructed by plotting the peak area of uracil against the standard addition amount. This was then used to calculate the uracil concentration in the unspiked samples.

## Targeted metabolomics

Targeted metabolomics was performed using the same instrument setup as described above for the untargeted analysis with modifications to the chromatography. The linear gradient was shortened to 15 min, and the flow was increased to 200 µl min⁻¹. The mass spectrometry detection was carried out with the same parameters as above over the shortened LC runtime. Polar

metabolites were extracted from serum by the addition of ice-cold extraction solution (1:50 dilution, 20% water, 50% methanol and 30% acetonitrile) followed by centrifugation at $16,000 \times g$ for 10 min at 4 °C. In cases where metabolites were extracted from tissue, 25 mg of tissue was used per mL of extraction solvent, samples were homogenised at 4 °C (Precelly's 24 homogeniser with homogenising tubes for hard tissue) followed by centrifugation at $16,000 \times g$ for 10 min at 4 °C. The metabolite extract (supernatant) was stored at −74 °C until LC–MS analysis.

Targeted metabolomics analysis was performed using Skyline (Adams et al, 2020), version 21.2.0.369. A transition list containing the compound names, preferred adduct/polarity and measured retention times (with 2-min retention time tolerance) was generated using reference standards on the same LC–MS method. Extracted ion chromatograms were generated for each compound, and the corresponding chromatographic peaks were carefully inspected and integrated. Relative quantification between sample groups was performed.

## $^{13}C_9,^{15}N_2$-uridine metabolic tracing

Neutrophils were isolated from the femurs of MMTV-*PyMT* tumour-bearing mice using MojoSort Mouse Neutrophil Isolation Kit (Biolegend), with neutrophil retention via negative selection strategy. Neutrophils were plated at a density of 450,000 cells/well in a six-well plate using Plasmax (Vande Voorde et al, 2019) as a physiologically relevant cell culture medium, including 30 µM $^{13}C_9,^{15}N_2$-uridine (Cambridge Isotope Laboratories Inc), or unlabelled equivalent (Sigma), for 24 h at 37 °C/5% $CO_2$. Cells where then removed from the plates by manual pipetting and centrifuged at $400 \times g$ 4 °C for 5 min to isolate the cell and media fractions. Polar metabolites were extracted using ice-cold extraction solution (20% water, 50% methanol and 30% acetonitrile). For intracellular extractions, 100 µL was used per $1.50 \times 10^5$ cells. For extracellular extraction, a 1:50 dilution of media was used in the extraction solution. Solutions were vortexed, followed by centrifugation at $16,000 \times g$ for 10 min at 4 °C. The metabolite extracts (supernatant) were then stored at −74 °C until LC–MS analysis. Targeted metabolomics was performed as described above. As neutrophils are semi-adherent, the exact number of suspension neutrophils harvested may have differed to the number plated. For this reason, intracellular values were normalised to the cell count recorded per individual sample prior to lysis in extraction solution (implemented as a relative fold change between samples to preserve the peak area).

## Serum metabolomics of the CD11b-DTR mouse model

Quantification of metabolites in serum from CD11b-DTR animals was performed as previously described (Sullivan et al, 2019a; Sullivan et al, 2019b). Briefly, 7 libraries containing 149 metabolites were serially diluted in water from 5 mM to 1 µM concentration. These external standard pools were used to calibrate isotopically labelled internal standards and allow for quantification of metabolites where internal standards are not available (Appendix Table S5). Metabolites were then extracted from 5 µL of external standard pools or biofluid samples using 45 µL of 75:25:0.1 acetonitrile:methanol:formic acid extraction mix to which the isotopically labelled internal standards described in Appendix Table S5 were added. After the addition of the extraction mix,

samples were vortexed for 10 min at 4 °C and then centrifuged at 15,000 rpm for 10 min at 4 °C to pellet insoluble material. The supernatant containing polar metabolites was then moved to sample vials for analysis by LC–MS as previously described (Sullivan et al, 2019a; Sullivan et al, 2019b).

XCalibur 2.2 software (ThermoFisher Scientific) was used for metabolite identification and quantification. Metabolite identification was performed using the external standard libraries as a reference to confirm the m/z and retention time for each metabolite. For quantification, when isotopically labelled internal standards were available, the concentration of internal standard in the extraction mixture was first determined by referencing the peak area of the isotopically labelled metabolite to peak areas of the unlabelled metabolite in the external standard dilution series. Upon determining the internal standard concentrations in the extraction mix, the peak areas of unlabelled metabolites in the biofluid samples were compared to the peak area of the quantified isotopically labelled internal standard to determine the concentration of the metabolite in the biofluid sample. For quantification of metabolites when were isotopically labelled internal standards were not available, the peak area of the metabolite was normalised in each sample to the peak area of an isotopically labelled internal standard with similar retention time. The normalised peak area of the metabolite in the biofluid sample was then compared to the normalised peak areas of the metabolite in the external standard dilution series to interpolate the concentration of the metabolite in the biofluid sample.

## Histological assessment of fixed murine tissue

Formalin-fixed paraffin-embedded (FFPE) lungs and liver were cut into a series of 4-µm serial sections and metastatic burden assessed by manual assessment of H&E stain in sections 5, 10 and 15 (using HALO image analysis software v3.6.4134.137). For each mouse, the average number of metastases identified across the three sections of the lung was recorded, in addition to the average metastatic burden across the three lung sections (defined as (area of lung covered by metastases/total lung area) × 1000). For assessment of microscopic diaphragm metastases, single H&E-stained slides of the diaphragm were scanned (Aperio AT2 slide scanner), visualised using Aperio software (ImageScope v12.4.3.5008) and the presence/absence of metastasis was recorded.

## Live precision-cut lung slice imaging

Live-cell imaging of precision-cut lung slices was performed as described previously (Fercoq et al, 2024). Briefly, lungs were inflated with low-melting point agarose (Sigma), placed into ice-cold PBS, and sliced into 300-µm-thick sections using a vibratome. Lung slices were stained directly in chamber wells by direct incubation with Ly6G-AF647(1:200, Biolegend), CD31-AF488 (1:200, Biolegend) (diluted in phenol-red free DMEM with 5% FBS) and imaged on a Zeiss LSM880 confocal microscope at 37 °C/5%$CO_2$ for 20–40 min with z-stacks of ~20 µm. Acquisition was performed with a 32-channel Gallium arsenide phosphide (GaAsP) spectral detector plan Apo objective lenses (20×/0,8 NA - dry, Carl Zeiss). Samples were excited simultaneously with 405, 488, 561 and 633 laser lines, with signal collected onto a linear array of the 32 GaAsp detectors in lambda mode with a resolution of 8.9 nm over

the visible spectrum. Spectral images were then unmixed with Zen software (Carl Zeiss) using reference spectra acquired from unstained tissues (tissue autofluorescence) or slides containing single fluorophores. Time-lapse image analysis was performed using Imaris (Bitplane). Neutrophils were segmented and tracked using the surface tool on Ly6G fluorescently labelled neutrophils, with surface detection and tracks assessed manually to ensure accuracy. In instances where the influence of the M1/70 blocking antibody on cell motility was assessed, 10 µg M1/70 or isotype control (ThermoFisher) was added into the medium of the chamber-well.

## Flow cytometry

Lungs were mechanically dissected using dissecting scissors and shaken in RPMI medium (ThermoFisher) supplemented with 5% FBS, at 37 °C for 20 min. Lung suspensions were filtered through a 70-µm cell strainer, rinsed with RPMI medium, centrifuged at $400 \times g$ 4 °C for 5 min, and resuspended in RPMI medium supplemented with 5% FBS and 2 mM EDTA (ThermoFisher). To prepare cells from the spleen, spleens were pushed through a 70-µm filter and red blood cells lysed using 2 mL red blood cell lysis buffer (ddH$_2$0, 5% FBS (v/v), 20 mM EDTA, 0.001 g/mL potassium bicarbonate, 0.008 g/mL ammonium chloride) for 3 min. Reaction was quenched with 5 mL RPMI/5% FBS/2 mM EDTA. Cells were then centrifuged for at $400 \times g$ 4 °C for 5 min, resuspended in RPMI/5% FBS/2 mM EDTA and filtered for a final time through a 70 µm filter. To prepare cells from lymph nodes (LN), auxillary LN were collected in PBS, punctured and split open using forceps, and placed in 500 µL of RPMI containing 200 µg/mL DNase 1 (Roche), 100 U/L collagenase I (Worthington Biochemical), and 500 U/L collagenase IV (Worthington Biochemical); and incubated for 20 min at 37 °C. Samples were triturated with a 1-mL pipette and incubated for a further 20 min at 37 °C. Samples were passed through a 70-µm filter into a 1.5-mL Eppendorf tube and centrifuged at $300 \times g$ 4 °C for 5 min. Cells were transferred to a 96-well V-bottomed plate for staining. Briefly, cells were pelleted, washed in PBS, and resuspended in live/dead stain at room temperature protected from the light for 20 min. Cells were then washed in PBS, pelleted, resuspended in Fc Block and incubated at room temperature for 15 min. Cells were then stained with the relevant antibody cocktail in BD Horizon™ brilliant stain buffer (BD Biosciences) (room temperature for 15 min in the dark), followed by sequential washing and fixation in PFA. Neutrophils were defined as Dead/Dump negative>CD45$^+$>Ly6G$^+$CD11b$^+$, using the antibody panel described for neutrophil characterisation (Appendix Table S2). Total cell characterisation was performed using the antibody panel described in Appendix Table S3. T cells were further characterised by a lymphoid antibody panel (Appendix Table S6). Fluorescence minus one (FMO) controls were prepared to set gates. Count beads were added to samples to enable the accurate determination of cell numbers. Data were acquired on the BD Fortessa (BD Biosciences) and analysed using FlowJo (BD Biosciences v10.9.0).

## Intracellular T-cell staining

Lungs were mechanically dissociated with dissecting scissors and transferred into DMEM medium (ThermoFisher) supplemented with 1 mg/mL collagenase D (Roche) and 25 µg/mL DNase 1 (ThermoFisher). Enzymatic dissociation was carried out using the

gentleMACS Octo Dissociator, run: 37C_m_LDK_01 (Miltenyi Biotec), as described (Curio and Belz, 2022). Consequent lung suspensions were then filtered through a 70-µm cell strainer, and enzyme activity was quenched by the addition of FCS followed complete DMEM medium (supplemented with 10% FCS, 2 mM L-glutamine (ThermoFisher) and 10,000 U/mL penicillin/streptomycin (ThermoFisher)). Cell pellets were lysed of red blood cells using commercially available 1× Red Blood Cell Lysis buffer (Thermo-Fisher), and following resuspension in PBS containing 0.5% BSA, cell number was acquired. After 3-h stimulation with PMA and ionomycin, together with Brefeldin A (BioLegend), single-cell suspensions were incubated for Fc block (BioLegend) in 0.5% BSA/PBS buffer for 20 min on ice. Cell surface antibodies were added for 30 min at 4 °C in the dark. Cell surface molecules were stained, and dead cells were identified with Zombie Green or Zombie NIR viability dye (BioLegend). Intracellular staining was performed after fixation and permeabilization with staining kit (00-5523-00, eBioscience). All antibodies used are described in Appendix Table S7. Gating strategy was Dead/Dump(CD19, CD11b and CD11c) negative>CD3$^+$ > CD8$^+$>iL2$^+$, IFNγ$^+$, Granzyme B$^+$ or TNFα$^+$ as appropriate. Fluorescence minus one (FMO) controls were prepared to set gates. Count beads were added to samples to enable the accurate determination of cell numbers. Data were acquired on the BD Fortessa (BD Biosciences) and analysed using FlowJo (BD Biosciences v10.9.0).

## T-cell proliferation

Neutrophils were isolated from the blood of MMTV-*PyMT* tumour-bearing mice using MojoSort Mouse Neutrophil Isolation Kit (Biolegend), with neutrophil retention via a negative selection strategy. T cells were isolated from the spleen and lymph nodes of FVB/N mice using the MojoSort Mouse CD8 T-cell isolation kit (Biolegend) as per the manufacturer's protocol for negative selection strategy. T cells were stained with CellTrace Yellow Proliferation kit (ThermoFisher Scientific). A total of 200,000 T cells were placed into a well of a flat-bottomed 96-well plate, and co-cultured with 200,000 neutrophils in 250 µL Iscove's modified Dulbecco's medium (ThermoFisher), 10% FBS, 50 µM 2-mercaptoethanol (Sigma) with 2 mmol/L L-glutamine (ThermoFisher) and 100 U/mL penicillin–streptomycin (ThermoFisher). T-cell proliferation was stimulated by the addition of CD3/CD28 dynabeads at a ratio of one cell per bead (Gibco). Cells were harvested following 48 h of incubation at 37 °C/5%CO$_2$ and assessed by flow cytometry, using the antibody panel described in Appendix Table S7.

## RNA-sequencing

RNA-sequencing on flow cytometry-sorted immune cell populations from vehicle and LPS-treated mice was performed by Mackey et al (Mackey et al, 2021). To prepare RNA for RNA-sequencing of mammary gland and MMTV-*PyMT* tumours, RNA was prepared from ~25 mg of frozen tissue using Trizol (ThermoFisher) as per the manufacturer's instructions. For *PyMT*$^+$ cell lines, cells lines were previously generated by Gounis et al (Gounis et al, 2025), and plated at $0.5 \times 10^6$–$1 \times 10^6$ cells per 10-cm dish, 37 °C/5% CO$_2$ for 24 h to achieve 80% confluence the next day. Cells were washed twice in ice-cold PBS, lysed into 1 mL Trizol, and RNA prepped as per the manufacturer's instructions. For mouse embryonic fibroblasts (MEFs) treated with uracil: MEFs were plated at

$1 \times 10^6$ cells per 10-cm$^2$ dish in full growth media (DMEM (ThermoFisher) supplemented with 10% FBS, 2 mM L-glutamine (ThermoFisher) and 100 U/mL penicillin/streptomycin (Thermo-Fisher)) supplemented with 10 or 30 μM uracil (Sigma) and incubated at 37 °C/5% $CO_2$ for 24 h. Cells were then washed with ice-cold PBS, scraped into Trizol, and RNA prepped as per the manufacturer's instructions. In all cases, RNA quality was assessed using Agilent 2200 Tapestation using RNA screentape. Libraries for cluster generation and DNA sequencing were prepared using Illumina TruSeq Stranded Total RNA Lib Prep, Gold kit. Quality and quantity of the DNA libraries were assessed on an Agilent 2200 Tapestation (D1000 screentape) and Qubit (ThermoFisher Scientific), respectively. For mouse tissue and MEF experiments, the libraries were sequenced on the Illumina Next Seq 500 using the High Output v2.5, 150 cycles kit ($2 \times 75$ cycles, paired end reads, dual index). For MEFs, $n = 6$ biological replicates were prepared for each condition, however one sample from the vehicle and 1 sample from the 30 μM uracil arm were removed at the data analysis stage due to human RNA contamination. For $PyMT^+$ cells, RNA 6000 Pico Kit and Bioanalyzer (Agilent Technologies) were used to assess RNA integrity, sample library preparation was performed using the Illumina TruSeq RNA Library Prep kit v2, and sequencing was conducted on the NextSeq500 platform using a High Output 75 cycle kit. Normalised read counts are presented.

## Cytokine measurements in mouse sera

Cytokines were measured in mouse serum samples using the Proteome Profiler Mouse XL Cytokine Array (Bio-techne) as per the manufacturer's instructions using 50 μl serum per mouse. Images were taken on a BioRad ChemiDoc imaging system, with dot intensity measured using BioRad Image Lab. The average intensity of duplicate cytokine spots was taken and normalised to the average intensity of the internal control for each membrane.

## Upp1 CRISPR cell line generation

Guide RNA (gRNA) sequences were designed to target Upp1 (mUPP1 Guide 1 F: CACCGCACCAACCTGGACGCGCAGC; mUPP1 Guide 1R: AAACGCTGCGCGTCCAGGTTGGTGC), and cloned into a lentiCRISPR vector (Addgene) (O'Prey et al, 2017; Shalem et al, 2014). HEK293T cells were used to produce lentivirus. Briefly, HEK293K cells were transfected with 10 μg Upp1 lentiCRISPR construct, 7.5 μg pSPAX2 and 4 μg pVSVG packaging plasmids by the calcium phosphate method. After 24 h, the media were refreshed, and recipient mouse embryonic fibroblasts (MEFs) were plated in parallel in 10-cm dishes. The next days, HEK293T media was collected, filtered through a 0.45-μm filter, supplemented with 2.5 μL polybrene (Sigma, 10 mg/mL stock in $H_2O$), and used to replace media on recipient MEFs. Media was also replenished on HEK293T cells, and viral collection and treatment of MEFs were repeated the following day. Viral-transfected cells were selected using blasticidin-containing media, and cells were passaged upon reaching 80% confluence.

## qRT-PCR

Neutrophils were isolated from the femurs of mice using MojoSort Mouse Neutrophil Isolation Kit (Biolegend), with neutrophil retention via negative selection strategy. Cells were collected into Trizol (Life Tech), and RNA isolated as per the manufacturer's instructions. RNA was quantified and 260/280 ratio checked using Nanodrop (ThermoFisher). cDNA was synthesised using 1 μg RNA as template using the Quantitect Reverse Transcription Kit (Qiagen) as per the manufacturer's instructions. qPCR was performed using ×1 SyBr green (Qiagen), 1:10 dilution of cDNA template, 0.5 μM forward primer (ThermoFisher), 0.5 μM reverse primer (ThermoFisher), and reactions conducted using a three step protocol, corresponding to 95 °C for 3 min, ×40 cycles of 95 °C for 20 s, 60 °C for 20 s, 72 °C for 20 s, with a final extension of 72 °C for 5 min, and melt-curve from 65 °C to 95 °C in 0.5 °C increments. Data were collected and analysed in CFX Manager Software (BioRad CFX Maestro 1.0 Version 4.0.2325.0418). Expression of genes of interest was normalised to Actb. Primer sequences are described in Appendix Table S8.

## Cell-derived matrices

Cell culture dishes were coated in 0.2% gelatin (Sigma), cross-linked with 1% glutaraldehyde (Sigma), and quenched with 1 M glycine (Sigma). Dishes were then washed with PBS and incubated with cell growth medium (DMEM (ThermoFisher) supplemented with 10% FBS, 2 mM L-glutamine (ThermoFisher) and 100 U/mL penicillin/strepto-mycin (ThermoFisher)) before addition of mouse embryonic fibroblasts at a concentration of $2 \times 10^5$ cells per well in a six-well plate. The following day, or when cells were deemed 100% confluent, medium was changed to full growth medium supplemented with 50 μg/mL ascorbic acid (Sigma), plus 0, 10 or 30 μM Uracil (Sigma) as per described experimental conditions, and medium was replaced and refreshed every other day for 7 days. Cells were incubated at 37 °C/5%$CO_2$ during the 7-day ECM deposition period. Cells were then washed with D-PBS with calcium and magnesium (named D-PBS from here on in, Sigma), and cells were then removed by incubation with pre-warmed extraction buffer (20 mM $NH_4OH$, 0.5% Triton X-100 in D-PBS (Sigma)). Residual DNA was digested with 10 μg/mL DNase I (Roche), and denuded matrices were then washed and stored in D-PBS with 100 U/mL penicillin/streptomycin (ThermoFisher) at 4 °C until use.

## Time-lapse microscopy

Primary tumour cells derived from MMTV-$PyMT$ mammary tumours ($PyMT^+$ cells) were plated at a density of 80,000 cells per well, in a six-well plate, and allowed to adhere for ~5 h at 37 °C/5% $CO_2$. Cells were then imaged by time-lapse microscopy, at 37 °C/5% $CO_2$, using an inverted phase-contrast microscope (Nikon Eclipse Ti, 10x/0.3 Plan Fluor objective) taking images in 6 positions per well every 10 min for 16 h. Images were sequentially processed to generate time-lapse movies, and assessment of pseudopod length was performed using ImageJ software (Version 1.49).

## Immunofluorescence

For immunofluorescence of fibronectin in FFPE lung, slides were dewaxed using x2 5-min xylene washes, x2 3-min 100% ethanol washes, x1 1-min 95% ethanol wash, x1 1-min 80% ethanol wash, and x1 5-min $dH_2O$ wash. Antigen retrieval was performed by boiling in citrate buffer (10 mM Citric Acid, pH 6, 0.05% Tween 20) for 25 min, samples were then cooled to room temperature (20 min) followed by

endogenous peroxidase blocking in 3% $H_2O_2$ (Sigma-Aldrich, 95329) followed by avidin block (Invitrogen, 004303) and biotin block (Invitrogen, 004303), each for 15 min at room temperature, separated by x3 3-min TBS-T washes. Protein blocking solution (Abcam/SpringBio, DFB125) was then added and slides incubated for 30 min at room temperature. Rabbit polyclonal anti-Fibronectin (Abcam ab2413, 1:100) diluted in antibody diluent (Life Technologies) was added to the slides overnight at 4 °C. Fibronectin was detected using biotinylated goat anti-rabbit Ab (Vector Labs, BA-9400), Avidin-HRP (Vectastain Elite, ABC-HRP Reagent, Peroxidase, RTU, PK-7100) and the Tyramide Signal Amplification (TSA-FITC) system (PerkinElmer). For Fig. 5A,B, confocal images were collected on a Zeiss 710 point-scanning confocal microscope, built on an upright Zeiss Axio Imager.Z2 stand. Images were acquired using a 20×/0.8 Plan-Apochromat objective lens with a confocal pinhole diameter of 33 μm. Multi-channel images were captured sequentially: DAPI using 405 nm excitation and 410–495 nm emission bandwidth, FITC-fibronectin using 488 nm excitation and 495–630 nm emission. Images were collected with a 1× zoom, yielding a pixel size of 0.12 × 0.12 μm, and a 0.47 μs pixel dwell time. For each image, 3 × 3 tiles were captured using the motorised stage (SCAN 130 × 85, Marzhauser) and stitched in the acquisition software (Zen LSM 2.1 Black, Zeiss) to create a resulting image size of 9632 × 9632 pixels. Fibronectin intensity in the lung parenchyma was quantified using FIJI software (Version 2.9.0). For Fig. EV5A, images were acquired using the Olympus VS200 Slide Scanner, enabling whole sample area to be scanned both in brightfield and fluorescence (DAPI (nuclear) and FITC (fibronectin)) at ×20 magnification with a pixel resolution of 0.325 × 0.325 μm, and fibronectin intensity of lung parenchyma was quantified using QuPath (Version 0.5.1). In both instances, to pool data gathered from slides stained/imaged on different days, the mean fluorescence intensity of an individual slide was normalised to the average fluorescence intensity obtained per experimental group by experimental day. Both experimental groups were always represented in each batch of staining and imaging.

For immunofluorescence of cell-derived matrices (CDMs), CDMs were derived as previously described, in glass-bottomed dishes (MatTek). Immediately after production, CDMs were fixed in 4% PFA, blocked in 1% BSA, and incubated in primary antibody for 2 h at room temperature (Mouse anti-fibronectin: BD Pharmingen 610078, diluted 1:100 in blocking buffer). Following washing, CDMs were then incubated with Alexa-555 conjugated secondary antibody (ThermoFisher 1:400). CDMs were imaged using the Opera Phenix High Content Screening System (Harmony High-content Imaging and analysis software version 4.9, PerkinElmer). Fibronectin fibres were identified and quantified using the Columbus Image Data Storage and Analysis System (PerkinElmer version 2.8.0).

## Organotypic collagen plugs

Organotypic plugs were generated from rat tail-derived collagen as previously described (Timpson et al, 2011). Briefly, $2 \times 10^6$ mouse embryonic fibroblasts were added to rat tail collagen in minimal essential medium (ThermoFisher) that had the pH adjusted to slightly acidic conditions, with either 10 or 30 μM uracil (Sigma) as defined per experimental group. Collagen plugs were set at 37 °C/5% $CO_2$ for 10 min, and then fibroblast growth medium was added (DMEM medium supplemented with 10% FCS, 2 mM L-glutamine

(ThermoFisher) and 100 U/mL penicillin/streptomycin (ThermoFisher)), supplemented with 10 or 30 μM uracil (Sigma). Plugs were kept at 37 °C/5% $CO_2$ and medium was refreshed every other day for 7 days. Plugs were cut in half, fixed in 4% PFA, embedded in paraffin in cross-section, and stained using immunohistochemistry techniques. Fibronectin (ab2413, Abcam) was stained on a Leica Bond Rx, with antigen retrieval using ER2 solution (Leica) for 20 min at 100 °C, and sections were rinsed with Leica wash buffer before peroxidase block was performed using an Intense R kit (Leica). To complete IHC staining, sections were rinsed in tap water, dehydrated through graded ethanol's, placed in xylene and then mounted onto coverslips in xylene using DPX mountant (CellPath). Fibrillar collagen in fresh organotypic plugs was visualised using a Trimscope multiphoton microscope (Lavision) equipped with a Ti:Sapphire laser at a wavelength of 850 nm. For each field of view, Z-projections were acquired, and a threshold was applied to the signal. The area of second harmonic generation (SHG) coverage per field of view was determined using ImageJ (Version 1.49). To assess the organisation of fibrillar collagen in the plugs, we applied the grey level co-occurrence matrix (GLCM) and focused on the parameter correlation. This texture analysis method determines the probability of pixels at increasing distances having similar intensities. Thus, if an image consists mainly of anisotropic fibres, it is possible to travel in a straight line away from a given point without much alteration to intensity, and this would give raise to longer decay distances for a given intensity correlation, in comparison to images containing isotropic fibres. For fields of view with unambiguous two-phase decay curves, the mean decay distance was calculated using method as described in (Novo et al, 2018).

## Receptor trafficking

Cells were plated in full growth medium supplemented with either 10 or 30 μM uracil (Sigma) for 24 h at 37 °C/5% $CO_2$. Medium was then removed, cells transferred to ice, washed twice in ice-cold PBS, and surface-labelled at 4 °C with 0.2 mg/mL NHS-SS-biotin (Pierce) in PBS for 30 min. Cells were transferred to full growth medium for 30 min at 37 °C/5% $CO_2$ to allow internalisation of the tracer. Cells were then returned to ice, washed twice with ice-cold PBS and biotin was removed from proteins remaining at the cell surface by reduction with MesNa (Fluka). The internalised fraction was then chased from the cells by returning them to 37 °C/5% $CO_2$ in full growth medium. At the indicated times, cells were returned to ice and biotin removed from recycled proteins by a second reduction with MesNa (Fluka). Biotinylated $\alpha_5\beta_1$ was then determined by capture-ELISA using Maxisorp (Nunc) plates coated with antibodies specific to CD29 (Clone 9EG7, BD Pharmingen 550531) for the ligand-engaged conformation of $\alpha_5\beta_1$, CD49e (Clone 5H10-27, BD Pharmingen 553319) for total $\alpha_5\beta_1$, and transferrin receptor (BD Pharmingen 553264).

## Computational approaches

Survival curves presented in Fig. 1 were generated using KM Plotter (Gyorffy, 2024b). Normalised sequence reads for *Upp1* presented in Appendix Fig. S6A are from the flow cytometry-sorted cells and transcriptomics described in (Mackey et al, 2021). Appendix Fig. S6B uses a Microenvironment Cell Populations-counter

(MCP method) (Becht et al, 2016) on GSE33113. For analysis of publicly available scRNA-seq data, in Fig. 2B and Appendix Fig. S5A, *Upp1* and *Upp2* data were extracted from Fig. 1 of https://github.com/chris-mcginnis-ucsf/pymt_atlas (raw data deposited as GSE263963). For Fig. 2C, Appendix Fig. S4A–E and Appendix Fig. S5B,C, GSE139125 was obtained from the GEO database. Briefly, groups were integrated, normalised, variable features selected, scaled and then linearly dimensionally reduced. Clusters were identified using default settings and identified using key markers of neutrophils. Featureplots and Dotplots were created using Seurat using default settings (Seurat v.5.0.1).

## Statistics

All statistical analyses were performed in GraphPad Prism (Version 10.1.2). Correlations were plotted using Spearman's rank correlation coefficient, which measures the strength and direction of the monotonic relationship between two variables. Unpaired *t* tests were used when considering two independent populations of equal variance. Ordinary one-way ANOVA with Šídák's multiple comparisons test was used as standard when making comparisons across multiple independent populations of equal variance. If data were not normally distributed, a Kruskal–Wallis test was employed, and these instances are described in the relevant figure legends. When experimental units were harvested on different days (such as mice bearing tumours at specific size endpoints), data are presented as a fold change compared to an appropriate control sampled in parallel on the same day. In these instances, one-sample *t* tests have been used to assess whether the experimental group is different to 1. Paired *t* tests were used on paired biological samples subjected to different treatments. All n are represented as dots on graphs and reported numerically in the corresponding figure legends.

## Data availability

All metabolomics data, including raw spectra and associated experimental information, are available in the MassIVE Repository as Study ID MSV000097229, except for data from CD11b-DTR PDAC plasma which is deposited in Metabolomics Workbench as Study ID ST003759. RNA-Sequencing datasets are deposited as GSE292202, GSE292203, GSE292205 in the GEO database.

The source data of this paper are collected in the following database record: biostudies:S-SCDT-10_1038-S44319-025-00520-7.

## Peer review information

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

## Acknowledgements

The authors are grateful to all funders, described below, and the services and resources provided by the CRUK Scotland Institute. The authors express special acknowledgement to the Biological Services Unit, the Flow Cytometry Facility, the Histology Facility, Transgenic Technologies, the Metabolomics Facility and the Beatson Advanced Imaging Resource (BAIR) Research Resource Identifier (RRID): SCR_023875 of the CRUK Scotland Institute. The authors specifically highlight the contributions of the BAIR team members Peter Thomason, Ryan Corbyn and Lyn McGarry for supporting the authors with the data acquisition and analysis required to deliver this manuscript, and Nikki Paul and Claire Mitchell for their constructive review of our methods. The authors thank all members of the Jim Norman lab for constructive feedback throughout the course of this project. Mouse embryonic fibroblasts were kindly provided by the lab of Professor Sara Zanivan, PyMT+ cells from the lab of Professor Karen Blyth, advice regarding use of GEMMs of pancreatic cancer was provided by Professor Jen Morton, technical assistance monitoring KPC mouse colonies was provided by Erin Cumming, and assistance depositing transcriptomic datasets was provided by Robin Shaw, all at the CRUK Scotland Institute. The authors also thank Professor Jos Jonkers (Netherlands Cancer Institute) for the original KP mouse model. All illustrations were created using BioRender.com. The authors reserve special thanks to the patients who allowed their samples to be used for scientific gain, and the murine models used throughout the course of this work. This paper was critically reviewed by Catherine Winchester, and the authors thank her for her contribution and role in research integrity at the CRUK Scotland Institute. The work described in this manuscript was supported by funding awarded to CJC: CRUK (PRCBTP-May24/100005), Wellcome Trust (318046); to JCN: CRUK (Core

Programmes A18277 and A28291), CRUK (A273110), the Medical Research Council (MR/P01058X/1), Breast Cancer Now (2019NovPR1268); to KB: CRUK (Core Programme A29799), UK Medical Research Council (MC_PC_21042); to JVV: CRUK (RCCCDF-Nov23/100001); a Pancreatic Cancer UK Research Innovation Grant awarded to CJC and JVV (2021RIF_22_Voorde), and CRUK Scotland Institute Core Funding (OJS, CRUK Scotland Institute Director, A31287).

## Author contributions

**Declan Whyte**: Formal analysis; Investigation; Writing—review and editing. **Sophie L Fisher**: Formal analysis; Investigation. **Christopher GJ McKenzie**: Formal analysis; Investigation. **David Sumpton**: Data curation; Formal analysis; Investigation; Methodology. **Sandeep Dhayade**: Resources; Formal analysis; Investigation. **Emmanuel Dornier**: Resources; Formal analysis; Investigation. **Madeleine Moore**: Formal analysis; Investigation. **David Novo**: Formal analysis; Investigation. **Jasmine Peters**: Formal analysis; Investigation. **Robert Wiesheu**: Formal analysis; Investigation. **Michalis D Gounis**: Resources; Formal analysis; Investigation. **Dale M Watt**: Formal analysis; Investigation. **John BG Mackey**: Resources. **Amanda J McFarlane**: Methodology. **Frédéric Fercoq**: Methodology. **Carolina Dehesa Caballero**: Formal analysis; Investigation. **Keara L Redmond**: Formal analysis. **Louise E Mitchell**: Formal analysis; Investigation. **Eve Anderson**: Resources; Investigation. **Gemma Thomson**: Resources. **Ann Hedley**: Formal analysis. **William Clark**: Resources; Investigation. **Shannen Leroi**: Resources. **Lindsey N Dzierozynski**: Resources. **Juan J Apiz Saab**: Resources. **Caroline A Lewis**: Resources. **Alexander Muir**: Resources; Data curation; Supervision. **Christopher J Halbrook**: Resources; Supervision. **Douglas Strathdee**: Resources; Supervision. **Rene Jackstadt**: Resources. **Colin Nixon**: Resources; Supervision; Methodology. **Philip Dunne**: Supervision. **Leo M Carlin**: Supervision; Methodology; Writing—review and editing. **Iain R Macpherson**: Resources. **Edward W Roberts**: Formal analysis. **Seth B Coffelt**: Supervision; Writing—review and editing. **Karen Blyth**: Conceptualisation; Resources; Supervision; Funding acquisition; Writing—review and editing. **Owen J Sansom**: Conceptualisation; Supervision; Funding acquisition. **Jim C Norman**: Conceptualisation; Formal analysis; Supervision; Funding acquisition; Investigation; Writing—review and editing. **Johan Vande Voorde**: Conceptualisation; Formal analysis; Supervision; Funding acquisition; Investigation; Writing—review and editing. **Cassie J Clarke**: Conceptualisation; Resources; Data curation; Formal analysis; Supervision; Funding acquisition; Investigation; Visualisation; Writing—original draft; Project administration; Writing—review and editing.

Source data underlying figure panels in this paper may have individual authorship assigned. Where available, figure panel/source data authorship is listed in the following database record: biostudies:S-SCDT-10_1038-S44319-025-00520-7.

## Disclosure and competing interests statement

The authors declare no competing interests.

# Expanded View Figures

**Figure EV1. Primary tumours, diet and age, are not responsible for the UPP1-dependent increases in uracil seen in metastatic cancer.**

(A) Serum uracil was assessed by LC–MS in mice FVB/N ($n = 5$ mice), or mice that were transplanted with KP tumour fragments (tumours grown to 10 mm diameter, $n = 4$ $Upp1^{+/+}$ recipient mice, $n = 4$ $Upp1^{-/-}$ recipient mice). (B) Upp1 was assessed in mammary gland ($n = 8$ mice), or $PyMT^+$ mammary tumour ($n = 31$ mice) by RNA-seq. (C) Upp1 was assessed by qRT-PCR in $PyMT^+$ mammary tumours from mice that had 0 ($n = 7$ mice), 1–10 ($n = 16$ mice) or >10 ($n = 6$ mice) metastasis detectable by histological assessment of the lung. (D) Upp1 in MMTV-PyMT tumours was assessed by qPCR and serum uracil in matched mice assessed by LC–MS ($n = 29$ mice in total, black dots represent mice with no metastasis, and pink and green dots represent mice with 1–10 or >10 metastases, respectively). (E) Upp1 was assessed by RNA-seq from cell lines derived from primary mammary tumours (that were a consequence of fat-pad transplantation of $PyMT^+$ cell lines), or isogenic cells derived from micrometastases that formed in the lung following primary tumour resection (cell line generation and characterisation described in (Gounis et al, 2025) ($n = 6$ independent cell lines from $n = 6$ mice). (F, G) FVB/N mice were dosed daily, by oral gavage, with normal water or water containing 10 mM uracil for a total of 4 days. Mice were sacrificed 2 h post dosing on day 4. Uracil was then assessed in the serum (F), or kidney and lung (G) by LC–MS ($n = 4$ mice per experimental group). (H) Serum uracil in FVB/N mice younger and older than 90 days ($n = 3$ mice <90 days, and $n = 6$ mice >90 days old). (I) Age of mice at clinical endpoint presented in relation to the number of lung metastasis per mouse ($n = 30$ mice). Data Information: In bar graphs, data are presented as mean ± SEM. In (A–D, F–I) dots represent individual mice. In (E), each dot represents the average of technical triplicate repeats for each cell line. When more than 1 comparison is made (A, C, G) $P$ values were calculated through one-way ANOVA, when 2 experimental groups were compared (B, E, F, H) unpaired $t$ test is used, ns = not statistically significant. For correlation plots, Spearman Correlation statistics are presented.

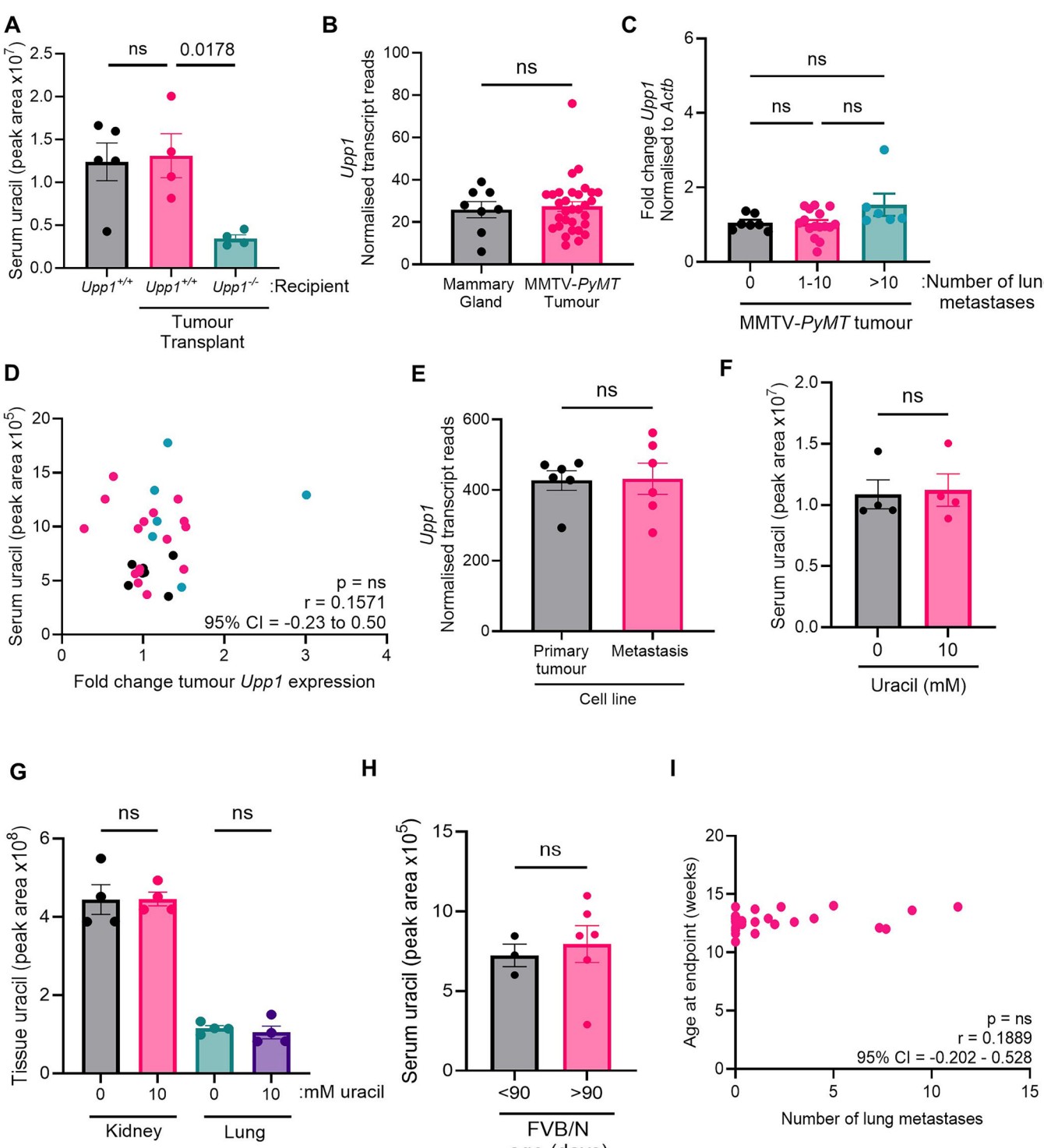

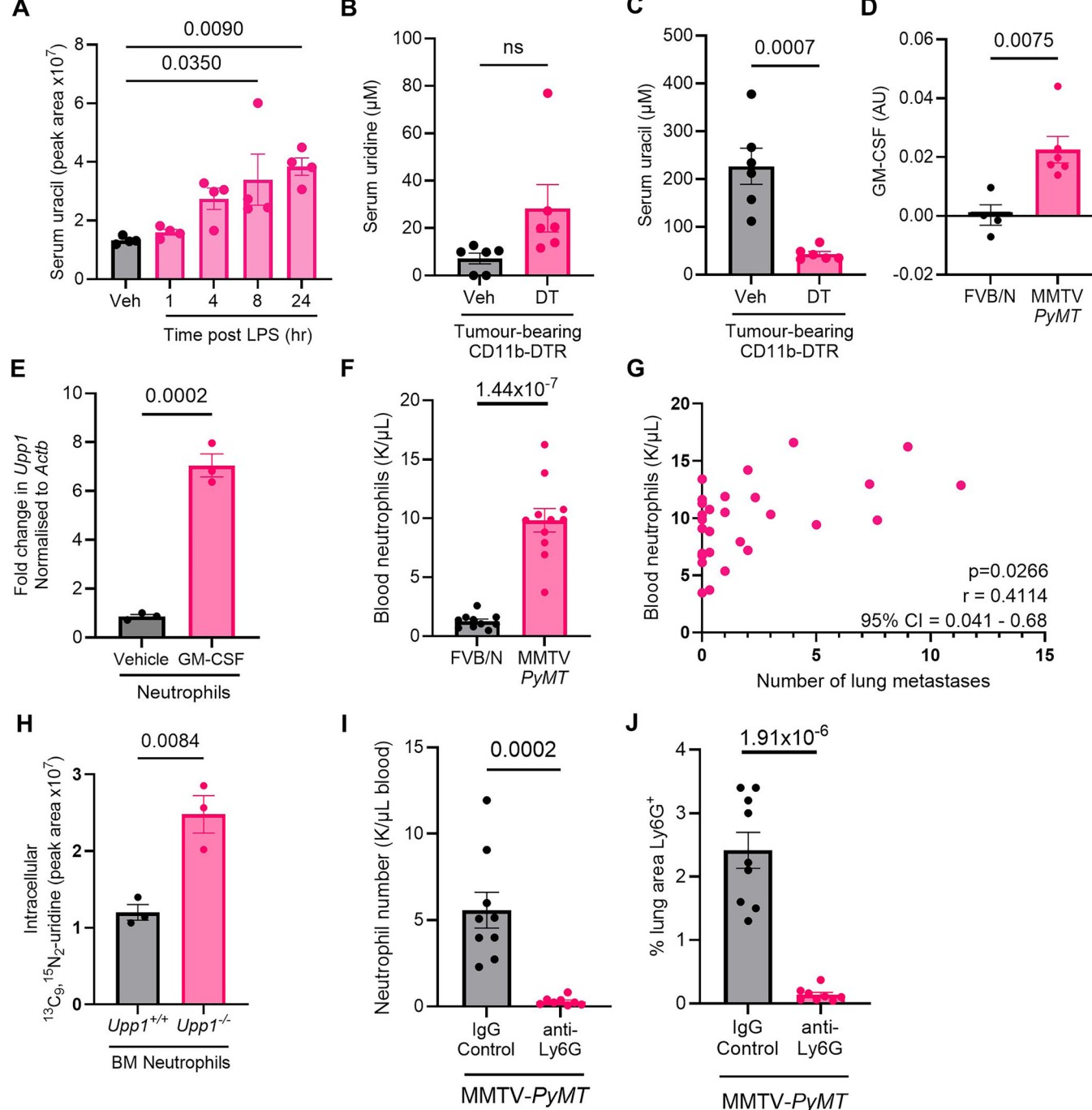

**Figure EV2. Neutrophils are the significant source of *Upp1* in metastatic mammary cancer.**

(A) FVB/N mice were dosed intraperitoneal with 0.5 mg/kg lipopolysaccharide (LPS), or vehicle control, and serum uracil measured by LC–MS at defined timepoints post dosing ($n = 4$ mice per experimental group). (B, C) Mice bearing orthotopic PDAC tumours were treated with diphtheria toxin (DT) to ablate myeloid cells, and serum uridine (B) and uracil (C) were assessed after 24 h by LC–MS ($n = 6$ mice per experimental group). (D) GM-CSF was measured in the serum of FVB/N ($n = 4$ mice) and MMTV-*PyMT* tumour-bearing mice ($n = 6$ mice) by ELISA. (E) *Upp1* was assessed by qRT-PCR in BM neutrophils treated ex vivo with vehicle or 20 ng/mL GM-CSF for 24 h ($n = 3$ mice). (F) Blood neutrophils determined by IDEXX in MMTV-*PyMT* tumour-bearing mice ($n = 11$ mice) and 14-week-old FVB/N controls ($n = 10$ mice). (G) Number of blood neutrophils determined by IDEXX in MMTV-*PyMT* tumour-bearing mice at clinical endpoint ($n = 29$ mice), compared to number of lung metastases determined by histological analysis. (H) Intracellular $^{13}C_9,^{15}N_2$-uridine detected by LC–MS in BM neutrophils isolated from female FVB/N *Upp1$^{+/+}$* and *Upp1$^{-/-}$* mice, that were incubated for 24 h with $^{13}C_9,^{15}N_2$-uridine ($n = 3$ mice per experimental group). (I) Blood neutrophils determined by IDEXX and in MMTV-*PyMT* tumour-bearing mice treated with IgG control ($n = 9$ mice) or anti-Ly6G ($n = 8$ mice). (J) The proportion of lung area positive for Ly6G by immunohistochemistry for mice described in I (IgG control $n = 9$ mice; anti-Ly6G $n = 8$ mice). Data Information: In bar graphs dots represent individual mice and data are presented as mean ± SEM. When more than 1 comparison was made (A) $P$ values were calculated through one-way ANOVA, when 2 experimental groups are compared (B–F, H–J) unpaired $t$ test is used, ns = not statistically significant. For correlation plots (G), Spearman Correlation statistics are presented.

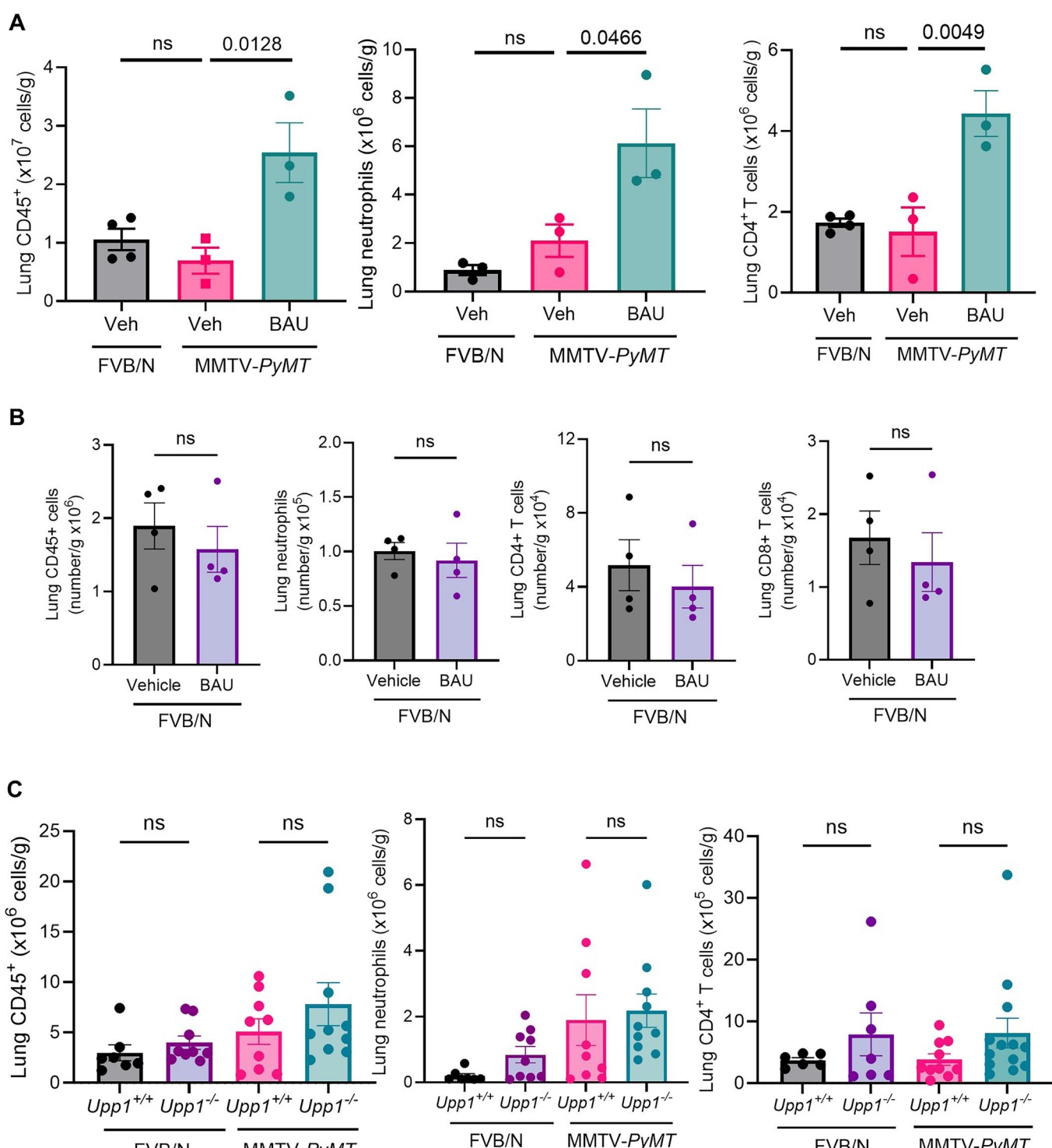

**Figure EV3. The effect of UPP1 on the immune landscape of the lung.**

Cell number per gram of lung for CD45$^+$, neutrophils, and CD4$^+$ T cells, assessed by flow cytometry, in the lungs of (A) MMTV-*PyMT* mice treated with vehicle ($n = 3$ mice) or BAU ($n = 3$ mice) from palpable tumour until mammary tumours reached 10–15 mm in diameter, and FVB/N ($n = 4$ mice) treated with vehicle for time-matched periods. (B) FVB/N mice treated with vehicle ($n = 4$ mice) or BAU ($n = 4$ mice) for 12 days. (C) MMTV-*PyMT*;*Upp1*$^{+/+}$ ($n = 9$ mice) and MMTV-*PyMT*;*Upp1*$^{-/-}$ ($n = 10$ mice) mammary tumour-bearing mice were harvested when one tumour measured 10–15 mm diameter. $N = 7$ and $n = 9$ FVB/N *Upp1*$^{+/+}$ and *Upp1*$^{-/-}$ mice were taken as age-matched controls, respectively. Data Information: In all cases, dots represent individual mice and data are presented as mean ± SEM. When more than 1 comparison is made (A, C) $P$ values are calculated through one-way ANOVA, when 2 experimental groups are compared (B) unpaired $t$ test is used, ns = not statistically significant.

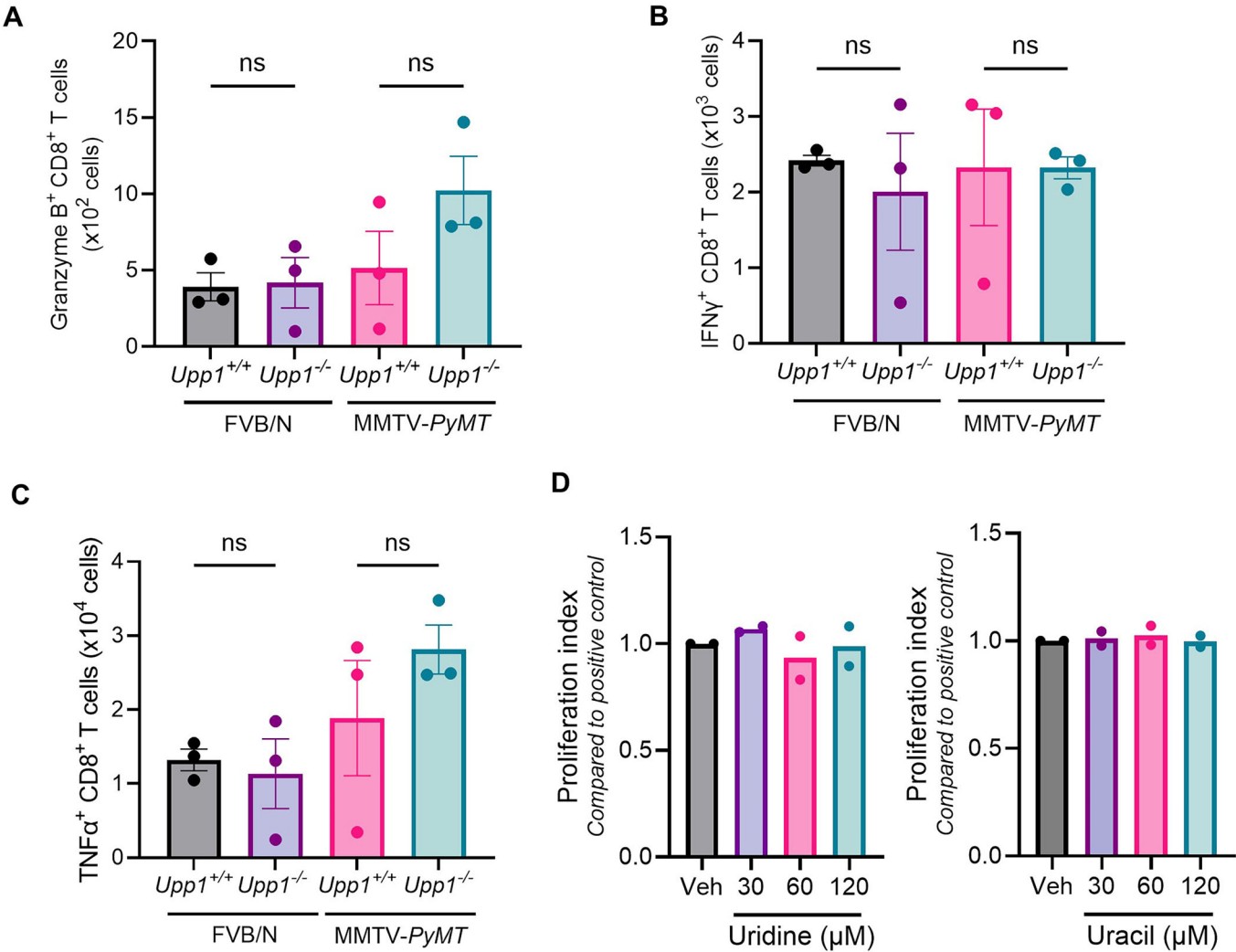

**Figure EV4. Understanding T-cell effector function in the absence of *Upp1*.**

(A–C) Lungs of MMTV-*PyMT;Upp1*$^{+/+}$ and MMTV-*PyMT;Upp1*$^{-/-}$ tumour-bearing mice were harvested when one tumour measured 10–15 mm diameter. FVB/N age-matched controls were taken in parallel. Cells were prepared as described in the methods for intracellular T-cell staining ($n = 3$ mice per experimental group). (D) T cells were isolated from FVB/N mice, incubated in culture with CD3/CD28 dynabeads to stimulate proliferation in medium containing 30, 60 and 120 µM uridine or uracil. Proliferation index was calculated compared to the positive control of T cells and vehicle alone stimulated with beads ($n = 2$ mice). Data Information: In all cases, dots represent individual mice. (A–C) Data are mean ± SEM, and *P* values were calculated by one-way ANOVA. As data in (D) is $n = 2$, data are presented as mean with datapoints, without error bars and statistics.

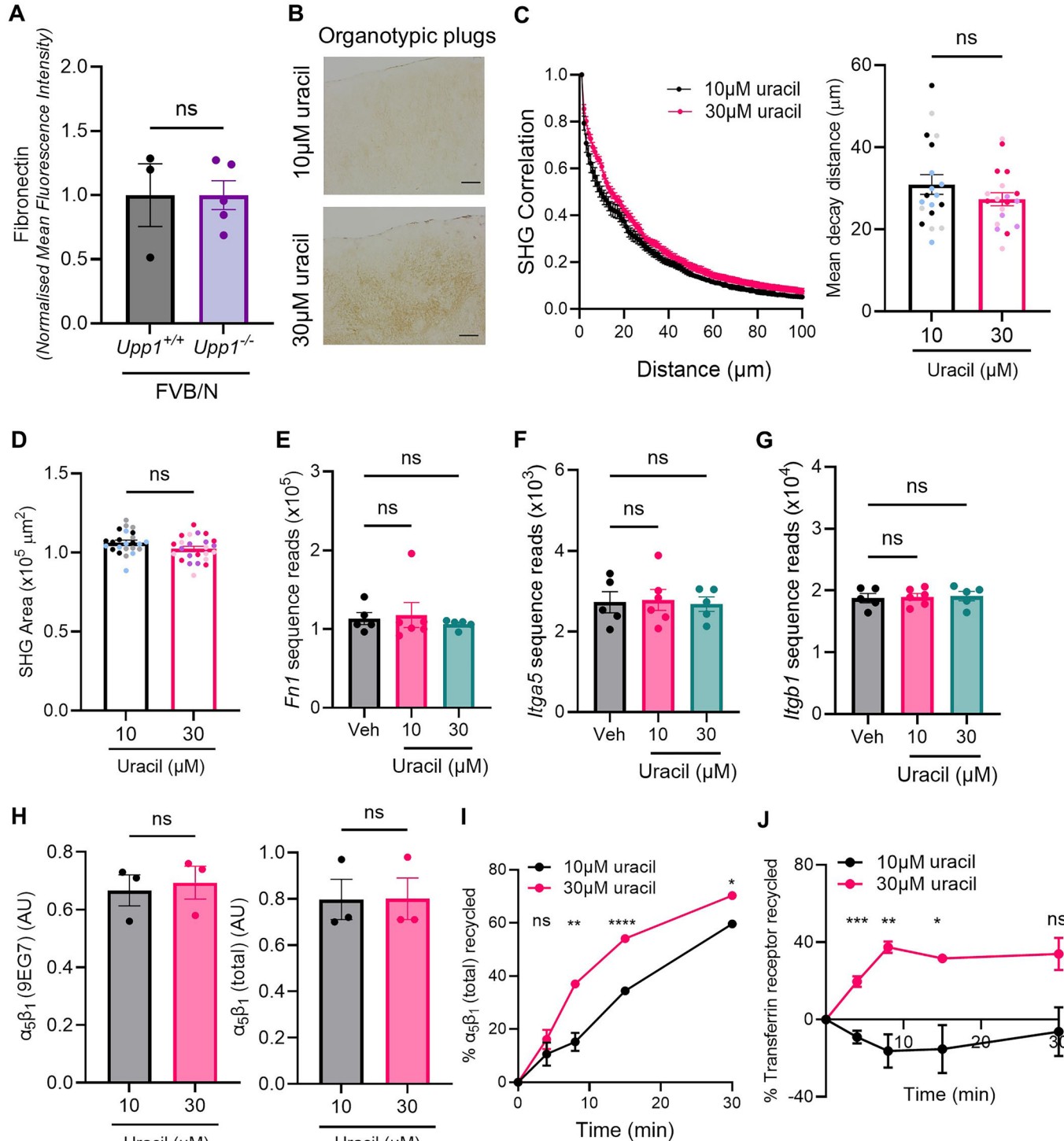

◄ **Figure EV5. Understanding the effect of *Upp1* and uracil on ECM deposition.**

(A) Lung fibronectin was assessed in FVB/N *Upp1*$^{+/+}$ ($n = 3$ mice) and *Upp1*$^{-/-}$ ($n = 5$ mice) mice by immunofluorescence, age-matched to those presented in Fig. 5A, B. (B) Fibroblasts were added to rat tail collagen with medium containing 10 µM or 30 µM uracil to make organotypic plugs. Plugs were contracted by fibroblasts for 7 days and stained for fibronectin by immunohistochemistry. Representative images are shown for $n = 4$ biological repeat experiments, scale bar 100 µm. (C, D) Fibrillar collagen was imaged by second harmonic generation (SHG) microscopy in organotypic plugs contracted by fibroblasts supplemented with 10 or 30 µM uracil. A threshold was applied to the SHG signal and the area of SHG coverage per field of view was determined. The organisation of fibrillar collagen in each field of view was assessed by applying grey level co-occurrence matrix. The mean correlation decay curves from each experimental condition, and the mean of the decay distances derived from those curves, are presented (n = 3 biological repeat experiments colour coded). (E–G) RNA-Seq transcript reads from fibroblasts treated with vehicle, 10 µM and 30 µM uracil for 24 h, for fibronectin (*Fn1*), integrin α$_5$ (*Itga5*) and integrin β$_1$ (*Itgb1*) ($n = 5$ biological repeats for vehicle and 30 µM condition, $n = 6$ biological repeats for 10 µM uracil). (H) Total protein levels quantified from total cell lysate via ELISA, arbitrary units (AU) ($n = 3$ biological repeats). (I) Recycling of the total levels of the fibronectin receptor, α$_5$β$_1$ integrin, assessed in fibroblasts treated with 10 µM or 30 µM uracil for 24 h ($n = 3$ biological repeats). (J) Recycling of transferrin receptor assessed in fibroblasts treated with 10 µM or 30 µM uracil for 24 h ($n = 3$ biological repeats). Data Information: In all graphs, data are mean ± SEM. In (A) dots represent individual mice. In (C, D), dots are $n = 24$ fields of view across $n = 3$ colour coded biological repeat experiments. $N = 19$ fields of view are represented for mean decay distance as 6 fields of view did not conform to 2 phase decay curves and so could not be fitted in the equation. In (E–J), dots represent biological experiment repeats. When 2 experimental groups are compared (A, C, D, H–J) unpaired *t* test is used. For (I), *$P = 0.0042$; **$P = 0.0041$, ****$P = 4.62 \times 10^{-5}$. For (J), *$P = 0.0207$; **$P = 0.0042$; ***$P = 0.0025$. When more than 1 comparison is made (E–G) *P* values are calculated through one-way ANOVA, ns = not statistically significant.

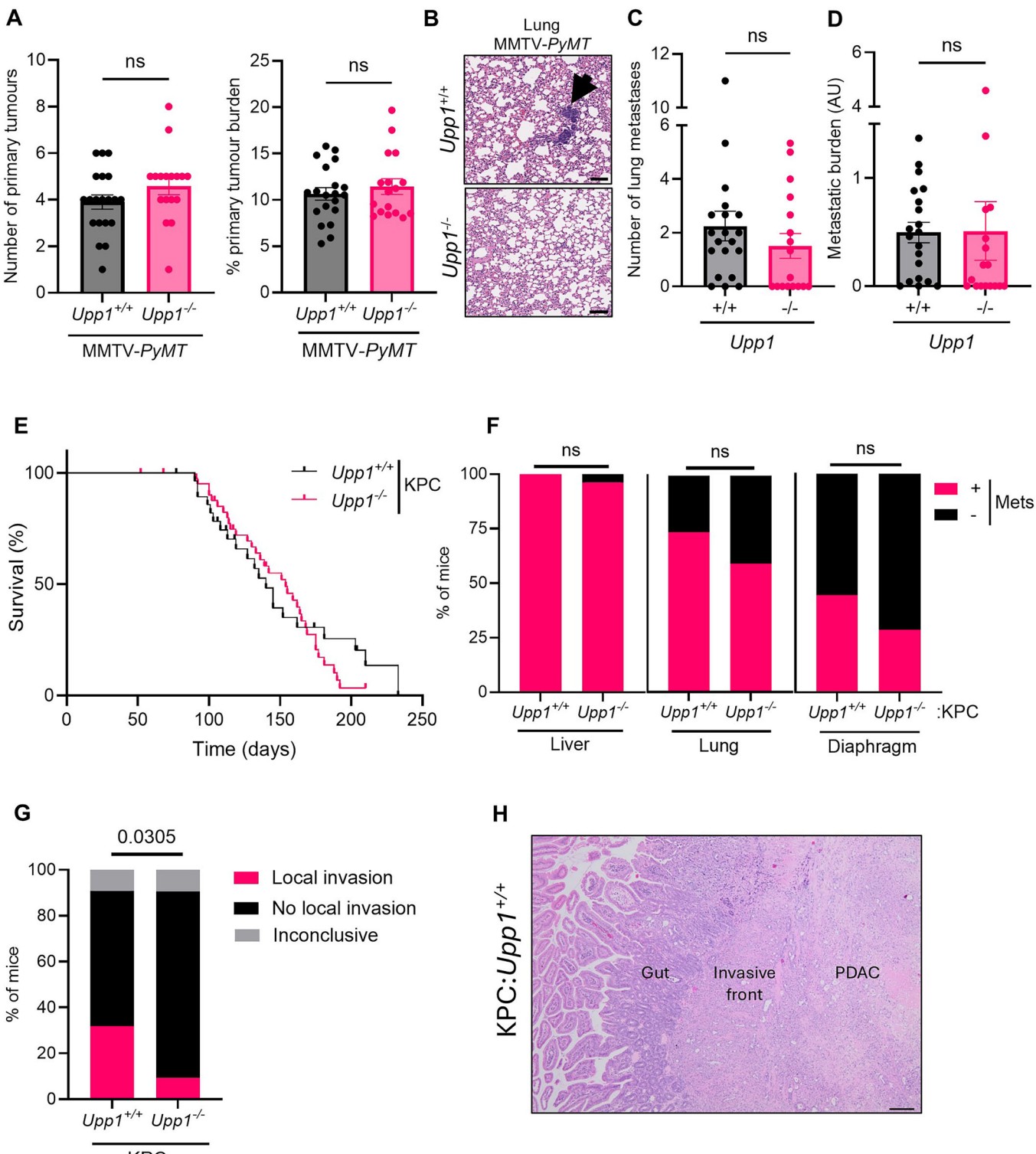

◀ **Figure EV6. Understanding the effect of *Upp1* loss in genetically engineered mouse models of metastatic cancer.**

(A) Mammary tumour number and burden in mice MMTV-*PyMT;Upp1*$^{+/+}$ (n = 20 mice) and MMTV-*PyMT;Upp1*$^{-/-}$ (n = 17 mice) at clinical endpoint. (B) Representative H&E stains for the lungs of MMTV-*PyMT;Upp1*$^{+/+}$ and MMTV-*PyMT;Upp1*$^{-/-}$ mice described in (A), scale bar 100 μm. Arrow highlights a metastatic lesion. (C, D) Average number of lung metastases (C) and average lung metastatic burden (D), determined by histological H&E assessment of serial lung sections from MMTV-*PyMT;Upp1*$^{+/+}$ (n = 20 mice) and MMTV-*PyMT;Upp1*$^{-/-}$ (n = 17 mice). (E) Overall survival of KPC:*Upp1*$^{+/+}$ (n = 22 mice) and KPC:*Upp1*$^{-/-}$ mice (n = 33 mice). Mice that were sacrificed due to indications other than PDAC were censored (amounting to n = 7 *Upp1*$^{+/+}$ mice and n = 10 *Upp1*$^{-/-}$ mice). (F) The proportion of mice with metastasis detected by histological H&E assessment of serial sections of liver and lung, KPC:*Upp1*$^{+/+}$ (n = 19 mice) and KPC:*Upp1*$^{-/-}$ (n = 27 mice), and the proportion of mice with microscopic diaphragm metastasis for KPC:*Upp1*$^{+/+}$ (n = 18 mice) and KPC:*Upp1*$^{-/-}$ mice (n = 21 mice). (G) Local invasion was determined for KPC:*Upp1*$^{+/+}$ (n = 22) and KPC:*Upp1*$^{-/-}$ (n = 32) mice. Inconclusive annotation refers to samples that were unclear at necropsy as to whether full invasion had occurred. (H) Representative image of local invasion that was scored at necropsy, namely the attachment and invasion of the primary PDAC to other organs within the abdominal cavity. Example shown is from a KPC:*Upp1*$^{+/+}$ mouse, with PDAC invading intestinal tissue, scale bar 200 μm. Data Information: In (A, C, D) dots represent individual mice, data are presented as mean ± SEM, and *P* values were calculated by unpaired *t* test, ns = not statistically significant. In (E), log-rank (Mantel–Cox) test *P* = 0.5926. In (F, G), *P* values calculated by chi-squared test.

