## [Peer Review File · EMBO Reports]

Uridine Phosphorylase-1 supports metastasis by altering immune and extracellular matrix landscapes

Declan Whyte, Sophie Fisher, Christopher McKenzie, David Sumpton, Sandeep Dhayade, Emmanuel Dornier, Madeleine Moore, David Novo, Jasmine Peters, Robert Wiesheu, Michalis Gounis, Dale Watt, John Mackey, Amanda McFarlane, Frédéric Fercoq, Carolina Dehesa Caballero, Keara Redmond, Louise Mitchell, Eve Anderson, Gemma Thomson, Ann Hedley, William Clark, Shannen Leroi, Lindsey Dzierozynski, Juan Apiz Saab, Caroline Lewis, Alexander Muir, Christopher Halbrook, Douglas Strathdee, Rene-Filip Jackstadt, Colin Nixon, Philip Dunne, Leo Carlin, Iain Macpherson, Edward Roberts, Seth Coffelt, Karen Blyth, Owen Sansom, Jim Norman, Johan Vande Voorde, and Cassie Clarke

Corresponding author(s): Cassie Clarke (c.clarke@crukscotlandinstitute.ac.uk) , Jim Norman (j.norman@crukscotlandinstitute.ac.uk), Johan Vande Voorde (johan.vandevoorde@glasgow.ac.uk)

Review Timeline:

Submission Date:	2nd Jul 24
Editorial Decision:	22nd Jul 24
Revision Received:	18th Mar 25
Editorial Decision:	28th Apr 25
Revision Received:	20th Jun 25
Accepted:	25th Jun 25

Editor: Achim Breiling

Transaction Report:

Dear Dr. Clarke

Thank you for the submission of your manuscript to EMBO reports. I have now received the reports from the three referees that were asked to evaluate your study, which can be found at the end of this email.

As you will see, the referees find the study very interesting, but they also have several comments, concerns, and suggestions, indicating that a major revision of the manuscript is necessary to allow publication of the study in EMBO reports. As the reports are below, and all the concerns need to be addressed, I will not detail them further here.

Acceptance of your manuscript will depend on a positive outcome of a second round of review. It is EMBO reports policy to allow a single round of revision only and acceptance of the manuscript will therefore depend on the completeness of your responses included in the next, final version of the manuscript.

1) a .docx formatted version of the final manuscript text (including legends for main figures, EV figures and tables), but without the figures included. Figure legends should be compiled at the end of the manuscript text.

2) individual production quality figure files as .eps, .tif, .jpg (one file per figure), of main figures and EV figures. Please upload these as separate, individual files upon re-submission.

4) a complete author checklist, which you can download from our author guidelines

(<https://www.embopress.org/page/journal/14693178/authorguide>). Please insert page numbers in the checklist to indicate where the requested information can be found in the manuscript. The completed author checklist will also be part of the RPF.

5) that primary datasets produced in this study (e.g. RNA-seq, ChIP-seq, structural and array data) are deposited in an appropriate public database. If no primary datasets have been deposited, please also state this in a dedicated section (e.g. 'No primary datasets have been generated and deposited'), see below.

The accession numbers and database should be listed in a formal "Data Availability" section (placed after Materials & Methods) that follows the model below. This is now mandatory (like the COI statement). Please note that the Data Availability Section is restricted to new primary data that are part of this study. This section is mandatory. As indicated above, if no primary datasets have been deposited, please state this in this section

Data availability

8) Regarding data quantification and statistics, please make sure that the number "n" for how many independent experiments were performed, their nature (biological versus technical replicates), the bars and error bars (e.g. SEM, SD) and the test used to calculate p-values is indicated in the respective figure legends (also for EV figures and all those in an Appendix). Please also check that all the p-values are explained in the legend, and that these fit to those shown in the figure. Please provide statistical testing where applicable. Please avoid the phrase 'independent experiment', but clearly state if these were biological or technical replicates. Please also indicate (e.g. with n.s.) if testing was performed, but the differences are not significant. In case n=2, please show the data as separate datapoints without error bars and statistics. See also: <http://www.embopress.org/page/journal/14693178/authorguide#statisticalanalysis>

9) Please add scale bars of similar style and thickness to microscopic images, using clearly visible black or white bars (depending on the background). Please place these in the lower right corner of the images themselves. Please do not write on or near the bars in the image but define the size in the respective figure legend.

10) Please also note our reference format:

12) We now use CRediT to specify the contributions of each author in the journal submission system. CRediT replaces the author contribution section. Please use the free text box to provide more detailed descriptions and do NOT provide your final manuscript text file with an author contributions section. See also our guide to authors: <https://www.embopress.org/page/journal/14693178/authorguide#authorshipguidelines>

13) All Materials and Methods need to be described in the main text using our 'Structured Methods' format, which is required for all research articles. According to this format, the Materials and Methods section should include a Reagents and Tools Table (listing key reagents, experimental models, software, and relevant equipment and including their sources and relevant

identifiers), uploaded as separate file, followed by a Methods and Protocols section in which we encourage the authors to describe their methods using a step-by-step protocol format with bullet points, to facilitate the adoption of the methodologies across labs. More information on how to adhere to this format as well as downloadable templates (.doc) for the Reagents and Tools Table can be found in our author guidelines (section 'Structured Methods'):

14) Please reduce the number of keywords to five and order the manuscript sections like this, using these names: Title page - Abstract - Keywords - Introduction - Results - Discussion - Methods - Data availability section - Acknowledgements - Disclosure and Competing Interests Statement - References - Figure legends - Expanded View Figure legends

I look forward to seeing a revised form of your manuscript when it is ready.

Please use this link to submit your revision: <https://embor.msubmit.net/cgi-bin/main.plex>

Yours sincerely,

Referee #1:

Whyte and colleagues investigate the immunometabolism of cancer and metastasis. The authors start by profiling circulating metabolites in mouse models of cancer and highlight uracil, a metabolite generated from uridine by the enzyme uridine phosphorylase (Upp1/Upp2 genes). They focus their investigation on neutrophils and show high cancer-related Upp1 activity in neutrophils, leading to an altered expression pattern of adhesion molecules on the surface of these cells. Additionally, they report an increased T cell number and reduced fibronectin in the lungs of Upp1^{-/-} (or inhibited) animals. Finally, the authors demonstrate that Upp1 is required for metastasis formation *in vivo*.

I found this article interesting, well-written, and sound. I would, of course, have liked to get more mechanistic insights into how uracil works, but the work presented in this article is already substantial. After a few revisions, I believe it will be a solid candidate for EMBO Reports.

Comments:

- The authors should clarify which aspects of the phenotype in the Upp1^{-/-} (or inhibited) mouse are attributable to neutrophils and which are not. For example, in the abstract, it states, "Uracil is generated by neutrophils expressing Upp1," which implies that neutrophils are the major source of uracil in the organism and are thus responsible for most of the phenotype. This is also depicted in the graphical abstract in Fig. 6D, which is centered on a neutrophil. I don't believe this is the case, since other immune cells, the diet and the tumors themselves generate high levels of uracil. In the absence of a neutrophil-specific knockout of Upp1, I would encourage the authors to either provide additional data, or strongly attenuate their abstract/conclusions by more clearly stating that they chose to focus on neutrophils and discuss that other cell lines from the myeloid lineage, the diet as well as the tumors are known to express high Upp1 levels and were not investigated here.
- Related to the above point, the authors justify focusing on neutrophils based on the CD11b-DTR results. CD11b is a pan-myeloid marker, and monocytes and macrophages. Those cells were recently highlighted for their high, immune-related Upp1 expression (PMIDs 37198494, 37198474), and they will also be deleted upon expression of CD11b-DTR. Here too, the authors should better explain why they decided to focus on neutrophils and highlight the potential role played by other cells from the myeloid lineage, especially macrophages, and the fact that they were not investigated here.
- By which mechanism is Upp1 upregulated in neutrophils?
- The Upp1^{-/-} mouse model developed by the authors will certainly be an interesting tool to investigate uridine metabolism *in vivo*. It should be noted that a Upp1^{-/-} line was previously generated (PMID 15772079 - perhaps this should be cited?), but no immune phenotype was reported. Could the authors better characterize the immune system of the Upp1^{-/-} mouse, even

globally? Besides the lung, are the global numbers of circulating immune cells similar when compared to +/- mice? Is the deposition of fibronectin in the lungs of Upp1^{-/-} mice also decreased in healthy animals? Some figures, such as Fig. 3B, 4A, 5A/B and S5B, lack a control in healthy animals.

- Could the authors clarify why there are less metastasis in Upp1^{-/-} mice? Is it due to the inability of the cancer to form metastasis (as was reported for primary pancreatic cancers growth) - or it is due to the immune system? What would happen in a CD11b-DTR model (similar experiment as in S3C but counting metastasis burden)?
- Based on the authors observation, does Upp2 play any role in systemic uridine (immuno)metabolism?

Referee #2:

The manuscript by Whyte et al. reports elevated Uracil concentrations with progression in mouse models with different cancers and patients with cancer and identifies neutrophils to express higher levels of the uracil-synthetizing enzyme UPP1 upon LPS stimulation or in advanced tumour settings. They further characterize activities of neutrophils with UPP1 deficiency or inhibition in vitro and show that MMTV-PyMT full body UPP1 knockout mice have reduced lung fibrosis, increased lung CD8⁺ T cell numbers and reduced lung metastasis. This is a descriptive manuscript without an investigation of the underlying mechanism. The metabolite Uracil/ uridine-derived ribose and tumour cell-expressed UPP1 (as an oncogene) have been shown previously to drive progression of several cancer types, including lung and pancreatic cancer (PMID: 38385072, PMID: 37198494, PMID: 37198474, PMID: 38331898, PMID: 32583596). Yet, the notion that neutrophils are an important source of Uracil and the precise effects of neutrophil-derived Uracil are interesting and novel. The authors conduct cell autonomous experiments with neutrophils in vitro and complement those with full body knockout mice for UPP1 in vivo. However, due to the expression of UPP1 by tumour cells (and the published resulting effects) the in vivo relevance of this finding (neutrophil-derived-uracil or neutrophil intrinsic loss of UPP1) is not clear. Moreover, the authors exclusively focus on effects of uracil/UPP1/neutrophils in the lung foremost in one breast cancer model, despite detecting increased Uracil systemically in several different cancer models. Considering the lack of mechanistic insights in the present manuscript, the description & characterization of the uncovered phenomenon should be strengthened.

For those reasons, my more detailed comments below should be addressed before publication of the manuscript in EMBO reports.

Main comment 1 - It is unclear if neutrophil-derived Uracil / neutrophil-intrinsic loss of UPP1 is relevant for cancer progression/metastasis?

As mentioned above, the authors do report cell-autonomous effects of UPP1-deficient neutrophils in culture, but the in vivo relevance of such activities remains poorly investigated due to the use of a full body UPP1 knockout mouse model. Hence, in vivo, UPP1 expressed by cancer cells (as previously described), T cells and fibroblast themselves could account for observed effects. Also, it is not clear if neutrophil-expressed UPP1 is relevant for systemic Uracil concentrations. Ideally, the best experiment would be using mice with a neutrophil-specific deletion of UPP1, however I appreciate that this would be challenging. Hence, to address those important open questions, I suggest the following experimental approaches:

- (1) Perform experiments as in Supp Fig. 3B-C (measuring systemic Uracil) by specifically deleting neutrophils using the anti-Ly6G antibody (for example as in PMID: 26649828). Quantify Upp1 expression in CD11b⁺ cells in the lung (including macrophages). CD11b also affects macrophages, which is a very prevalent cell type. Those experiment will determine the importance of neutrophil-derived uracil to systemic levels.
- (2) Measure systemic Uracil levels (as in Supp. Fig. 3B-C) and metastatic progression (as in Figure 6) in control and UPP1-ko FVB mice transplanted with WT tumours and quantify UPP1 expression in MMTV-PyMT tumour cells. This will uncover the contribution of UPP1 expressed by tumour cells to Uracil accumulation and metastatic progression.
- (3) Measure UPP1 expression in lung fibroblasts and lung CD8⁺ T cells in control and MMTV-PyMT mice. Analyse Fibronectin accumulation (as in Fig.5A) and CD8⁺ T cell numbers (as in Fig.4A-C) in the lung of Neutrophil-depleted (with anti-Ly6G antibody) WT and UPP1-ko MMTV-PyMT mice. If UPP1-expressing neutrophils ultimately cause fibronectin deposition or T cell inhibition in vivo, total neutrophil depletion should have an effect in WT mic but not UPP1-ko mice.

Main comment 2 - are the effects of neutrophil UPP1 specific for altering the lung microenvironment (and metastasis to the lung) in breast cancer? Or across organs/cancer types?

The authors report high systemic Uracil levels in several mouse models of metastatic cancer. Yet, they focus their characterization of neutrophils entirely on the lung of the MMTV-PyMT cancer model. To understand the relevance of neutrophil-derived UPP1 in the metastatic progress of the other used cancer models and metastasis to a second site, I strongly recommend the following experiments: Analysis of progression (especially lung and liver metastasis) in KPC (PDAC model) and KPN (CRC model) upon treatment with BAU (alternatively UPP1-deficiency, but that would require additional crossing of mice and could be avoided using the inhibitor) and neutrophil depletion.

Main comment 3 - Integrated metabolomics analysis of the serum of mouse cancer/metastasis models is poorly reported. Right now, the analyses and data not very clearly depicted in Figure 1 and Supp Figure 1. Please, for the sake of open science, indicate all identified metabolites (16 + 17 + 5 + 19) in the analysis Supp. Fig. 1B in a table. Also, as far as possible, please add

the complete analysis of the metabolomics of the serum of the other mouse models as well (instead of just showing one selected metabolite). Finally, adding a graphical overview of the mouse models used to Figure 1 would be very helpful.

Minor comments

- (1) A main emerging activity of neutrophils to promote metastasis is the formation of NETs across many cancer types (PMID: 36827980). Could the authors assess if NET formation is impaired in UPP1-deficient neutrophils?
- (2) Please indicate the statistical tests used in the Figure legends

Referee #3:

REVIEW REPORT: Uridine Phosphorylase-1 supports metastasis of mammary cancer by altering immune and extracellular matrix landscapes of the lung

Summary

1. Does this manuscript report a single key finding? YES
- UPP1 promotes metastasis of breast cancer by altering immune landscapes and extracellular matrix composition in the lung.
2. Is the reported work of significance? YES
-The study describes a novel role for UPP1 in facilitating breast cancer metastasis, offering potential therapeutic targets.
3. Is it of general interest to the molecular biology community? YES
-This work is relevant to researchers in breast cancer biology, immunology, metabolism and extracellular matrix research because it links immune metabolic enzymes with metastatic processes.
4. Is the single major finding robustly documented using independent lines of experimental evidence? YES
-The study employs various experimental approaches, including genetically engineered mouse models, cell culture, metabolomics, and biochemical and immunological assays to support its claims.

Constructive criticism for the authors

General comments:

The manuscript titled "Uridine Phosphorylase-1 supports metastasis of mammary cancer by altering immune and extracellular matrix landscapes of the lung" is very well written and presents a thorough investigation into the role of UPP1 in mammary cancer metastasis. The study is significant and offers novel insights into how UPP1 influences neutrophil behavior, immune suppression, and extracellular matrix deposition in the lung. However, several areas require improvement, particularly in the clarity of presentation, data interpretation, and methodological details.

Major strengths:

- Novel insights into the role of UPP1 and neutrophil-derived uracil in breast cancer metastasis.
- Comprehensive experimental approach.

Major claims and significance:

- UPP1 promotes metastasis by altering immune landscapes and ECM composition.
- This claim is significant as it identifies UPP1 as a potential therapeutic target.

Novelty and convincing claims:

- The expression of UPP1 by neutrophils and its role in altering immune and ECM landscapes is novel.

Interest and audience:

- Researchers in cancer biology, immunology, metabolism and ECM research.
- The study links metabolic enzyme activity with metastatic processes, offering new therapeutic insights in cancer metastasis.

Standing out:

- The study stands out due to its multidisciplinary approach combining metabolomics and immunology.

Data quality:

- Data quality is high.

Abstract

A sentence highlighting the findings of the study in clinical samples could be included

Introduction:

The introduction provides a comprehensive background on metastasis and the significance of UPP1 in cancer progression. However, the section could benefit from a clearer statement of the study's objectives and hypotheses. The last paragraph of the introduction could be rephrased to reinforce the role of neutrophils in expressing UPP1 and producing uracil.

Materials and Methods:

While the methods are detailed, certain aspects require further clarification:

- The choice of specific mouse models and their relevance to the study should be justified more thoroughly.
- Details on how metabolites were quantified and validated should be expanded to ensure reproducibility. For example, the LC-MS profile and peaks of different metabolites identified in representative samples could be shown in supplementary material. Why only polar metabolites were analyzed?

Results:

- o Figure 1: Are pymt mice used for metabolomics littermates of FVB controls? Please, comment on material and methods section.
- o PyMt mice were culled when tumours reached 15mm. Did the mice reach this timepoint with the same age? How would you expect to anything correlate with tumour burden if this has been the criteria to uniformly cull the mice? If the age of mice was different when they reached the clinical endpoint, how do metastasis and uracil levels correlate with age?
- o Supp Fig1: Which is the cutoff selected for the fold change in the volcano plot? I can see a cutoff for p value in the graph but not for fold change. Are these 125 and 89 differentially expressed metabolites the ones ranked in Supp FIG 1B?
- o Has DT any effects in uracil and uridine levels in non-tumour bearing mice? DT on its own may be affecting the immune response of mice.
- o Figure 2B: statistical analysis is missing. Can you please also statistically compare UPP1 levels in Ly6G int and high neutrophils in veh and LPS treated conditions in Fig2B and Fig. S3D-E?
- o Statistical analysis is also missing in Fig 5F
- o Figure S3G. Number of neutrophils and metastatic burden are two parameters that are age-dependent? Were the mice same age at culling time point?
- o Figure 3: Do you observe different motility in neutrophils isolated from UPP wt and KO mice?
- o Fig4 A,B: Looks like the tumour itself does not affect the number of CD8 T cells in the lungs. Is then the increase observed with BAU and Upp1 Ko conditions an aberrant unnecessary increase? Same for CD45, neutrophils and CD4 T cells in fig S5B. Please, discuss.
- o Please discuss the result shown in S5B for neutrophils: Does the increase of lung neutrophils with BAU could be detrimental by promoting immune-suppression and leading to increased metastasis? What is the effect of BAU treatment in lung metastasis in MMTV-PyMT mice?
- o Please, discuss whether, even if integrin alpha 5 beta 1 happened to be recycled in a greater manner, how this will make a difference in activation, as no changes of activation have been shown in vitro.
- o Fig6B: Which was the reason to show primary tumour burden as a percentage of body weight? Which was the distribution of animal weights at the time of culling? Can you please show the graph of quantification of primary tumour burden without normalization?

Discussion:

The discussion addresses the study's findings and their implications effectively but there are some overstatements:

- The authors claim that UPP1 is a potential therapeutic target based on the data, but this claim is premature. More experimental validation is necessary to support this assertion. Please, include in the discussion.
- The discussion on the limitations of the study is brief and does not fully acknowledge the extent of these limitations. More detailed consideration of potential alternative effectors and explanations is needed.

General questions:

- UPP1 activity results in decrease of uracil and increase of two metabolites: uracil and ribose-1-phosphate. How do you know that the increase in uracil is the effector of the phenotype? Some controls for the effect of uridine or R1P could be added.
- Can you speculate on the role of blood levels of uridine and uracil in metastasis? Or do you think that UPP1 effects in metastasis are only relevant in the TME?
- Is UPP1 really generating uracil in tumours (in addition to isolated neutrophils? Have you quantified uridine and uracil in MMTV-PyMT WT and UPP1 KO tumours?
- The association between uracil, UPP1 and neutrophils should be further explored in clinical data.
- Fig1 F: In the metastatic BC patient's data (n=99), is there any correlation between uracil in plasma and BC subtype or metastasis in a particular organ? Is there any association with survival? Do uracil levels correlate with NLR in blood samples of these patients?
- Have you tried to stain UPP1 in metastatic BC samples? Is it expressed by cancer cells and neutrophils? Is there correlation between FN and UPP1 expression in clinical samples?
- As mentioned in the article, UPP1 expression in cancer cells has shown to be important for cancer progression. Can you discuss which is the relative expression of UPP1 in cancer cells and neutrophils in your models? Analysis of scRNAseq data of

MMTV-PYMT mice could help to clarify this.

- The association between UPP1 and metastasis shown in Figure 6 is weak and the decrease % of mice with lung metastasis cannot be attributed to UPP1 expression in neutrophils as UPP1 is depleted in the whole organism. Could you perform orthotopic injection of UPP1 WT cancer cells in Upp1 WT and KO animals and analyze tumour burden and metastasis to discern the role of UPP1 in tumour cells and the stroma?
- Increased FN deposition by uracil may be driven by UPP1 expression in cancer cells. How could you discriminate the role of neutrophils in FN deposition by CAFs? Fig 6 has to be corrected with maybe a question mark linking neutrophils and CAFs.
- Finally, do you have any data on what is driving UPP1 expression in neutrophils in the cancer context? You show that inflammation (LPS) promotes uracil secretion and UPP1 expression in neutrophils (Figure 2). Is there any inflammatory cytokine promoting UPP1 expression in neutrophils?

Minor comments:

- Last section of results: repeated word (this) in line 2.
- Fig4B: The graph shows the same trend of increased CD8 T cells in Upp1 KO mice in FVB mice than in PyMT mice. The significant p value in PyMT may be due to increased number of animals in these experimental groups. Can you please check this statistical analysis again? It might be a mistake.

Dr Cassie J Clarke
Garscube Estate
Switchback Road
Bearsden
Glasgow G61 1BD
United Kingdom

Tel: +44(0)141 330 8698

Email: c.clarke@crukscotlandinstitute.ac.uk

Tuesday 18th March 2025

RE: Revision of Whyte *et al.*, EMBOR-2024-59902V1

Dear Achim,

We are delighted to submit our revised version of “Uridine Phosphorylase-1 supports metastasis by altering immune and extracellular matrix landscapes” (EMBOR-2024-59902V1) for continued consideration at EMBO Reports. In the process of revising this paper the title has now changed to “Neutrophil derived uridine phosphorylase-1 supports metastasis by altering the lung microenvironment”.

Firstly, we would like to thank you for your understanding, and clear communication, regarding our timeline. This was influenced by the *in vivo* nature of the experiments that we felt were necessary to address reviewers' comments using the most appropriate methods, and an extended time required to deposit the large number of datasets now associated with this manuscript. Importantly, we feel that our manuscript has now been significantly strengthened by these additions, and we thank yourself and the reviewers for the opportunity to substantiate our finding in such a way. We have been able to address all comments, most of which are supported by providing new experimental data. A point-by-point response to all comments is amended to this letter. To aid re-review of the manuscript, we have highlighted new additions to the main manuscript by use of red text. I can confirm that no similar or related papers have been published since our original submission.

We have deposited all primary datasets generated during this study. To access to these datasets before publication, please use the following details: Metabolomic data generated at the CRUK Scotland Institute <https://massive.ucsd.edu/ProteoSAFe/dataset.jsp?task=4e350a90e51d494399860358dc4a2453> Username: MSV000097229_reviewer Password: Uridine. For metabolomic data generated at the University of California Irvine, the study is scheduled for release on 2025-03-21 as Study ID ST003759. Before this date, please use <https://dev.metabolomicsworkbench.org:2222/data/DRCCMetadata.php?Mode=Study&StudyID=ST003759&Access=YyhJ4908>. RNA-Seq datasets are deposited as GSE292202, GSE292203, GSE292205, and can be viewed using the private access tokens mvsrsgqwvbvqhfgv, gtcraqmclnwbjud and kryjwuoaxhsbfkn respectively.

We also attach an image for consideration for the journal cover – a metaphorical representation of our story. In this image the neutrophil is central, sitting in a sticky swamp (the metastatic niche). The fibroblasts are in the vicinity laying down fibronectin. Across the swamp T cells watch the scene but are unable to interfere. Cancer cells are growing in the forefront. We would be delighted if you considered this image for the cover of your journal.

To conclude, we thank yourself and the reviewers, for your time, consideration and constructive comments, and look forward to receiving your response following re-review.

Yours Sincerely,

Dr Cassie Clarke

We would like to thank all reviewers for their comments and suggestions. In addressing this feedback, we feel that we have now significantly strengthened the manuscript. A point-by-point response to each of the referees' comments is detailed below.

Referee 1:

We would like to thank the reviewer for their positive feedback on our manuscript which highlighted how interesting, well-written, and sound the article was, and that the work presented was substantial and solid for EMBO Reports.

Comments:

- The authors should clarify which aspects of the phenotype in the *Upp1*^{-/-} (or inhibited) mouse are attributable to neutrophils and which are not. For example, in the abstract, it states, "Uracil is generated by neutrophils expressing *Upp1*," which implies that neutrophils are the major source of uracil in the organism and are thus responsible for most of the phenotype. This is also depicted in the graphical abstract in Fig. 6D, which is centered on a neutrophil. I don't believe this is the case, since other immune cells, the diet and the tumors themselves generate high levels of uracil. In the absence of a neutrophil-specific knockout of *Upp1*, I would encourage the authors to either provide additional data, or strongly attenuate their abstract/conclusions by more clearly stating that they chose to focus on neutrophils and discuss that other cell lines from the myeloid lineage, the diet as well as the tumors are known to express high *Upp1* levels and were not investigated here.

We now provide the following new data to support our claim that neutrophils are a major source of uracil:

i) To assess tumours as a source of increased UPP1 we have assessed:

- *Upp1* in normal mammary tissue versus mammary tumour and show this is not altered (Figure EV1A)
- *Upp1* in primary mammary tumours that do and do not metastasise and show that this is not altered (Figure EV1B)
- *Upp1* in cell lines derived from tumours grown in the mammary fat pad versus cell lines derived from metastasis in the lungs and show that it is not altered (Figure EV1C)

ii) To determine whether dietary uracil can influence circulating uracil, we dosed mice by oral gavage daily with 10mM uracil for a total of 4 days. This is the maximum concentration of uracil possible for stable solubility in water. However, this protocol did not influence the levels of serum or tissue uracil in these mice (Reviewer Response Fig. R1):

iii) To understand if other immune cell types are a significant source of *Upp1* we have analysed scRNA-Seq data from the lung of MMTV-*PyMT* tumour bearing mice and show that neutrophils are the only immune cell type that upregulate *Upp1* in the presence of a tumour (Figure 2B&C; Appendix Figure S3).
iv) Finally, we have used an anti-Ly6G antibody approach to deplete circulating neutrophils (Fig. EV1L,M), and show that this significantly decreases levels of circulating uracil in MMTV-*PyMT* tumour-bearing mice (Figure 2G).

We feel that, collectively, these data sufficiently support the view that neutrophils are a significant source of circulating uracil.

2. Related to the above point, the authors justify focusing on neutrophils based on the CD11b-DTR results. CD11b is a pan-myeloid marker, and monocytes and macrophages. Those cells were recently highlighted for their high, immune-related *Upp1* expression (PMIDs 37198494, 37198474), and they will also be deleted upon expression of CD11b-DTR. Here too, the authors should better explain why they decided to focus on neutrophils and highlight the potential role played by other cells from the myeloid lineage, especially macrophages, and the fact that they were not investigated here.

We believe that the new scRNA-Seq data presented in Figure 2B&C and Appendix Figure S3 support our focus on neutrophils. Indeed, in PMID 37198494, macrophage depletion in did not alter tumour uracil, suggesting that while macrophages may indeed express some *Upp1*, they do not contribute significantly to levels of uracil in the extracellular microenvironment.

3. By which mechanism is *Upp1* upregulated in neutrophils?

We now present new data which show that a cytokine (GM-CSF), the circulating levels of which we find to be elevated in the presence of a primary tumour (Figure EV1G), can directly stimulate upregulation of *Upp1* in neutrophils (Figure EV1H).

4. The *Upp1*^{-/-} mouse model developed by the authors will certainly be an interesting tool to investigate uridine metabolism in vivo. It should be noted that a *Upp1*^{-/-} line was previously generated (PMID 15772079 - perhaps this should be cited?), but no immune phenotype was reported. Could the authors better characterize the immune system of the *Upp1*^{-/-} mouse, even globally? Besides the lung, are the global numbers of circulating immune cells similar when compared to +/+ mice? Is the deposition of fibronectin in the lungs of *Upp1*^{-/-} mice also decreased in healthy animals? Some figures, such as Fig. 3B, 4A, 5A/B and S5B, lack a control in healthy animals.

We have now cited PMID15772079 at the point in which we introduce our *Upp1*^{-/-} mouse model.

To better characterise the immune system of *Upp1*^{-/-} globally, we have now determined the numbers of immune cell types in the spleen and lymph nodes of *Upp1*^{+/+} and *Upp1*^{-/-} mice, in the presence and absence of a tumour (Appendix Figure S7).

We have now assessed lung fibronectin deposition in the lungs of healthy mice (age-matched to those presented in the manuscript) that are *Upp1*^{+/+} and *Upp1*^{-/-} and find that this is not altered (Figure EV4A). Given that neutrophils express negligible amounts of *Upp1* in the absence of a tumour, this suggests that the UPP1 influence is specific to the increased deposition of fibronectin stimulated by the presence of a tumour, rather than basal levels under normal conditions.

We have addressed the concerns regarding appropriate controls as follows:

- We have now strengthened the data to demonstrate conclusively that treatment of non-tumour-bearing mice with BAU for 12 days does not alter neutrophil migration in the lung (new Appendix Figure S6C). This confirms that UPP inhibition only influences lung neutrophil speed in tumour-bearing mice. We also demonstrate conclusively that treatment of non-tumour-bearing mice for BAU for 12 days does not alter immune populations in the lung (Fig. EV2B).
 - We have now assessed lung fibronectin deposition in the lungs of non-tumour-bearing mice that are *Upp1*^{+/+} and *Upp1*^{-/-} (age-matched to the MMTV-PyMT clinical endpoint cohort) and find that this is not altered. These data are now presented in a new Figure EV4A.
5. Could the authors clarify why there are less metastasis in *Upp1*^{-/-} mice? Is it due to the inability of the cancer to form metastasis (as was reported for primary pancreatic cancers growth) - or it is due to the immune system? What would happen in a CD11b-DTR model (similar experiment as in S3C but counting metastasis burden)?

In the proportion of *Upp1*^{-/-} mice that do develop metastasis, there is no difference in the total number of metastases, or the overall burden of metastasis, in the lungs of those animals, suggesting that the lack of *Upp1* does not hinder their growth if they do indeed get the opportunity to establish (Figure EV5C&D). Indeed, the significant (~30%) reduction in metastatic penetrance that we see in the MMTV-PyMT *Upp1*^{-/-} cohort reflects change that might be expected when interfering with a mechanism that involves priming the pre-metastatic niche, and so we believe that our data supports the role of the immune system and pre-metastatic niche priming of *Upp1* influenced metastasis.

Regarding metastasis in the CD11b-DTR model, the tumour models used are too aggressive to study metastasis, as the mice succumb to primary tumours before metastasis are detectable. Furthermore, myeloid cells can only be depleted for 2 weeks in this model, as mice start to suffer adverse effects beyond a 10-day period. So, as this model can only be used to assess acute responses, it is not appropriate for long-term studies.

6. Based on the authors observation, does *Upp2* play any role in systemic uridine (immuno)metabolism?

Whilst BAU inhibits both UPP1 and UPP2, our *Upp1*^{-/-} mouse is specific to the *Upp1* isoform. Given the striking depletion of uracil in tissues and serum from *Upp1*^{-/-} mice (Appendix Figure S2), this suggests that UPP1 is the predominant isoform in systemic uridine metabolism. This view is reinforced by analysis by scRNA-Seq which indicate low levels of expression of *Upp2* in this dataset (Appendix Figure S4) and further verified by our inability to detect *Upp2* by qPCR in neutrophils *ex vivo*.

Referee 2:

We thank the reviewer for highlighting that our manuscript is interesting and novel. We also thank the reviewer for their comments and suggestions. We have addressed these points below:

Comments:

Main comment 1 - It is unclear if neutrophil-derived Uracil / neutrophil-intrinsic loss of UPP1 is relevant for cancer progression/metastasis?

As mentioned above, the authors do report cell-autonomous effects of UPP1-deficient neutrophils in culture, but the *in vivo* relevance of such activities remains poorly investigated due to the use of a full body UPP1 knockout mouse model. Hence, *in vivo*, UPP1 expressed by cancer cells (as previously described), T cells and fibroblast themselves could account for observed effects. Also, it is not clear if neutrophil-expressed UPP1 is relevant for systemic Uracil concentrations. Ideally, the best experiment would be using mice with a neutrophil-specific deletion of UPP1, however I appreciate that this would be

challenging. Hence, to address those important open questions, I suggest the following experimental approaches:

1. Perform experiments as in Supp Fig. 3B-C (measuring systemic Uracil) by specifically deleting neutrophils using the anti-Ly6G antibody (for example as in PMID: 26649828). Quantify Upp1 expression in CD11b⁺ cells in the lung (including macrophages). CD11b also affects macrophages, which is a very prevalent cell type. Those experiment will determine the importance of neutrophil-derived uracil to systemic levels.

We have depleted neutrophils in MMTV-*PyMT* tumour-bearing mice using an antibody recognising Ly6G and found that this leads to significant reduction in the levels of circulating uracil. These data are now shown in new Figure 2G.

We have analysed scRNA-Seq data from the lungs of FVB/N versus MMTV-*PyMT* tumour-bearing mice and show that *Upp1* is specifically upregulated in the neutrophils, but not in other immune cell types found in the lung (Figure 2B). This is further reinforced by an additional scRNA-Seq dataset derived from the spleen (Figure 2C; Appendix Figure S3A). With specific reference to macrophages, we also highlight PMID 37198494, where the authors depleted macrophages in tumour bearing mice, but this did not alter levels of tumour uracil.

2. Measure systemic Uracil levels (as in Supp. Fig. 3B-C) and metastatic progression (as in Figure 6) in control and UPP1-ko FVB mice transplanted with WT tumours and quantify UPP1 expression in MMTV-*PyMT* tumour cells. This will uncover the contribution of UPP1 expressed by tumour cells to Uracil accumulation and metastatic progression.

Transplant of KP tumour fragments into FVB/N recipient mice offers the opportunity to assess very early changes in neutrophil motility in the pre-metastatic lung. This approach and early timepoint does not stimulate the same extent of neutrophil driven by *PyMT* and thus enables us to dissect the contribution of tumour UPP1 from the neutrophil/metastasis associations seen in the MMTV-*PyMT* model. We have performed these experiments and find that transplant of KP tumour fragments into FVB/N *Upp1*^{+/+} and *Upp1*^{-/-} recipients is not sufficient to increase circulating uracil levels (Reviewer Response Fig. R2).

We have also performed RNA-Seq on tissue and tumour extracts and found that levels of the mRNA encoding *Upp1* do not differ between normal mammary tissue and MMTV-*PyMT* mammary tumours (Figure EV1A). Furthermore, qPCR analysis indicates that levels of mRNA encoding *Upp1* also do not differ between MMTV-*PyMT* mammary tumours from mice that had no lung metastases or those that displayed moderate (1-10 metastases) or high (>10 metastases) levels of metastasis (Figure EV1B). We also find that *Upp1* expression in the primary tumour does not correlate with levels of circulating uracil, and these data are shown below (Reviewer Response Fig. R3).

Reviewer Response Figure R3: Tumour *Upp1* levels do not correlate with circulating uracil. *Upp1* in MMTV-*PyMT* tumours was assessed by qPCR, and correlations with serum assessed by Spearman correlation. Each dot represents an individual mouse, $n=29$ total. Black dots represent mice with no metastasis, and pink and green dots represent mice with 1 – 10 or >10 metastases respectively. qPCR samples were normalised using *Atcb* as an endogenous control, and one mouse from the no metastasis group was used to calculate relative fold change across all samples.

3. Measure UPP1 expression in lung fibroblasts and lung CD8+ T cells in control and MMTV-*PyMT* mice. Analyse Fibronectin accumulation (as in Fig.5A) and CD8+ T cell numbers (as in Fig.4A-C) in the lung of Neutrophil-depleted (with anti-Ly6G antibody) WT and UPP1-ko MMTV-*PyMT* mice. If UPP1-expressing neutrophils ultimately cause fibronectin deposition or T cell inhibition *in vivo*, total neutrophil depletion should have an effect in WT mic but not UPP1-ko mice.

We have analysed scRNA-Seq data from the lungs of FVB/N versus MMTV-*PyMT* tumour-bearing mice and show that *Upp1* is specifically upregulated in the neutrophils alone and not in other immune cell types within the lung (Figure 2B).

To experimentally address this point, we depleted neutrophils using the Anti-Ly6G (1A8) antibody depletion method (as described in PMID 25822788). To achieve this in a manner which standardised treatment start times and resulted in a treatment window of 2 – 3 weeks, MMTV-*PyMT* mice were treated with either control IgG, or anti-Ly6G at a time when the largest mammary tumour measured 5mm in diameter. Importantly, this is the limit of what we may perform ethically and in accordance with our Home Office licence regulations as, in our hands, FVB/N mice have limited tolerance to neutrophil depletion. Whilst depletion of neutrophils for this period was sufficient to deplete neutrophils (Figure EV1L-M) and decrease circulating uracil levels (Figure 2G), it was insufficient to decrease the proportion of mice that presented with metastasis (Appendix Figure S8). Thus, as we are unable to deplete neutrophils for long enough to effect metastasis in this cohort of mice, and we do not currently have a neutrophil-specific *Upp1* knockout mouse, the ethical and technical limitations of the experimental tools available to us have hindered our ability to experimentally address this point. We have added text to the discussion to highlight this limitation of our study.

Main comment 2 - are the effects of neutrophil UPP1 specific for altering the lung microenvironment (and metastasis to the lung) in breast cancer? Or across organs/cancer types?

The authors report high systemic Uracil levels in several mouse models of metastatic cancer. Yet, they focus their characterization of neutrophils entirely on the lung of the MMTV-PyMT cancer model. To understand the relevance of neutrophil-derived UPP1 in the metastatic progress of the other used cancer models and metastasis to a second site, I strongly recommend the following experiments: Analysis of progression (especially lung and liver metastasis) in KPC (PDAC model) and KPN (CRC model) upon treatment with BAU (alternatively UPP1-deficiency, but that would require additional crossing of mice and could be avoided using the inhibitor) and neutrophil depletion.

Whilst we appreciate that conducting experiments with two additional genetically engineered mouse models of cancer would be valuable to understand the role of *Upp1* in the specificity and tropism of metastasis, we feel that the time and resources required for this experimentally are beyond the scope of this manuscript. We have thus focused attention on one of these suggestions; namely determining the consequences of *Upp1* knockout in tumour progression and metastasis in the KPC mouse model of metastatic pancreatic cancer. We find that *Upp1* knockout does not significantly influence overall survival or micrometastases to the liver, lung or diaphragm in KPC mice (Figure EV5E-F). However, loss of *Upp1* does result in a significant decrease in local invasion of PDAC (Figure EV5G-H). We have added text to the manuscript to discuss these results.

Main comment 3 - Integrated metabolomics analysis of the serum of mouse cancer/metastasis models is poorly reported. Right now, the analyses and data not very clearly depicted in Figure 1 and Supp Figure 1. Please, for the sake of open science, indicate all identified metabolites (16 + 17 + 5 + 19) in the analysis Supp. Fig. 1B in a table. Also, as far as possible, please add the complete analysis of the metabolomics of the serum of the other mouse models as well (instead of just showing one selected metabolite). Finally, adding a graphical overview of the mouse models used to Figure 1 would be very helpful.

As this was an untargeted metabolomic analysis, we do not have positive identifications for the majority of compounds and so, instead, have presented mass and retention time for each of these metabolites observed. The names of metabolites that we had definitively confirmed by comparison with a commercially obtained standard are provided. These data were provided in the original manuscript as Supplementary Table 1. However, we have now edited the text and figure legends to make this clearer, and the data are now included in Appendix Table S1. Moreover, these metabolomic data, including the associated metadata, are deposited and accessible in the Mass Spectrometry Interactive Virtual Environment (MassIVE) (Study ID MSV000097229) enabling all researchers to access all data generated. A schematic describing all mouse models used is also now available in Appendix Figure S1A.

Minor comments

- (1) A main emerging activity of neutrophils to promote metastasis is the formation of NETs across many cancer types (PMID: 36827980). Could the authors assess if NET formation is impaired in UPP1-deficient neutrophils?

We have performed immunohistochemistry for neutrophil elastase (NE), myeloperoxidase (MPO), and citrullinated Histone H3 (H3cit) in sections from the lungs of MMTV-PyMT *Upp1*^{+/+} and *Upp1*^{-/-} mice and find no significant difference in expression of these markers of NETosis between tumour-bearing *Upp1*^{+/+} and *Upp1*^{-/-} mice, as shown below (Reviewer Response Fig. R4):

(2) Please indicate the statistical tests used in the Figure legends

All statistical tests are now presented in figure legends.

Referee 3:

We thank this reviewer for their supportive review, including recognising that the work is novel and significant, of interest to the community, and robustly documented using independent lines of experimental evidence.

Comments:

1. Abstract: A sentence highlighting the findings of the study in clinical samples could be included

We have now modified the abstract to highlight our result showing that uracil is increased in the serum of patients with metastatic breast cancer.

2. Introduction: The introduction provides a comprehensive background on metastasis and the significance of UPP1 in cancer progression. However, the section could benefit from a clearer statement of the study's objectives and hypotheses. The last paragraph of the introduction could be rephrased to reinforce the role of neutrophils in expressing UPP1 and producing uracil.

Additional phrases have now been added the introduction to state the aim of the study, and to reinforce the role of neutrophils in expressing UPP1 and producing uracil.

3. Materials and Methods:

While the methods are detailed, certain aspects require further clarification:

- The choice of specific mouse models and their relevance to the study should be justified more thoroughly.

We have specifically focused on mouse models of cancer where the disease is well-established to be highly metastatic. The text has been edited to outline this information.

- Details on how metabolites were quantified and validated should be expanded to ensure reproducibility. For example, the LC-MS profile and peaks of different metabolites identified in representative samples could be shown in supplementary material. Why only polar metabolites were analyzed?

Compound identification was assigned by matching the mass and retention time of observed peaks to an in-house library generated using metabolite standards (mass tolerance of 5ppm and retention time tolerance of 0.5min), and this information is described fully in the methods. All data have been deposited in the Mass Spectrometry Interactive Virtual Environment (MassIVE) (Study ID MSV000097229), enabling the community to assess all LC-MS profiles and peaks as required. Given our interest in characterising the metabolic landscape of metastasis, we chose to focus our metabolite extractions on polar metabolites, which includes amino acids, nucleic acids, sugars and small organic acids, as these encompass a large proportion of well-characterised metabolic pathways that are known to be dysregulated in cancer.

4. Results:

o Figure 1: Are pymt mice used for metabolomics littermates of FVB controls? Please, comment on material and methods section.

Yes, this is correct, and a sentence has now been added to the materials and methods section as requested.

o PyMt mice were culled when tumours reached 15mm....
...Did the mice reach this timepoint with the same age?

Yes - no significant difference in age at clinical endpoint what is shown by the survival curve (Figure 6A).

How would you expect to anything correlate with tumour burden if this has been the criteria to uniformly cull the mice?

In the MMTV-PyMT model of mammary cancer the mice can have multiple tumours, as tumours can form in any of the 10 mammary glands. Mice reach clinical endpoint when one tumour reaches 15mm in diameter. Thus, a mouse with two 10mm and one 15mm tumour will have a larger overall primary tumour burden than a mouse that has just one 15mm tumour alone.

If the age of mice was different when they reached the clinical endpoint, how do metastasis and uracil levels correlate with age?

We have measured circulating uracil and found that this does not significantly differ between mice of different ages (i.e. when comparing mice younger and older than 90 days) (Reviewer Response Fig. R5, left panel). Furthermore, the mean age of mice that were sampled in the poorly metastatic (and lower uracil) KPC^{flC} and highly metastatic (and higher uracil) KPC cohorts does not differ (Reviewer Response Fig. R5, right panel), reinforcing that age differences are not responsible for the observed changes in metastasis-associated circulating uracil.

Reviewer Response Figure R5: Serum uracil does not correlate with age. Left graph: Serum uracil in FVB/N mice younger and older than 90 days, n=3 mice <90 days, and n=6 mice >90 days old. Right graph: Age of mice taken at palpable pancreatic tumour that were either KP^{fl}C (n=10 mice, lower serum uracil levels as per Figure 1E) or KPC (n=6 mice, higher serum uracil levels as per Figure 1E), mean ± SEM, unpaired t-test.

o Supp Fig1: Which is the cutoff selected for the fold change in the volcano plot? I can see a cutoff for p value in the graph but not for fold change. Are these 125 and 89 differentially expressed metabolites the ones ranked in Supp FIG 1B?

Red values have a $-\log_{10}$ adj. p-value greater than 1.3. Fold change was not included as a factor. The 125 (increased) and 89 (decreased) differentially expressed metabolites are those detected at significantly different levels when comparing serum from MMTV-PyMT tumour bearing mice with their FVB/N littermate controls. The table in original Figure Supp Fig 1B (now Appendix Figure S1C) is our selective assessment of metabolites that significantly correlated with metastasis. We have edited the text to make this distinction clearer. Further details of the 125 and 89 differentially expressed between MMTV-PyMT tumour bearing mice and FVB/N littermate controls are available via (Study ID MSV000097229).

o Has DT any effects in uracil and uridine levels in non-tumour bearing mice? DT on its own may be affecting the immune response of mice.

The lack of effect of DT on wild-type mice has been well characterised previously (PMID: 27402485), and subsequent experiments, presented throughout our manuscript paper address the specificity of the immune and metabolic changes associated with neutrophil mobilisation in much more detail.

o Figure 2B: statistical analysis is missing. Can you please also statistically compare UPP1 levels in Ly6G int and high neutrophils in veh and LPS treated conditions in Fig2B and Fig. S3D-E?

Many thanks for highlighting this omission. The statistical analysis in Fig. 2B was accidentally removed during formatting and is now included. The requested additional statistics have also been performed and are now included.

o Statistical analysis is also missing in Fig 5F

Many thanks for highlighting this omission. This statistical analysis was accidentally removed during formatting and is now included.

o Figure S3G. Number of neutrophils and metastatic burden are two parameters that are age-dependent? Were the mice same age at culling time point?

These mice were all in the narrow age-range of 12 – 14 weeks and had reached clinical endpoint (one tumour measuring 15mm in diameter) when these measurements were made. However, we have interrogated the data to determine if there is a relationship between metastatic burden or circulating neutrophils and age. This indicated that there is no significant correlation between age and number of metastases, or between age and neutrophil count (Reviewer Response Fig. R6).

o Figure 3: Do you observe different motility in neutrophils isolated from UPP wt and KO mice?

The motility of neutrophils from *Upp1*^{+/+} and *Upp1*^{-/-} mice has now been assessed and is shown in Appendix Figure S6D. This indicates that the motility of neutrophils does not differ between *Upp1*^{+/+} and *Upp1*^{-/-} mice if they are not tumour-bearing.

o Fig4 A,B: Looks like the tumour itself does not affect the number of CD8 T cells in the lungs. Is then the increase observed with BAU and *Upp1* Ko conditions an aberrant unnecessary increase? Same for CD45, neutrophils and CD4 T cells in fig S5B. Please, discuss.

Previously published work has shown that the immunosuppressive capability of neutrophils in lungs of mice with metastatic mammary cancer are evoked through influencing the CD8⁺ T cell effector phenotype (number of CD62L⁻ CD44⁺ IFN- γ ⁺ cells), and not the total number of CD8⁺ T cells (Coffelt et al., 2015). Our data are in line with this, wherein the absence of *Upp1* there is a trend of increasing T cell numbers (which reaches significance with the combination of tumour plus inhibition (or knockout) of UPP1), but the most striking effect is seen when assessing the effector function of the T cells, namely the number of IL2⁺, and actively proliferating, CD8⁺ T cells.

o Please discuss the result shown in S5B for neutrophils: Does the increase of lung neutrophils with BAU could be detrimental by promoting immune-suppression and leading to increased metastasis? What is the effect of BAU treatment in lung metastasis in MMTV-*PyMT* mice?

Our data suggests that UPP1 influences the immunosuppressive function of neutrophils (Figure 4D-F). Thus, our argument is that, even in the presence of increased neutrophil numbers, inhibiting UPP1 would inhibit their immunosuppressive capabilities and thus decrease metastasis.

o Please, discuss whether, even if integrin alpha 5 beta 1 happened to be recycled in a greater manner, how this will make a difference in activation, as no changes of activation have been shown in vitro.

The antibody used in (Figure 5E), 9EG7, is specific to the active conformation of integrin β_1 . The antibody used to measure integrin recycling in (new) figure EV4I recognises $\alpha_5\beta_1$ integrin in both the active and inactive conformations. Thus, addition of uracil to fibroblasts increases the recycling of $\alpha_5\beta_1$ whether it be in the active or inactive conformations and without, necessarily, altering the proportion of active to inactive heterodimer. We have edited the text to explain this further.

o Fig6B: Which was the reason to show primary tumour burden as a percentage of body weight? Which was the distribution of animal weights at the time of culling? Can you please show the graph of quantification of primary tumour burden without normalization?

As standard practice, we report primary tumour burden as a percentage of body weight as this corrects for differences in mouse size. Mouse and primary tumour weights are plotted below and are not different in the presence and absence of *Upp1* (Reviewer Response Fig. R7).

Reviewer Response Figure R7: Mouse weights and tumour weights are unaltered in the presence and absence of *Upp1*. Total mouse weight, and weight of dissected primary mammary tumours, was assessed at clinical endpoint in MMTV-PyMT *Upp1*^{+/+} (n=20) and MMTV-PyMT *Upp1*^{-/-} mice (n=17), mean \pm SEM, unpaired t-test.

5. Discussion:

The discussion addresses the study's findings and their implications effectively but there are some overstatements:

The authors claim that UPP1 is a potential therapeutic target based on the data, but this claim is premature. More experimental validation is necessary to support this assertion. Please, include in the discussion.

We agree and have moderated our claim to highlight that further study is necessary to substantiate this postulate.

The discussion on the limitations of the study is brief and does not fully acknowledge the extent of these limitations. More detailed consideration of potential alternative effectors and explanations is needed.

We feel that the additional data, regarding the lack of effect of cancer cell specific or stromal cell *Upp1* (stromal cell contribution is addressed in response to comments below), that we have provided have reinforced the view that neutrophils are the effector in the studies we have described. We have added text to the discussion to outline more clearly the limitations of our approaches and conclusions.

6. General questions:

-UPP1 activity results in decrease of uracil and increase of two metabolites: uracil and ribose-1-phosphate. How do you know that the increase in uracil is the effector of the phenotype? Some controls for the effect of uridine or R1P could be added.

Increased UPP1 activity can lead to depletion of uridine. However, uridine can also be synthesised *de novo*. In mouse models of metastatic cancer, we do not see decreased circulating uridine levels, and this is likely due to its *de novo* production. We appreciate that there may be altered local levels of uridine in areas of high UPP1 activity. We have therefore assessed T cell proliferation in a range of exogenously added uridine concentrations. This indicates that *ex vivo* T cell proliferation is unchanged by supplementation of uridine (or uracil, for that matter) (Figure EV3D).

The LC-MS techniques that we have used in this study are unable to distinguish Ribose 1-phosphate from Ribose 5-phosphate, xylulose 5-phosphate, and ribulose 5-phosphate, as they are all 5-carbon sugars with the same molecular mass. Consequently, we integrate an LC-MS peak, comprising 5 maxima (of identical molecular mass). We term this complex peak 'pentose phosphate' and have found that this is not altered following loss of *Upp1* (Reviewer Response Fig. R8; left panel) or following depletion of neutrophils (Reviewer Response Fig. R8; right panel). Whilst published work has indeed shown that ribose derived from uridine can support tumour growth, this is a phenotype reported to occur only in the absence of glucose. Because neutrophils will not tolerate *ex vivo* glucose depletion, we have been unable to determine the contribution made by UPP1-mediated cleavage of uridine to 5-carbon metabolism.

Reviewer Response Figure R8: Pentose phosphate is not altered in the absence of *Upp1* or in neutrophil depleted conditions. (Left panel) Pentose phosphate was assessed by LC-MS in mammary tumours from MMTV-PyMT mice, *Upp1*^{+/+} n=20, *Upp1*^{-/-} n=17. (Right panel) Pentose phosphate was assessed by LC-MS in serum of MMTV-PyMT mice treated with IgG control or anti-Ly6G to deplete neutrophils, n=9 IgG control, n=8 anti-Ly6G, mean ± SEM, unpaired t-test.

Can you speculate on the role of blood levels of uridine and uracil in metastasis? Or do you think that UPP1 effects in metastasis are only relevant in the TME?

Altered blood levels of uracil and uridine provide the opportunity for circulating metabolites to influence metabolite landscapes in distant tissues and thus facilitate non-cell autonomous effects at those sites. Whilst our work here has focused on the role of neutrophil UPP1 metabolism in metastasis, neutrophilia is also a feature of early disease, and of disease associated with elevated cancer risk, thus we are now pursuing a programme of work aimed at determining whether UPP1 can influence microenvironment in a way that contributes to early pre-cancerous changes.

Is UPP1 really generating uracil in tumours (in addition to isolated neutrophils? Have you quantified uridine and uracil in MMTV-PyMT WT and UPP1 KO tumours?

We found that *Upp1* levels do not alter between normal mammary tissue and mammary tumours (Figure EV1A). We have also found that *Upp1* levels do not differ between primary mammary tumours that do and do not metastasise (Figure EV1B). We therefore do not feel that tumour specific UPP1 is playing a role in this process. To address the second point, we have assessed uridine and uracil in tumours from in MMTV-PyMT *Upp1*^{+/+} and *Upp1*^{-/-} mice and now present this data in Appendix Figure S2C.

The association between uracil, UPP1 and neutrophils should be further explored in clinical data.

Given that the mechanism we describe in mammary cancer is specific to the metastatic site (the lungs in our case), we have been unable to source lung samples and/or data to substantiate this effect further in human patients. The only source of human patient samples we can find in this regard are those from the PEACE (Posthumous Evaluation of Advanced Cancer Environment) study. Whilst we have had productive conversations with the PEACE study leader, there are insufficient breast cancer cases within this study to be able to answer this question.

Fig1 F: In the metastatic BC patient's data (n=99), is there any correlation between uracil in plasma and BC subtype or metastasis in a particular organ? Is there any association with survival? Do uracil levels correlate with NLR in blood samples of these patients?

The n=99 breast cancer patients whose blood samples are represented in Fig. 1F all had confirmed metastatic disease. We have further interrogated these data with regard to cancer diagnoses/pathology and are unable to find any correlations between stage of disease, hormone receptor status, or organ of metastasis (Reviewer Response Fig. R9). We do not have survival data associated with this cohort, nor do we have NLR from matched blood samples.

Have you tried to stain UPP1 in metastatic BC samples? Is it expressed by cancer cells and neutrophils? Is there correlation between FN and UPP1 expression in clinical samples?

There are no antibodies commercially available that work well and specifically for UPP1, in our hands, in formalin fixed and paraffin embedded human tissues (a previously published antibody that was reported has now been discontinued). We also find that our effects are specific to the metastatic target organ of the lung, not the primary tumour, and this again limits our ability to source appropriate human tissue of interest, and/or published data, to address these questions.

- As mentioned in the article, UPP1 expression in cancer cells has shown to be important for cancer progression. Can you discuss which is the relative expression of UPP1 in cancer cells and neutrophils in your models? Analysis of scRNAseq data of MMTV-PYMT mice could help to clarify this.

We have assessed *Upp1* levels and found that this is not altered between normal mammary tissue and mammary tumours (Figure EV1A). We have also found that *Upp1* levels do not differ between primary mammary tumours that do and do not metastasise (Figure EV1B). Furthermore, we show by analysis of scRNA-Seq that neutrophils are the only immune cell type that display increased *Upp1* expression in the lungs of tumour-bearing mice (Figure 2B&C) and, importantly, that depletion of neutrophils leads to reduced circulating uracil levels (Figure 2G).

- The association between UPP1 and metastasis shown in Figure 6 is weak and the decrease % of mice with lung metastasis cannot be attributed to UPP1 expression in neutrophils as UPP1 is depleted in the whole organism. Could you perform orthotopic injection of UPP1 WT cancer cells in *Upp1* WT and KO animals and analyze tumour burden and metastasis to discern the role of UPP1 in tumour cells and the stroma?

The ~30% reduction in metastatic penetrance that we see in the MMTV-PyMT *Upp1*^{-/-} cohort is reflective of the level of change that would be expected when interfering with a mechanism that involves priming the pre-metastatic niche. Thus, the *Upp1*^{-/-} mice that do present with metastasis may do so due to late dissemination that is less dependent on metastatic niche priming. To address this experimentally we would need to use a transplant and resection model, but this would take 10 – 12 months in total and would not be guaranteed to work due to frequent re-growth of primary tumours following resection. Instead, we have addressed the contribute to cancer cell specific UPP1 as described above, and we present data regarding the stromal contribution in our response to comments below.

- Increased FN deposition by uracil may be driven by UPP1 expression in cancer cells. How could you discriminate the role of neutrophils in FN deposition by CAFs? Fig 6 has to be corrected with maybe a question mark linking neutrophils and CAFs.

We have addressed the contribution of cancer cell UPP1 to tumour progression in previous responses. With regard to the contribution of stromal UPP1, we have interrogated published proteomic data available from normal human fibroblasts (NF) and cancer associated fibroblasts (CAFs). In PMID: 24025712, UPP1 was not identified in the global list of proteins identified or differentially expressed in whole cellular extracts from CAFs. In PMID: 29593339, UPP1 was not identified as a differentially expressed protein between NFs and CAFs. In PMID: 32984808, UPP1 was downregulated in CAFs in comparison to NFs (Reviewer Response Fig. R10).

We have also decreased *Upp1* expression in fibroblasts using CRISPR technology, and performed experiments as described in Figure 5D, to dissect differences between cell intrinsic *Upp1* expression and exogenously added uracil and find that it is the exogenous uracil which encourages fibroblasts to deposit a pro-migratory extracellular matrix (Reviewer Response Fig. R11). Thus, to conclude, although fibroblasts can express UPP1 we do not have any evidence that this could contribute to the metastasis-associated phenotypes described in our manuscript.

Reviewer Response Figure R11: The promigratory matrix deposited by fibroblasts is dependent on exogenous uracil and not endogenous *Upp1*. Cellular derived matrices were deposited by fibroblasts expressing an empty vector, or *Upp1* Crispr construct that resulted in a confirmed knockdown of *Upp1* expression, in the presence and absence of 30µM uracil. Fibroblasts were removed, and the ability of the deposited matrix to support invasive *PyMT*⁺ cancer cell behaviour was assessed by timelapse microscopy. Dots represent the protrusion length of *PyMT*⁺ cells, n=30 measured per experiment, for two independent matrix depositions (n=60 cells total). Data presented are mean ± SD, one-way ANOVA Šídák's multiple comparisons test.

- Finally, do you have any data on what is driving UPP1 expression in neutrophils in the cancer context? You show that inflammation (LPS) promotes uracil secretion and UPP1 expression in neutrophils (Figure 2). Is there any inflammatory cytokine promoting UPP1 expression in neutrophils?

We now present new data which show that a cytokine (GM-CSF), the circulating levels of which we find to be elevated in the presence of a primary tumour (Figure EV1G), can directly stimulate upregulation of *Upp1* in neutrophils (Figure EV1H).

Minor comments:

- Last section of results: repeated word (this) in line 2.

Many thanks, this error has now been corrected.

- Fig4B: The graph shows the same trend of increased CD8 T cells in *Upp1* KO mice in FVB mice than in *PyMT* mice. The significant p value in *PyMT* may be due to increased number of animals in these experimental groups. Can you please check this statistical analysis again? It might be a mistake.

We thank the reviewer for highlighting this point. Upon checking the statistical analysis for this figure, we can confirm that it was in fact a Kruskal-Wallis test that was used, as opposed to a Šídák's multiple comparisons test. This was because, upon testing, data within the MMTV-*PyMT Upp1*^{-/-} group was shown not to be normally distributed. The Kruskal-Wallis test is thus the most appropriate test to use. We have now, edited the figure legends, and materials and methods, to explicitly describe all the statistical tests used throughout the manuscript.

Dear Dr. Clarke,

Thank you for the submission of your revised manuscript to our editorial offices. I have already forwarded the reports from the 3 referees that I asked to re-evaluate your study, you will find again below. I also have received your provisional point-by-point-response (further revision plan), addressing the remaining concerns of referee #2. After looking through this, I decided to invite a final revised manuscript that addresses the remaining referee points as indicated in your final revision plan. I would suggest to add the data shown ("Reviewer Response 2; Figure 1") to the manuscript (see also below). Please also provide a detailed final point-by-point-response to the remaining referee points.

- I noted there is a large number of co-authors. Please make sure that each co-author meets all the criteria indicated in our author guidelines. See:

<https://www.embopress.org/page/journal/14693178/authorguide#authorshippinguidelines>

- There are author name discrepancies. It is Rene Jackstadt in the manuscript text file but Rene-Filip Jackstadt in the submission system, and Edward W. Roberts in the manuscript but Ed Roberts in the submission system. Please check.

- Please order the manuscript sections like this, using these names:

Title page - Abstract - Keywords - Introduction - Results - Discussion - Methods - Data availability section - Acknowledgements - Disclosure and Competing Interests Statement - References - Figure legends - Expanded View Figure legends

- As indicated by referee #2 and also above, it would make sense to include the data shown in the p-b-p-response letters into the manuscript. Please do this. We could also accommodate another main figure or an additional EV figure. Please arrange the figures (main, EV and Appendix figures) to include the additional data. Please note that new data that becomes part of a main figures needs to be accompanied by source data (see also below).

- Please make sure that all the funding information is also entered into the online submission system and that it is complete and similar to the one in the acknowledgement section of the manuscript text file. Presently, the grant 'Bloodwise (15041)' and the 'Lord Kelvin/Adam Smith (LKAS) Leadership Fellowship from the University of Glasgow ' are missing from the submission system. Please check.

- Please make sure that all figure panels or tables (main, EV and Appendix figures) are called out separately and sequentially. Presently, there are no callouts for panel 6C and for Appendix Figure S8. Moreover, the callout for Table S7 is missing the word 'Appendix'. Please check. Finally, there are callouts to Supplementary Figures in the methods section. Please update these callouts (Figure EVx or Appendix Figure Sx).

- Please add scale bars of similar style and thickness to microscopic images (main, EV and Appendix images), using clearly visible black or white bars (depending on the background). Please place these in the lower right corner of the images themselves. Please do not write on or near the bars in the image but define the size in the respective figure legend. Presently, some scale bars are too thin or have a different style (EV4B). Please check.

- Please check again that the number "n" for how many independent experiments were performed, their nature (biological versus technical replicates), the bars and error bars (e.g. SEM, SD) and the test used to calculate p-values is indicated in the respective figure legends. Please also check that all the p-values are explained in the legend, and that these fit to those shown in the figure. Please provide statistical testing where applicable. Please avoid the phrase 'independent experiment', but clearly state if these were biological or technical replicates. Please also indicate (e.g. with n.s.) if testing was performed, but the differences are not significant. In case n=2, please show the data as separate datapoints without error bars and statistics. See also:

<http://www.embopress.org/page/journal/14693178/authorguide#statisticalanalysis>

If n<5, please show single datapoints for diagrams. Presently, many diagrams have only partial statistics or 'ns' is missing. Moreover:

- Please note that information related to n is missing in the legends of figures 2B, EV1 M.

- Please note that the error bars are not defined in the legends of figures 2B, EV1 M.

- Please define the annotated p values ****/**/*/* as well as provide the exact p-values for the same in the legend of figure 5E, EV4 I, J as appropriate.

- Please note that the exact p values are not provided in the legends of figures 1D-F; EV1 I, M.

- Please indicate the statistical test used for data analysis in the legends of figure 2B.

- Please add to each legend (main, EV and Appendix figures, where applicable) a 'Data Information' section (or name the provided section like this) explaining the statistics used or providing information regarding replicates and scales. See:

- The Data Availability Section (DAS) is restricted to information regarding large primary datasets deposited at external databases. Thus, please remove the sentence 'All other data will be made available by the corresponding authors upon reasonable request' from the DAS. Moreover, please provide direct links to the deposited datasets and make sure the datasets are public latest upon online publication of the paper.

- Thank you for providing the requested source data. Please upload this as one folder per figure (with all files for one figure in one folder and ZIPed together).

In addition, I would need from you:

Please use this link to submit your revision: <https://embor.msubmit.net/cgi-bin/main.plex>

Best,

Referee #1:

The authors have adequately addressed my comments.

Referee #2:

I would like to congratulate the authors for an improved manuscript. However, I fear 2 of my key comments (out of 3) were not (sufficiently) addressed, although they represent a combination of approaches/techniques that the authors have already used. As stated previously, I think the manuscript is timely and relevant. When the findings of the authors are substantiated as highlighted before and again outlined below, this will be an interesting study for the community.

Comment concerning previous Major comment 1:

I appreciate that the depletion of neutrophils is technically challenging for longer than 2 weeks and metastasis may not have been established yet. However, that does not influence the ability of the authors to measure fibronectin accumulation and CD8+ T cell numbers in neutrophil-depleted lungs of tumour-bearing WT or UPP1-/- mice, or does it? As the title of the manuscript reads, the authors propose that neutrophils change the lung environment - this should be measurable in the lung before establishment of metastasis. Moreover, do fibroblasts express Upp1 (as asked before)? This would complete the picture.

Comment concerning previous Major comment 2:

I also appreciate that newly genetically-engineered mouse models are beyond the scope of this manuscript - therefore I suggested the use of the BAU inhibitor in combination with neutrophil depletion in the cancer models that the authors already used - which seems very feasible. Unfortunately, at this point, the main finding of the manuscript, that neutrophil-derived Upp1 has pro-metastatic effects is not very convincing. It relies mainly on one mouse model, which does not show a very pronounced phenotype. To substantiate their findings, the authors should demonstrate neutrophil-dependent anti-cancer effects of Upp1 blockade or deficiency at least in a second mouse cancer model (which is generally accepted practice). If the authors hypothesize that this is lung specific, or specific to breast cancer metastasizing to the lung, I would recommend the 4T1 (Balb/c background) or LLC (C57Bl6 background) cell lines in combination with Upp1 inhibition using BAU and neutrophil-depletion by anti-Ly6G.

Minor comment: The authors invested a notable amount of work to generate data that are only provided for the attention of the reviewers. Is there a reason not to include them in the manuscript? It seems a pity.

Referee #3:

The authors have satisfactorily addressed all the reviewers' comments and we feel that the paper is now suitable for publication in EMBO reports as it is. Thanks to the editor and congratulations to the authors for such an interesting study!

Referee #1:

The authors have adequately addressed my comments.

We thank Referee 1 for all the constructive suggestions through the revision process and we are glad they agree that we have addressed all comments. We feel the manuscript has been significantly improved by addressing the points that they raised.

Referee#2:

I would like to congratulate the authors for an improved manuscript. However, I fear 2 of my key comments (out of 3) were not (sufficiently) addressed, although they represent a combination of approaches/techniques that the authors have already used. As stated previously, I think the manuscript is timely and relevant. When the findings of the authors are substantiated as highlighted before and again outlined below, this will be an interesting study for the community.

Comment concerning previous Major comment 1:

I appreciate that the depletion of neutrophils is technically challenging for longer than 2 weeks and metastasis may not have been established yet. However, that does not influence the ability of the authors to measure fibronectin accumulation and CD8+ T cell numbers in neutrophil-depleted lungs of tumour-bearing WT or UPP1-/- mice, or does it? As the title of the manuscript reads, the authors propose that neutrophils change the lung environment - this should be measurable in the lung before establishment of metastasis. Moreover, do fibroblasts express Upp1 (as asked before)? This would complete the picture.

We appreciate the concerns of Reviewer 2, and did indeed assess metastasis, CD8 T cells and fibronectin in the lungs of neutrophil-depleted mice as requested. However, in performing these experiments it became clear that this experiment cannot directly answer this question for the following reason:

- To discriminate differences in lung metastases in FVB:PyMT+ mice it is necessary to conduct the analysis at clinical endpoint, as micrometastases are not detectable prior to this. However, we are technically constrained by the length of time that we can sustain neutrophil depletion in FVB mice. Thus, to ensure mice reach clinical endpoint with detectable metastases (and increased circulating uracil), we were limited in initiating neutrophil depletion two weeks prior to clinical endpoint. Although a 2-week neutrophil depletion significantly suppressed circulating uracil concentrations (Figure 2G), this was insufficient to alter fibronectin deposition, T-cell numbers or metastasis in the lung.

Our interpretation of this result is that the contribution made by neutrophils to metastatic niche priming manifests early in the metastatic cascade (i.e. prior to metastatic seeding), and so our intervention was too late to influence priming that had already occurred. Therefore, to determine the consequences of neutrophil depletion on the fibronectin deposition and T-cell landscapes associated with metastatic niche priming, it will be necessary to deplete neutrophils prior to metastatic seeding (from 8 weeks of age) to clinical endpoint (14 weeks of age), and we are unable to do this in this mouse model. The only suitable option to address this experimentally is to use neutrophil specific *Upp1* knock-out mice, which are not currently available to us.

On reflection, whilst we can conclusively show that *Upp1*-expressing neutrophils are a significant source of circulating uracil, it is perhaps premature to fully ascribe the role *Upp1* in metastasis to altered neutrophil function. With this in mind, and in agreement with Reviewer 2, we have changed the title of the manuscript, and the tone of the abstract to “Uridine Phosphorylase-1 supports metastasis by altering immune and extracellular matrix landscapes”. We feel this is a precise description of this exciting body of work, which shows for the first time that *Upp1* can influence niche-priming, and metastasis.

“Moreover, do fibroblasts express Upp1 (as asked before)? This would complete the picture.”

Fibroblasts do express detectable amounts of *Upp1*. To investigate potential roles of fibroblast-expressed UPP1 in extracellular matrix deposition, we have used CRISPR to delete *Upp1* from mouse fibroblasts and performed experiments as described in Figure 5D. Importantly, these experiments are designed to allow us to discriminate differences between cell behaviours influenced by cell intrinsic *Upp1* expression from those evoked by exogenously added uracil. We find that it is only exogenous uracil which encourages fibroblasts to deposit a pro-migratory extracellular matrix, whilst their intrinsic levels of UPP1 were ineffective in this regard and now include this in the revised manuscript as Appendix Figure S9. We also extensively showed in our previous response to Reviewer 3’s comments that UPP1 expression is not upregulated in cancer-associated fibroblasts in comparison to normal fibroblasts. Therefore, although fibroblasts can express UPP1, we do not have any evidence to suggest that this could contribute to the metastasis-associated phenotypes described in our manuscript. These data, and discussion of them, have now been added to the final manuscript (Appendix Table S4).

*Comment concerning previous Major comment 2:
I also appreciate that newly genetically-engineered mouse models are beyond the scope of this manuscript - therefore I suggested the use of the BAU inhibitor in combination with neutrophil depletion in the cancer models that the authors already used - which seems very feasible. Unfortunately, at this point, the main finding of the manuscript, that neutrophil-derived Upp1 has pro-metastatic effects is not very convincing. It relies mainly on one mouse model, which does not show a very pronounced phenotype. To substantiate their findings, the authors should demonstrate neutrophil-dependent anti-cancer effects of Upp1 blockade or deficiency at least in a second mouse cancer model (which is generally accepted practice). If the authors hypothesize that this is lung specific, or specific to breast cancer metastasizing to the lung, I would recommend the 4T1 (Balb/c background) or LLC (C57Bl6 background) cell lines in combination with Upp1 inhibition using BAU and neutrophil-depletion by anti-Ly6G.*

Assessment of mechanisms that influence early niche-priming at clinical endpoint are challenging, due to metastasis that can occur from alternative/late dissemination routes. Thus a 33% reduction in the proportion of mice that succumb to metastasis at clinical endpoint, as a result of inhibition of early metastatic niche priming, is highly significant and biologically relevant.

Regarding pharmacological inhibition of UPP1 with BAU, we chose to use genetic knockout of *Upp1* to ensure consistent and, importantly, specific targeting of our gene of interest throughout the lifetime of the animals.

Regarding the 4T1 and LLC experiments proposed by this reviewer, we thank them for their suggestions, but we do not feel that these are appropriate methods to address the question posed, for the following reasons:

- Neutrophils significantly infiltrate primary tumours in the 4T1 model (PMID 27798263), making it difficult to separate primary tumour and metastatic effects using this approach. Furthermore, neutrophilia in the 4T1 model is also dependent on G-CSF (PMID 27072748), which can skew the subtype of neutrophils mobilised (PMID 36801950).
- In the context of MMTV-PyMT driven tumours, a C57Bl6 background (as suggested for the LLC approach) does not recapitulate the same metabolic traits that we see in human breast cancer patients. Thus, the MMTV-PyMT model of mammary cancer recapitulates key aspects of the human breast cancer circulating metabolome when conducted on an FVB mouse background, but not in C57Bl6 mice. Moreover, we find that the incidence of metastases from MMTV-PyMT mammary tumours is very low in C57Bl6 mice. Taken together these observations indicate that the C57Bl6 mouse strain is not a good background in which to study metastasis of mammary tumours and its associated metabolic landscapes.
- Tail vein injections of lung cancer cells (LLC) is not an appropriate method to assess niche priming and consequent metastasis of (mammary) tumours to the lungs.

Regarding the number of mouse models used, our uracil observations have been observed in 4 distinct models (PyMT mammary cancer, KPC pancreatic cancer, KPN colon cancer, LPS immune challenge), and our mechanistic work on neutrophil motility and immune landscape have been performed in two different mammary cancer models (KP transplant and PyMT mammary). To substantiate our findings, in what we felt was the most appropriate way, we invested in fully understanding the ability of UPP1 to influence progression of a second cancer type – pancreatic cancer - as previously suggested by Reviewer 2. To do this rigorously, we crossed *Upp1*^{+/+} and *Upp1*^{-/-} animals with KPC mice. This indicated that *Upp1* had a tendency to influence metastasis and significantly affected invasion of pancreatic tumours at the local site (Figure EV6F-G). This strongly supports a role for UPP1 in behaviours that are important for driving early metastatic seeding in an alternative mouse model, and in another cancer type.

Minor comment: The authors invested a notable amount of work to generate data that are only provided for the attention of the reviewers. Is there a reason not to include them in the manuscript? It seems a pity.

We were keen to address all comments as best we could with new experimental data. Omission of data from the main manuscript was only to keep the story streamlined and not overwhelmed by large amounts of supplementary data. We have now incorporated all Reviewer Response Figures in the final manuscript. Please note, this has resulted in inclusion of an additional EV figure, and additional appendix figures, and so EV and Appendix figure numbers in the final manuscript may now differ from those references in the first response to reviewer comments.

Referee #3:

The authors have satisfactorily addressed all the reviewers' comments and we feel that the paper is now suitable for publication in EMBO reports as it is. Thanks to the editor and congratulations to the authors for such an interesting study!

Many thanks to Referee #3 for their enthusiasm and constructive comments that have significantly improved our revised manuscript.

Dr. Cassie Clarke
Cancer Research UK Scotland Institute
Garscube Estate
Switchback Road
Glasgow, Glasgow G61 1BD
United Kingdom

Dear Dr. Clarke,

Thank you for the submission of your final revised manuscript to our editorial offices. I now went through it and your final p-b-p-response and consider the remaining points by referee #2 as adequately addressed.

I thus am very pleased to accept your manuscript for publication in the next available issue of EMBO reports. Thank you for your contribution to our journal.

Yours sincerely,
